# Target cell cortical tension regulates macrophage trogocytosis

Caitlin E. Cornell[1], Aymeric Chorlay[1], Deepak Krishnamurthy[1], Nicholas R. Martin[2], Lucia Baldauf[3] & Daniel A. Fletcher ●[1,4,5,6] ✉

Macrophages are known to engulf small membrane fragments, or trogocytose, target cells and pathogens, rather than fully phagocytose them. However, little is known about what causes macrophages to choose trogocytosis versus phagocytosis. Here we report that cortical tension of target cells is a key regulator of macrophage trogocytosis. At low tension, macrophages will preferentially trogocytose antibody-opsonized cells, while at high tension, they tend towards phagocytosis. Using model vesicles, we demonstrate that macrophages will rapidly switch from trogocytosis to phagocytosis when membrane tension is increased. Stiffening the cortex of target cells also biases macrophages to phagocytose them, a trend that can be countered by increasing antibody surface density and is captured in a mechanical model of trogocytosis. This work suggests that the target cell, rather than the macrophage, determines whether phagocytosis or trogocytosis occurs, and that macrophages do not require a distinct molecular pathway for trogocytosis.

Phagocytosis by macrophages, first observed by Metchnikoff more than 140 years ago[1], plays a critical role in the immune system. Upon binding and recognition of unhealthy cells or foreign particles, macrophages can fully engulf and destroy the targets, contributing to the clearance of bacterial infections, cell corpses and, more recently, immunotherapy-targeted cancer cells[2,3]. One of the best-characterized mechanisms for targeting a particle for engulfment by macrophages is antibody-dependent cellular phagocytosis. Once labelled with antibodies, pathogens and other cells in the body are recognized by $F_c$ receptors on the macrophage surface, triggering a signalling cascade that results in growth and closure of an actin-driven phagocytic cup around the target, typically followed by destruction after internalization.

More recently, macrophages have been observed to sometimes simply 'nibble' portions from the target cell membrane rather than fully engulf the target[4,5]. This peculiar process, termed 'trogocytosis' (*Trogo-*; Greek for 'to nibble'), is emerging as an important effector function on its own that can both enhance and undermine immune cell function[6]. For example, human and mouse macrophages have

been observed to nibble hematopoietic stem cells, marking them for retention in the bone marrow[7]. In zebrafish, hematopoietic stem cells are also selectively nibbled by macrophages, marking them for clonal hematopoiesis[8]. Similarly, tissue resident macrophages of the brain, microglia, selectively nibble synapses of neurons to prune connections in the developing mouse brain[9]. Strikingly, macrophages have also been shown to trogocytose targets that are too large to phagocytose, such as the parasitic worm *Schistosoma*[10], or large tumours[11]. However, trogocytosis is sometimes an undesirable outcome, especially in the innate immune response to cancer immunotherapies. Instead of phagocytosing cancer cells targeted with therapeutic antibodies, macrophages will remove the antibodies from the cell surface via trogocytosis, leaving the cancer cells alive[4,12–14].

Beyond the mammalian immune system, trogocytosis has been observed in cell types as diverse as the pathogenic amoeba *Entamoeba histolytica* and the endodermal cells of *Caenorhabditis elegans*. The role of trogocytosis in these organisms is similarly diverse. For pathogenic *Entamoeba*, amoeba will first kill a cell via trogocytosis and then fully

[1]Department of Bioengineering, University of California, Berkeley, Berkeley, CA, USA. [2]Cardiovascular Research Institute, University of California, San Francisco, San Francisco, CA, USA. [3]London Centre for Nanotechnology, University College London, London, UK. [4]Graduate Group in Bioengineering, University of California, Berkeley, and University of California, San Francisco, San Francisco, CA, USA. [5]Division of Biological Systems and Engineering, Lawrence Berkeley National Laboratory, Berkeley, CA, USA. [6]Chan Zuckerberg Biohub, San Francisco, CA, USA. ✉e-mail: fletch@berkeley.edu

phagocytose their deceased target[15]. Interestingly, the amoeba will subsequently display protein from the target cell, a process known as 'cross-dressing', cloaking themselves from host immune attack[16]. Trogocytosis plays a gentler homeostatic role in *C. elegans*; lobes rich in mitochondria on primordial germ cells are pruned by endodermal cells nearby, potentially reducing the risk of reactive oxygen species in the germline[17].

Despite the importance of trogocytosis, the signals on the target cell that drive trogocytosis over phagocytosis or other cell–cell interactions are poorly understood. It is unlikely that this signal is a conserved cell surface molecule, antibody or soluble cytokine, as diverse target cells in a wide range of physiological systems appear to be able to promote trogocytosis. It is also unlikely to be a purely stochastic phenomenon, as not all cell–cell interactions—for example, between macrophages and bacterial cells—lead to trogocytosis as well as phagocytosis. For the primordial germ cells of *C. elegans*, cellular markers of distress (for example, reactive oxygen species) seem to mark cells for trogocytosis[17]; however, this has not been shown to be the case for microglia or pathogenic amoeba.

Here, we find that the physical properties of the macrophage–target interface, rather than specific molecular components of the target surface, drive the macrophage's decision to trogocytose rather than phagocytose. We quantitatively evaluate the trogocytic and phagocytic efficiency of macrophages interacting with antibody-labelled cells and membrane-only cell mimics using a combination of light microscopy, micropipette aspiration and flow cytometry. Using giant unilamellar vesicles (GUVs), we confirm that membrane tension is a sufficient signal to bias trogocytosis over phagocytosis by varying tension in the absence of surface molecules other than antibodies. We further demonstrate that trogocytic efficiency of antibody-labelled cells depends on target cortical tension and the density of antibodies coating the target cell surface. Finally, we propose a mechanical model of macrophage trogocytosis that accounts for the observed dependence on target surface tension and antibody density.

This work shows that a distinct molecular pathway is not required to explain why some target cells are trogocytosed rather than phagocytosed. Instead, the macrophage's ability to deform the target cell surface after binding is a sufficient signal to promote trogocytosis, suggesting that phagocytosis may be thought of as frustrated trogocytosis. The increase in trogocytosis with decreased cell cortical stiffness raises the possibility that tumour cells and other phagocytic targets could modulate their physical properties to evade phagocytosis.

## Results

### Macrophages phagocytose and trogocytose cultured cells

To study trogocytosis by macrophages, we developed a fluorescence-based assay that uses the key features of trogocytosis (surface engulfment) and phagocytosis (surface and volume engulfment) to separate the two processes. For a volume marker, we labelled the cytosol of target cells with the pH-sensitive rhodamine derivative pHrodo, which selectively fluoresces in low pH environments (for example, the phagolysosome). For a surface marker, we used a fluorescent antibody, AlexaFluor647 anti-CD47 IgG, that also activates macrophages through engagement with their Fcγ receptor (FcγR). CD47 is a marker of self and a potent 'don't eat me' inhibitory signal against macrophage phagocytosis, and antibody labelling attenuates its inhibitory effects. Macrophages that only trogocytose would only be marked by an AlexaFluor647 signal, while those that phagocytose (as well as those that phagocytose and trogocytose) would be marked by both an internalized pHrodo signal and AlexaFluor647 signal.

When we mix target Jurkat T cells with RAW 264.7 macrophage-like cells, we observe two clear phenotypes by fluorescence microscopy. Some macrophages fully engulf the labelled Jurkat T cells, with surface antibody signals and volume signals observed to be colocalized in the phagolysosome of the macrophage, which are labelled with

CellTracker Green CMFDA (Fig. 1a, bottom). Other macrophages internalize small 'bites' from the surface of Jurkat T cells that are positive for the anti-CD47 antibody surface signal but negative for the volume marker, pHrodo (Fig. 1a, top). From fluorescence microscopy images of macrophages incubated with opsonized Jurkat T cells for 1 h, we found that a large proportion of macrophages nibbled target cells, while full phagocytosis events were quite rare (Extended Data Fig. 1).

To test this observation with larger populations, we developed a flow cytometry assay to quantify trogocytic and phagocytic efficiency. We mixed cytosol-labelled macrophages with surface- and volume-labelled target cells and analysed the populations of macrophages that contained both target surface and volume signals (indicating phagocytosis), only the surface signal (indicating trogocytosis only) or neither signal. We define phagocytic and trogocytic efficiency as the percentage of the total number of macrophages that have undergone phagocytosis or trogocytosis, respectively.

Consistent with the microscopy experiments, we find that trogocytic events are notably more frequent than phagocytic events when target cells are opsonized with anti-CD47 (Fig. 1b). We estimate that trogocytosis is 3–4× more frequent than phagocytosis in our 1-h experiments with a 1:1 ratio of macrophages to target cells.

To test whether trogocytosis was dependent on CD47 labelling or could occur with labelling of other cell surface proteins, we non-specifically biotinylated the primary amines of proteins on the surface of target cells using an NHS-ester biotin compound. We then opsonized target cells with anti-biotin, labelled them and mixed them with macrophages. We found that trogocytic efficiency was nearly identical to that for the anti-CD47 labelling at the same antibody concentration, indicating that trogocytosis is not dependent on the surface molecule labelled (Fig. 1b).

To ensure that the trogocytosis we measure is dependent on FcγR binding to the antibody, we measured the trogocytic and phagocytic efficiency of macrophages first treated with an FcγR-blocking antibody (mouse anti-CD16/32). Both trogocytosis and phagocytosis are inhibited when the macrophage FcγR is blocked by 0.04 μM antibody (Extended Data Fig. 2a,b). Similarly, we observed that biotinylated, but non-opsonized, target cells with AlexaFluor647 streptavidin-labelled membranes are not trogocytosed or phagocytosed, confirming that trogocytosis is dependent on FcγR engagement (Extended Data Fig. 2c).

We next explored whether the trogocytosis we observe is unique to macrophage-like RAW 264.7 cells, which are convenient to work with but have known differences from primary mouse macrophages and other macrophage cell lines[18,19]. We quantified the trogocytic and phagocytic efficiency of opsonized Jurkat T cells incubated with primary bone marrow-derived macrophages (BMDMs), lipopolysaccharide (LPS)-stimulated RAW 264.7 cells and another immortalized mouse macrophage cell line, J774a.1 cells. We observed a similar trogocytic efficiency across all four macrophage cell lines (Fig. 1c). This shows that both the ability to trogocytose and the trogocytic efficiency are not strongly affected by macrophage cell type.

### Trogocytosis differs for target cells depending on their cortical tension

While different macrophages trogocytose the same target cell with similar efficiency, does the same macrophage trogocytose different target cells with similar efficiency? To test this, we used a panel of cultured cells derived from blood cancers, namely, Jurkat T cells, HL60 cells and Raji B cells. We opsonized the target cells with anti-CD47, which is highly expressed on all three cell types (Extended Data Fig. 3), and incubated them with RAW 264.7 macrophages in separate experiments. Interestingly, we see large differences in trogocytic and phagocytic efficiency for the different target cell lines, with the trogocytic efficiency of HL60 and Raji B cells at only ~20%, compared with ~70% for Jurkat cells (Fig. 2a). Notably, this is also the case for LPS-stimulated macrophages, especially for the filamin-deficient melanoma cell line

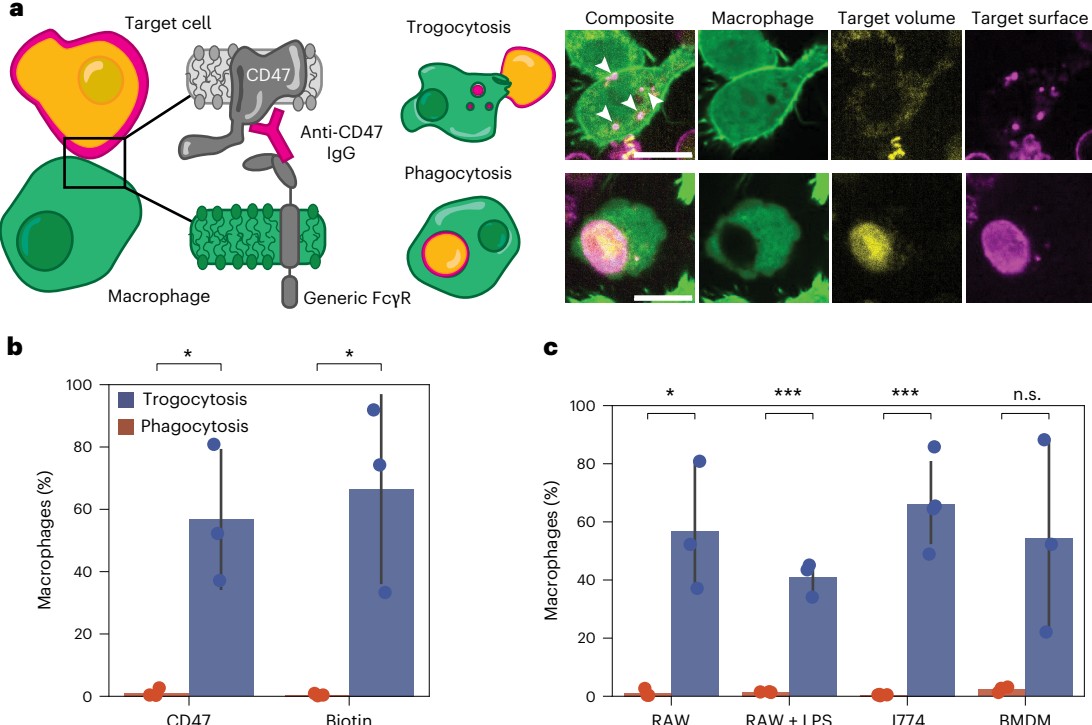

**Fig. 1 | Macrophages trogocytose and phagocytose Jurkat T cells.**
**a**, Left: Jurkat T cells are labelled with pH-sensitive fluorophore pHrodo and fluorescent (AlexaFluor647) anti-CD47, which binds to the FcγR on CellTracker Green CMFDA-labelled macrophages, initiating trogocytosis or phagocytosis. Right: Trogocytosis (top) is characterized by internalization of small, punctate 'bites' (marked with white arrowheads) that are positive for the anti-CD47 signal only. Phagocytosis (bottom) is characterized by colocalization of pHrodo and antibody signal in the phagolysosome of macrophages. Scale bars, 10 μm.
**b**, Cytometric-based analysis of the trogocytic and phagocytic efficiency of RAWs challenged with Jurkat T cells for 1 h, opsonized with anti-CD47 and anti-biotin, respectively. *n* = 3 independent experiments pooling together

$10^5$ RAWs per experiment co-incubated with Jurkat T cells opsonized with anti-CD47, and $10^5$ RAWs per experiment co-incubated with Jurkat T cells opsonized with anti-biotin, respectively. **c**, Trogocytic and phagocytic efficiency of RAW 264.7 cells, BMDMs, J774a.1 cells and LPS-stimulated RAW 264.7 cells challenged with anti-CD47 opsonized Jurkat T cells. *n* = 3 independent experiments pooling together $10^5$ RAWs per experiment, LPS-stimulated RAWs, J774 cells or BMDMs co-incubated with Jurkat T cells opsonized with anti-CD47, respectively. Error bars are drawn around the mean and represent the standard deviation between three independent experiments. Significance was determined using a two-sided Student's *t*-test. Asterisks indicate the *P* values. n.s., non-significant. *$P$ < 0.05, ***$P$ < 0.001.

M2. Strikingly, while LPS-stimulated macrophages phagocytose HL60 and Jurkat cells at a statistically similar amount, the ratio of trogocytosis to phagocytosis is 2× higher for Jurkat cells than for HL60 cells.

As Jurkat T cells have approximately 2.2× more CD47 on their cell surface than HL60 and Raji B cells (Extended Data Fig. 3), differences in trogocytic efficiency could be due to differences in antibody surface density. We altered antibody solution concentrations to achieve the same surface density of anti-CD47 across cell types (Fig. 2b), and we tested non-specific labelling of the target cells by biotinylating and opsonizing them with anti-biotin (Fig. 2c). When antibody surface density is matched, we observe that trogocytic efficiency is the highest in Jurkat cells, while phagocytic efficiency is the highest in HL60 cells. To ensure that our results are not simply from blocking the 'don't eat me' interaction of target CD47 with macrophage SIRPα (due to anti-CD47 or biotinylation of CD47), we co-incubated macrophages with Raji B cells opsonized with anti-CD19. We observed that Raji B cells opsonized with anti-CD19 show ~4× more trogocytosis than phagocytosis, which is comparable to the results shown in Fig. 2b, albeit with an overall 2× lesser magnitude (Extended Data Fig. 4).

As trogocytosis involves pinching off small (<1 μm) bits of membrane, one potentially relevant difference between the target cells could be the deformability of their cell surface. This deformability is dependent not only on membrane tension but also on the strength and abundance of linkages between the membrane and underlying cytoskeleton. Together, the resistance to deformation can be captured by 'cortical tension', often measured with micropipette aspiration[20–22].

We quantified the cortical tension of antibody-labelled target cells using a home-built micropipette aspiration system with an in-line pressure sensor. Using glass pipettes with an inner diameter of ~5 μm, we applied suction to individual cells until a 'tongue' was pulled into the syringe to a consistent length of ~1.5 μm (Fig. 2d). We then quantified cortical tension of the cell using equation (1), the Young–Laplace law

$$\gamma = \frac{\triangle P_{\text{suction}}}{2\left(\frac{1}{R_{\text{pipette}}} - \frac{1}{R_{\text{cell}}}\right)}, \tag{1}$$

where $\Delta P_{\text{suction}}$, $R_{\text{pipette}}$ and $R_{\text{cell}}$ are the suction pressure, the pipette radius and the cell radius, respectively.

We find that the cortical tension of HL60 cells, which are phagocytosed more than the other cells, is notably higher than that of the Jurkat T cells and Raji B cells (Fig. 2e). If we take the average trogocytic efficiency of target cells with matched anti-CD47 surface densities, the trogocytic efficiency for target cells decreases (Pearson correlation coefficient of −0.71) with increasing cell tension (Fig. 2f).

## Membrane tension of a cell mimic drives trogocytosis over phagocytosis

The cortical tension we measured for cells is a combination of membrane tension and interactions with the cortical cytoskeleton[23,24]. We wondered whether membrane tension alone could be used to alter macrophage trogocytosis, independent of any cytoskeletal interactions.

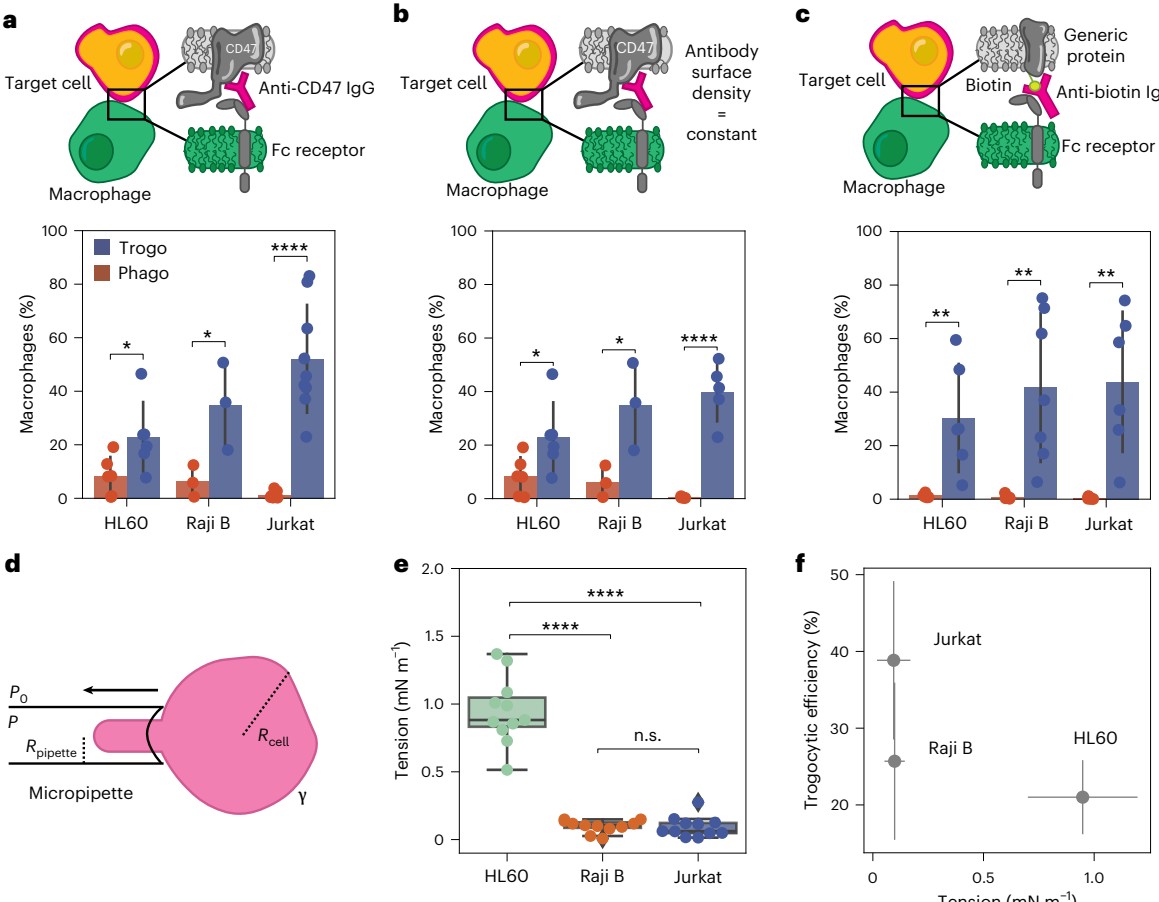

**Fig. 2 | Trogocytosis depends on cell cortical tension. a**, Cytometric-based analysis (bottom) of the phagocytic and trogocytic efficiency of RAWs challenged with Jurkat T cells, Raji B cells and HL60 cells opsonized with anti-CD47 (top) at a surface density of 336 ± 174 molecules per μm². *n* = 5 independent experiments pooling together 10⁵ RAWs per experiment co-incubated with HL60 cells, *n* = 3 independent experiments pooling together 10⁵ RAWs per experiment co-incubated with Raji B cells and *n* = 8 independent experiments pooling together 10⁵ RAWs per experiment co-incubated with Jurkat T cells. **b**, Cytometric-based analysis (bottom) of phagocytic and trogocytic efficiency of RAWs challenged with Jurkat T cells, Raji B cells and HL60 cells opsonized with anti-CD47 solution (top) matched to achieve equivalent surface coverage of antibody (300–500 antibodies per μm²). *n* = 5 independent experiments pooling together 10⁵ RAWs per experiment co-incubated with HL60 cells, *n* = 3 independent experiments pooling together 10⁵ RAWs per experiment co-incubated with Raji B cells and *n* = 5 independent experiments pooling together 10⁵ RAWs per experiment co-incubated with Jurkat T cells. **c**, Phagocytic and trogocytic efficiency (bottom) of RAWs challenged with Jurkat T cells, Raji B cells and HL60 cells with surface proteins non-specifically labelled by *N*-hydroxysuccinimide (NHS) biotin and then opsonized with anti-biotin (top) at a surface density of 683 ± 551 molecules per μm². *n* = 6 independent experiments pooling together 10⁵ RAWs per experiment co-incubated with HL60 cells, *n* = 7

independent experiments pooling together 10⁵ RAWs per experiment co-incubated with Raji B cells and *n* = 6 independent experiments pooling together 10⁵ RAWs per experiment co-incubated with Jurkat T cells. Error bars are drawn around the mean and represent the standard deviation between the independent experiments. Statistical significance was determined using a two-sided Student's *t*-test, and asterisks represent *P* values (see below). **d**, Micropipette aspiration set-up to measure cell tension. We measure $R_{cell}$, $R_{pipette}$ and *P* to calculate the surface tension, γ. $P_0$ is the atmospheric pressure. **e**, Tension of Raji B cells, HL60 cells and Jurkat T cells. *n* = 3 independent experiments pooling together 11 HL60 cells, 10 Raji B cells and 11 Jurkat T cells. There is a significant difference (calculated using one-way analysis of variance (ANOVA) followed by a Tukey pairwise test) between the tension of HL60, Raji B and Jurkat T cells, and asterisks represent *P* values (see below). Box plots represent the median (line) and interquartile range (IQR, box), and the whiskers extend to the smallest and largest values within 1.5× IQR. **f**, The trogocytic efficiency of target cells from Fig. 2b is negatively correlated with target cell tension (Pearson's correlation coefficient of −0.71). *n* = 5 independent experiments pooling together 10⁵ RAWs per experiment co-incubated with HL60 cells, *n* = 3 independent experiments pooling together 10⁵ RAWs per experiment co-incubated with Raji B cells and *n* = 5 independent experiments pooling together 10⁵ RAWs per experiment co-incubated with Jurkat T cells. n.s., non-significant. *P* < 0.05, **P* < 0.01, ****P* < 0.0001.

To test this, we created target particles with only membrane using GUVs composed of palmitoyl oleoyl phosphatidylcholine (POPC), 1 mol% biotin dioleoyl phosphatidylethanolamine (biotin-DOPE) and 0.5 mol% lissamine rhodamine phosphatidylethanolamine (liss rho PE). The GUVs were formed in a sucrose medium that was osmotically tuned using an osmometer to cause the GUVs to sink but remain isotonic in the cell culture media. To obtain GUVs of a consistent size and avoid the con-founding effects of small vesicles formed at the same time as the GUVs, we developed a home-built differential density column (SeparatorMax 3000; Extended Data Fig. 5) to separate the GUVs on the basis of size. We collected fractions of GUVs in the size range of 5–20 μm in diameter

and opsonized them with fluorescent anti-biotin at a surface density of ~400 antibodies per μm (ref. 2) for a 10-μm GUV.

We incubated opsonized GUVs with macrophages and observed by confocal microscopy that macrophages can both fully engulf GUVs and partially nibble them (Fig. 3a). When a GUV is trogocytosed, we observe colocalization of the lipid dye and the fluorescent antibody in trogocytic bites, indicating that the macrophages extract small patches of membrane from GUVs rather than detach the anti-biotin AF647 antibody. These initial experiments showed that antibody-opsonized GUVs can be both trogocytosed and phagocytosed, similar to target cells.

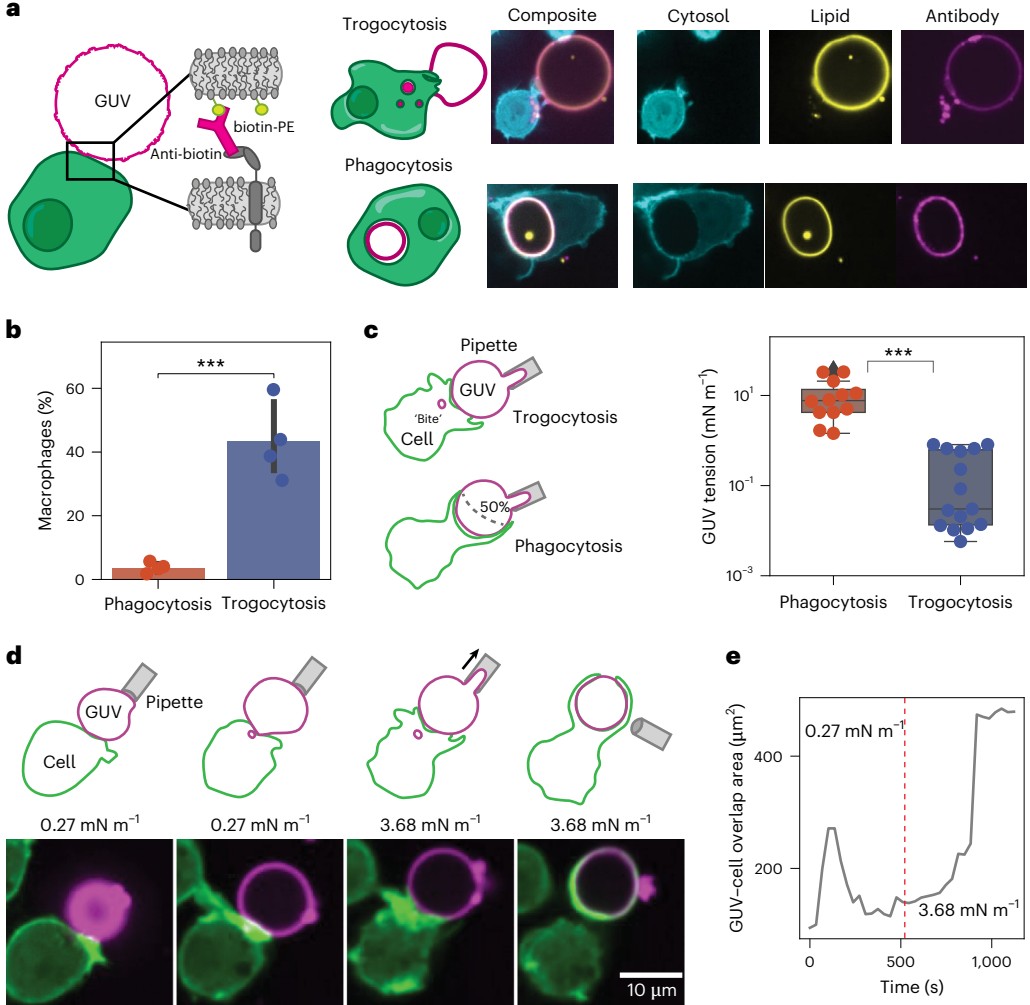

**Fig. 3 | Macrophages trogocytose GUVs. a**, GUVs composed of POPC, biotin-DOPE and liss rho PE are opsonized with AlexaFluor647 anti-biotin and mixed with macrophages (left). When GUVs are phagocytosed, the perimeter of a circular GUV can be observed within the macrophage phagolysosome (bottom right), and when GUVs are trogocytosed, punctate 'bites' can be observed within the macrophage (top right). **b**, Cytometric-based analysis of GUVs, showing that they are trogocytosed more frequently than they are phagocytosed by macrophages. Error bars are drawn around the mean and represent the standard deviation between $n = 4$ independent experiments, pooling together $10^5$ RAWs per experiment after co-incubation with anti-biotin opsonized GUVs. Significance was determined using a two-sided Student's $t$-test. Asterisks represent $P$ values. **c**, Tension (right) for GUVs that are phagocytosed (bottom left, determined by >50% surface coverage of the GUV by macrophage phagocytic extensions) in a high-tension regime (>1 mN m$^{-1}$) and GUVs that are trogocytosed (top left, determined by the presence of trogocytic 'bites' in the macrophage) in a low-tension regime (<1 mN m$^{-1}$). Significance was determined via a two-sided Student's $t$-test for 27 total vesicles pooled over $n = 3$ independent experiments, and asterisks represent $P$ values. Box plots represent the median (line) and IQR (box) for 12 phagocytosed vesicles and 15 trogocytosed vesicles. The whiskers extend to the smallest and largest values within 1.5× IQR. **d**, Schematic (top) and micrographs (bottom) of a GUV under low tension getting trogocytosed by a macrophage, followed by application of suction and an increase in tension, leading to full engulfment of the GUV by the macrophage. **e**, Time trace of the overlap area from one experiment between the macrophage channel (green) and the GUV channel (magenta). The dashed red line indicates when suction was applied to the GUV and tension was increased from 0.27 mN m$^{-1}$ (trogocytosis regime) to 3.68 mN m$^{-1}$ (phagocytosis regime). ***$P < 0.001$.

To quantify the trogocytic and phagocytic efficiency of macrophages incubated with GUVs, we loaded dark GUVs with a soluble rhodamine fluorophore and opsonized them with AlexaFluor647 anti-biotin. After 30 min of co-incubation with macrophages, we analysed the macrophages via flow cytometry and found that trogocytosis is a much more common event than phagocytosis, consistent with the target cells tested (Fig. 3b). We might have expected that all GUVs would either by trogocytosed or phagocytosed because they all have the same lipid composition and vary only minimally in size. However, there must be a difference—even in this minimal system—that it is sufficient to drive trogocytosis rather than phagocytosis.

Interestingly, GUVs created via electroformation are formed with a wide range of membrane tensions, even when they are 'isotonic' (Extended Data Fig. 6) due to the random formation of membrane pores[25]. To directly investigate the role of membrane tension in trogocytosis, we first used micropipette aspiration to measure the membrane tension of a population of GUVs. We incubated GUVs with macrophages in an imaging dish, allowed GUVs to settle and interact with the macrophages for 10 min; thereafter, we individually measured the membrane tension of GUVs that were being either phagocytosed or trogocytosed. We identified phagocytic events as GUVs whose surface was wrapped >50% by macrophage phagocytic extensions and trogocytic events as GUVs connected to macrophages in which small 'bites' were internalized in the macrophage. We chose >50% wrapping as the threshold for phagocytosis because the micropipette occupies one side of the GUV and because we and others[26] have found that >50% coverage leads to full engulfment >75% of the time. We found two distinct tension regimes: At tensions of 0.26 ± 0.33 mN m$^{-1}$, GUVs are

exclusively trogocytosed, while at tensions of $11.7 \pm 11.2$ mN m$^{-1}$, GUVs are phagocytosed (Fig. 3c).

These measurements show that membrane tension is a sufficient signal to drive macrophage trogocytosis or phagocytosis of GUVs. However, is that decision made at initial contact and activation, or does the macrophage dynamically sense membrane tension and alter its decision? To address this, we tested the response of macrophages to dynamic changes in membrane tension with the micropipette aspiration system and simultaneous confocal microscopy. We added opsonized GUVs to one side of an imaging chamber and selected a GUV with a low membrane tension (0.27 mN m$^{-1}$, in the trogocytosis regime). Upon presenting the GUV to a macrophage, we observed the macrophage forming filopodial extensions but then quickly retracting and beginning to internalize small 'bites' of a GUV membrane. Using the micropipette, we then applied suction to the GUV, increasing membrane tension above the threshold for phagocytosis (3.68 mN m$^{-1}$, in the phagocytosis regime). Within 3 min, the macrophage rapidly extended a phagocytic cup around the GUV and engulfed it completely within a matter of minutes (Fig. 3d,e and Supplementary Video 1; see Supplementary Videos 2–4 for more examples). These experiments show that membrane tension alone is a significant driver of trogocytosis in macrophages, and changes in membrane tension can act as a switch between phagocytosis and trogocytosis.

### Increasing tension in cultured cells suppresses trogocytosis and increases phagocytosis

Our experiments with cultured cells and GUVs indicate that the deformability of the target surface is sufficient to control whether the target is phagocytosed or trogocytosed. This raises the possibility that intentionally stiffening the cortex of cells could suppress trogocytosis and promote phagocytosis. Interestingly, cells that have undergone apoptosis are significantly stiffer (up to 3×) than their healthy counterparts[27–29]. Macrophages are responsible for efferocytosis, or the phagocytic clearance of apoptotic cells. We hypothesized that the increase in stiffness upon apoptosis would bias macrophages towards phagocytosis over trogocytosis.

To induce apoptosis in Jurkat T cells, we used raptinal, a potent activator of caspase-3, which induces apoptosis within 30 min of incubation, resulting in >80% cell death (as measured by propidium iodide) within 2 h (refs. 30,31). In the presence of 10 µM raptinal for 2 h, we measured 85% of Jurkat T cells dead as marked by propidium iodide (in contrast to 1% of untreated Jurkat T cells). After co-incubation with LPS-stimulated macrophages, we saw a 3× increase in phagocytosis (and a 15× decrease in trogocytosis) over healthy cells when Jurkat cells were opsonized with IgG (Fig. 4a, left). Because apoptotic cells have additional 'eat me' signals (for example, phosphatidylserine lipids flipped to the outer leaflet of the plasma membrane), we repeated the experiment without IgG opsonization (Fig. 4a, right). To measure trogocytosis without a fluorescent IgG, we biotinylated the target cells and then labelled the membrane with AlexaFluor647 streptavidin (and dark anti-biotin for opsonized cells). In Fig. 4a (right), phagocytosis and trogocytosis in both raptinal-treated and -untreated Jurkat cells are attenuated; however, phagocytosis is still higher for raptinal-treated Jurkat cells.

To further investigate the role of cortical tension in trogocytosis of living cells, we co-incubated LPS-stimulated macrophages with a melanoma-derived cell type, M2. M2 cells are deficient in filamin A, a crosslinker of F-actin at the cortex. The cortex of M2 cells is weakened and, as a result, they are measurably softer than similar cells expressing filamin A[32,33]. In comparison with Jurkat T cells and HL60 cells, M2 cells opsonized at the same surface density of anti-CD47 are trogocytosed significantly more (Fig. 4c), consistent with our findings that lower cortical tension results in greater trogocytosis.

We next sought to test whether controlled stiffening of the cortex would allow us to titrate the amount of phagocytosis we observed.

Previous work has shown that gentle glutaraldehyde fixation leads to measurable differences in cortical tension[34], with concentrations of glutaraldehyde below 0.01% leaving cells viable. We treated opsonized target cells with either 0%, 0.002% or 0.0025% glutaraldehyde for 30 min. At these fixation concentrations, anti-CD47 density on treated target cells is approximately the same as untreated cells ($300–500 \pm 100$ antibodies per µm$^2$).

To confirm the effects of gentle fixation on cortical tension, we measured treated target cells using micropipette aspiration. We chose HL60 and Jurkat cells because they engage in the least and most trogocytosis, respectively. Cortical tensions for treated cells increased above the tension regime we measured for trogocytosis of GUVs, especially for Jurkat and HL60 cells treated with 0.0025% glutaraldehyde (Fig. 4d). To quantify the trogocytic efficiency of macrophages with the opsonized and stiffened target cells, we washed and incubated them with macrophages for 1 h and then analysed them with flow cytometry. We found that trogocytosis was almost eliminated for cells treated with 0.0025% glutaraldehyde. By contrast, phagocytosis increased significantly for cells treated with 0.0025% glutaraldehyde (Fig. 4e).

As we previously observed that Jurkat cells, before antibody surface density matching with HL60 and Raji B cells, had greater trogocytic efficiency with greater antibody coverage (2.2× higher than HL60 and Raji B cells), we wondered whether we could rescue trogocytosis of the stiffened cells by increasing antibody surface density. When we titrated antibody surface density for gently fixed Jurkat and HL60 cells, we found that trogocytic efficiency was indeed increased at the highest antibody densities (Fig. 4f). Interestingly, trogocytic efficiency is rescued at much higher concentrations of antibody than for wild type target cells. Not only does it take more antibody to induce trogocytosis of gently fixed cells, but the trogocytic efficiency is never greater than what was achieved for wild type target cells at lower antibody densities (Extended Data Fig. 7). This suggests that we are unable to coat the surface sufficiently to rescue the full trogocytic efficiency of a soft wild type target.

### A mechanical model captures the relationship between cell tensions and antibody surface density in trogocytosis

Our experiments indicate that both cortical tension and antibody density play important roles in driving trogocytosis of cellular targets. How might the two interact to control interface mechanics between the macrophage and target cell? To gain intuition, we develop a simple scaling relationship for the interfacial mechanics. We assume the target antibody binds to macrophage FcγRs at a high-enough concentration to trigger downstream signalling machinery that leads to cytoskeleton-mediated active stresses at the interface (Fig. 5a). On the basis of prior experimental measurements, these stresses take the form of both extensile and compressive normal stresses on the target membrane[35]. We further assume that these stresses scale with the antibody density on the target (Supplementary Note). These normal stresses will tend to deform the interface, and this deformation is resisted by the target cortical tension. Balancing the forces due to active stresses and cortical tension at the interface, one gets equation (2)

$$\sigma_{\text{normal}}(\rho_{\text{AB}})R_{\min}^2 \approx \gamma_t R_{\min}. \tag{2}$$

This gives the length scale equation (3)

$$R_{\min} \approx \frac{\gamma_t}{\sigma_{\text{normal}}(\rho_{\text{AB}})}, \tag{3}$$

where $\gamma_t$ is the target cortical tension and $\sigma_{\text{normal}}(\rho_{\text{AB}})$ is the antibody-dependent normal stress at the interface. This scaling corresponds to the minimum length scale of local membrane deformation or bites because below this scale, tension dominates the effects of active stresses and damps out active fluctuations. A similar length scale can be derived by considering the balance between active stresses and

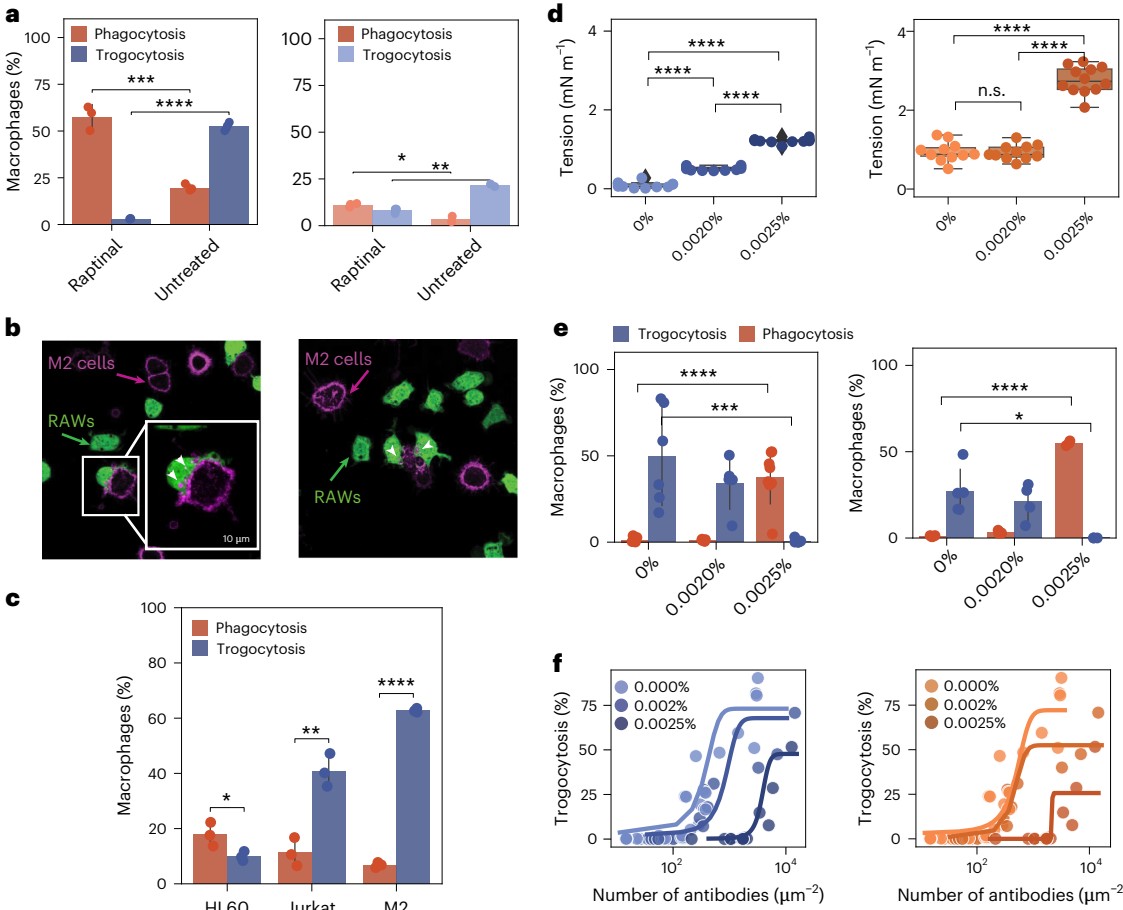

**Fig. 4 | Increasing cell cortical tension suppresses trogocytosis and increases phagocytosis. a**, Cytometric-based analysis of biotinylated Jurkat T cells labelled with AlexaFluor647 streptavidin and opsonized with anti-biotin IgG at 1385 ± 189 molecules per μm² (left) or unopsonized (right) and treated with 10 μM raptinal, or untreated, and co-incubated with LPS-stimulated macrophages for 2 h. Error bars represent the standard deviation over $n = 3$ independent experiments pooling together 10⁵ LPS-stimulated RAWs per experiment co-incubated with anti-biotin opsonized and raptinal-treated Jurkat T cells, anti-biotin opsonized and untreated Jurkat T cells, unopsonized and raptinal-treated Jurkat T cells or unopsonized and untreated Jurkat T cells, respectively. Statistical significance was determined using a two-sided Student's *t*-test. **b**, Two representative confocal micrographs (left and right) of LPS-stimulated RAW 264.7 cells labelled with CellTracker Green CMFDA trogocytosing M2 cells opsonized with AlexaFluor647 anti-CD47 IgG. White arrowheads indicate trogocytic bites. **c**, The percentage of LPS-stimulated macrophages that have phagocytosed (red bars) or trogocytosed (blue bars) anti-CD47 opsonized HL60 cells, Jurkat T cells and M2 cells. The antibody surface density was matched between the three cell types at 163 ± 8 molecules per μm². Error bars are drawn around the mean and represent the standard deviation between $n = 3$ independent experiments pooling together 10⁵ LPS-stimulated RAWs co-incubated with anti-CD47 opsonized HL60 cells, Jurkat T cells, or M2 cells, respectively. Statistical significance was determined using a two-sided Student's *t*-test. **d**, Cell tension measurements for Jurkat cells (left) and HL60 cells (right) treated with 0%, 0.002% and 0.0025% glutaraldehyde. $n = 3$ independent experiments pooling together untreated Jurkat cells (10), 0.002% glutaraldehyde-treated Jurkat cells (10), 0.0025% glutaraldehyde-treated Jurkat cells (10), untreated HL60 cells (11), 0.002% glutaraldehyde-treated HL60 cells (11) and 0.0025% glutaraldehyde-treated HL60 cells (12). Significance was determined using a one-way ANOVA followed by a Tukey's pairwise test. Box plots represent the median and IQR. The whiskers extend to the smallest and largest values within 1.5× IQR. **e**, Cytometric-based analysis of the trogocytic efficiency of macrophages challenged with glutaraldehyde-treated target cells (Jurkat cells, left; HL60 cells, right) and opsonized with anti-CD47 IgG to a surface density between 300 molecules per μm² and 1000 molecules per μm². Error bars are drawn around the mean and represent the standard deviation between $n = 6$ independent experiments pooling together 10⁵ RAWs per experiment co-incubated with untreated Jurkat T cells, $n = 5$ independent experiments pooling together 10⁵ RAWs per experiment co-incubated with 0.002% glutaraldehyde-treated Jurkat T cells, $n = 6$ independent experiments pooling together 10⁵ RAWs per experiment co-incubated with 0.0025% glutaraldehyde-treated Jurkat T cells, $n = 4$ independent experiments pooling together 10⁵ RAWs per experiment co-incubated with untreated HL60 cells, $n = 4$ independent experiments pooling together 10⁵ RAWs per experiment co-incubated with 0.002% glutaraldehyde-treated HL60 cells and $n = 3$ independent experiments pooling together 10⁵ RAWs per experiment co-incubated with 0.0025% glutaraldehyde-treated HL60 cells. Significance was determined using a two-sided Student's *t*-test. **f**, The trogocytic efficiency of macrophages challenged with glutaraldehyde-treated target cells with a surface density titration of anti-CD47 or anti-biotin. Each point represents a single pooled well of 10⁵ RAWs per experiment after co-incubation with Jurkat T cells (left) or HL60 cells (right), opsonized with either anti-CD47 or anti-biotin, from $n = 3$ independent experiments, and treated with either 0% glutaraldehyde, 0.002% glutaraldehyde or 0.0025% glutaraldehyde, respectively. Sigmoid curves (described in text) fit points to guide the eye. Asterisks represent *P* values. n.s., non-significant. \*$P < 0.05$, \*\*$P < 0.01$, \*\*\*$P < 0.001$, \*\*\*\*$P < 0.0001$.

membrane bending (Supplementary Note), which is typically smaller than $R_{min}$ for physiologically relevant values of tension and membrane bending stiffness.

Importantly, this scaling implies that increasing antibody density or decreasing target membrane tension has an equivalent effect on setting the minimum length scale of target membrane deformations. For small antibody density or large tensions, the active stresses are weak compared with those due to tension, and the minimum scale of deformation due to active stress becomes comparable or larger than the interface size $R$ ($R_{min} \gg R$). In this case, the macrophage is incapable

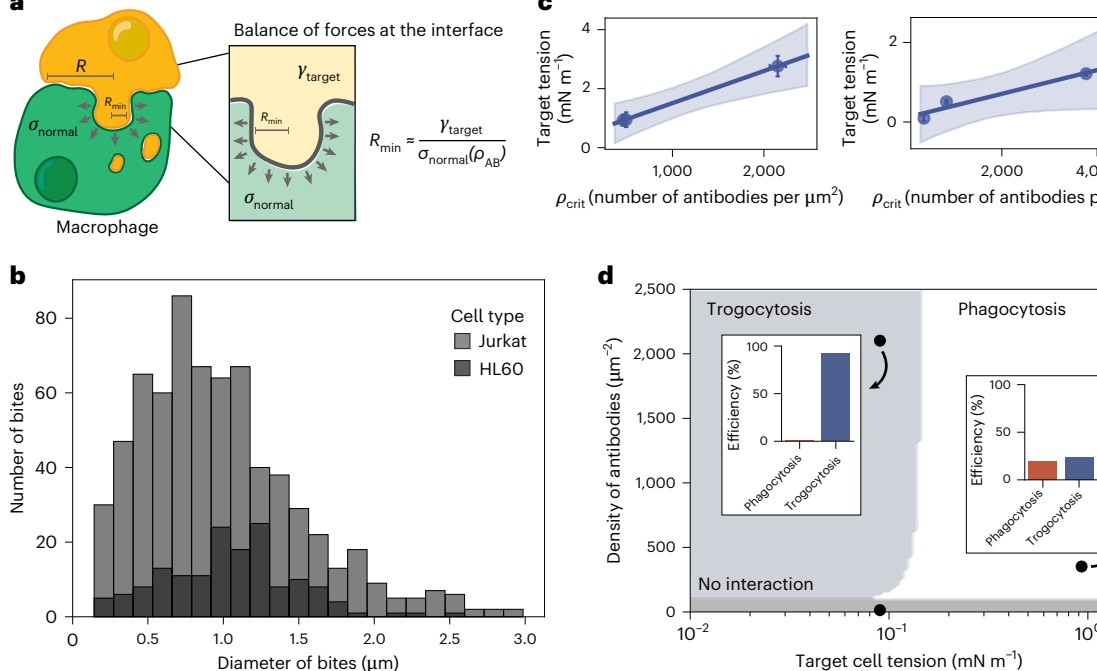

**Fig. 5 | Mechanical scaling relationship describing macrophage trogocytosis.**
**a**, A cartoon model of the macrophage–target interface. **b**, The distribution of
trogocytic 'bite' sizes for macrophages cocultured with Jurkat cells (light grey)
or HL60 cells (dark grey) measured from confocal images of macrophages after
coculture. $n = 2$ independent experiments pooling together measurements from
25 Jurkat T cells and 20 HL60 cells, respectively. **c**, $\rho_{crit}$ scales with the cell tension
for treated HL60 (left) and Jurkat cells (right). The shaded area corresponds to
the 95% confidence interval of the linear fit, which was calculated using bootstrap
resampling. $n = 3$ independent experiments pooling together untreated
Jurkat cells (10), 0.002% glutaraldehyde-treated Jurkat cells (10), 0.0025%
glutaraldehyde-treated Jurkat cells (10), untreated HL60 cells (11), 0.002%

glutaraldehyde-treated HL60 cells (11) and 0.0025% glutaraldehyde-treated
HL60 cells (12). The uncertainty on the points on the $x$ axis represents 95%
confidence intervals for the calculated inflection point of the sigmoid fits and
the standard deviation across cell tension measurements on the $y$ axis. **d**, An
approximate phase diagram of macrophage behaviours at different antibody
densities and effective membrane tensions based on the scaling relationship
for bite size. Points correspond to individual measurements from a single
experiment ($n = 1$) at a particular antibody density and cell tension and their
corresponding phagocytic (right inset, HL60 cells) and trogocytic efficiencies
(left inset, Jurkat cells). Targets in this case are either Jurkat T cells or HL60 cells.

of pinching off bites smaller than the interface and would completely
engulf the target (that is, phagocytosis). By contrast, for small tensions
or large antibody densities, the active stresses can locally overwhelm
membrane tension and pinch off bites smaller than the interface size
($R_{min} \ll R$), leading to trogocytosis. Below a second, lower critical anti-
body density, one expects no engagement, and neither behaviour is
observed. Thus, one can also interpret this scaling as a critical antibody
density or tension that separates the distinct behaviours of phagocy-
tosis and trogocytosis.

If we take our experimentally measured values for $\gamma_t \approx 10^{-2} –
10^{-1}$ mN m$^{-1}$ and the previously measured traction forces that give
normal stresses of order 100 Pa (ref. 35), our scaling predicts a defor-
mation size $R_{min}$ of 0.1–1 μm. Interestingly, this is consistent with the
scale of trogocytic 'bites' measured through confocal microscopy,
which are on the scale of 1 μm for macrophages incubated with Jurkat
T cells and HL60 cells (Fig. 5b). This suggests that the macrophage
can pinch off bites of a size close to the minimum scale of interfacial
deformation. Our scaling analysis assumes smooth deformations
in the membrane, which is controlled by the resistance of the mem-
brane to stretching or bending. For sharp deformations, we expect
the contribution from bending to dominate. However, to see sharp
indentations below a radius of 0.1 μm would require local active
stresses beyond what are expected physiologically (see Supplemen-
tary Note, equation (5)).

Our scaling also predicts that the critical antibody density that
separates phagocytic and trogocytic behaviours in experiments
must be proportional to the target cell cortical tension. To test this,
we fit the sigmoidal curve below to the data in Fig. 4c and extracted

the inflection point of the curve as the critical antibody density in
equation (4), $\rho_{crit}$

$$y = a * \left(1 - \frac{1}{1 + e^{\left(-\frac{x - \rho_{crit}}{d}\right)}}\right). \tag{4}$$

For both HL60 and Jurkat cells, we find that the density of antibody
necessary to initiate trogocytosis indeed increases with cortical ten-
sion (Fig. 5c). We assume that, while this scaling relationship is general,
the difference in slope between HL60 and Jurkat cells is probably due
to important differences in how the active stresses developed at the
interface are related to the local antibody density. In this simple scaling,
we do not consider the possible contributions from dynamic remodel-
ling of the cytoskeleton, spatial heterogeneities due to lipid rafts, or
receptor clustering and cooperative effects, which will probably have
cell-type-specific differences owing to distinct surface proteomes
and properties. For the parameters we measured, antibody density
and cortical tension, our modelling is consistent with the positive
correlation between the critical antibody density and cortical tension
that we observe (Fig. 5c).

From the scaling relationship described above and assuming an
interface size $R$ of ~1 μm, we predict a tension-dependent switch in
macrophage behaviour at a tension around 0.1–1 mN M$^{-1}$ for the range
of normal stresses measured in experiments (50–150 Pa)[35]. On the basis
of this prediction, we can construct an approximate phase diagram that
predicts the macrophage behaviour as a function of antibody density
and cortical tension (Fig. 5d). Our experimental data for Jurkat and

HL60 cell lines are quantitatively consistent with this phase diagram, with cortical tension predicted to have a more significant effect than antibody density.

## Discussion

Immune cells, including macrophages, are in regular physical contact with cells around them, both 'self' and foreign. It is increasingly clear that the physical properties of target cells[35–41] and their surface molecules[42–46] can influence the response of immune cells during phagocytosis, T cell activation, dendritic cell antigen presentation and B cell activation. In this work, we show that the cortical tension of target cells is a key regulator of macrophage decision-making between phagocytosis and trogocytosis.

Using fluorescence microscopy and flow cytometry, we demonstrate that trogocytosis is consistently more common than phagocytosis. Trogocytic efficiency with Jurkat T cells is regularly between 60% and 80% of events, while the highest phagocytic efficiency we measured for HL60 cells is only ~20%. Perhaps this is not surprising given that a target cell can be trogocytosed by many cells but phagocytosed by only one. Still, if trogocytosis is so ubiquitous, why is it rarely quantified in studies of macrophage effector functions? It is possible that trogocytosis is happening in most phagocytosis assays involving soft targets (that is, cells) but is missed if target particles or cells are not labelled with a membrane marker. In this study, we can only 'see' trogocytosis if the membrane of targets is sufficiently labelled. Indeed, in studies where the membrane is labelled, trogocytosis is observed and often characterized as a negative outcome in phagocytosis assays[4,47].

To address the fundamental question of what governs macrophage trogocytosis, we explored the conditions that lead to either phagocytosis or trogocytosis. We observed that trogocytosis and phagocytosis both require FcγR engagement with antibodies on the surface of the target cell and that both require a certain threshold of antibody density (~$10^2$ antibodies per µm²) to occur. Importantly, the identity of the opsonized surface protein does not appear to matter, while the total surface density of antibody influences the degree of trogocytosis for a given cell type.

In our previous experiments using lipid-coated glass beads as targets, macrophages readily phagocytosed but never seemed to trogocytose the membranes on the target beads. This led us to test whether macrophages could phagocytose free-floating and deformable GUVs composed of the same lipids typically found in lipid-coated glass bead assays. We saw that when a target particle is capable of being deformed, as is the case for GUVs, trogocytosis is readily observed and much more frequent than phagocytosis. Our surface tension measurements of GUVs show that membrane tension is a key regulator of trogocytosis by macrophages, with two distinct membrane tension regimes for phagocytosis and trogocytosis. By dynamically switching a GUV from one to the other, we observe that macrophages rapidly transition from trogocytosis to phagocytosis.

The pervasiveness of trogocytosis in antibody-mediated interactions with soft targets such as cells and GUVs is not entirely surprising if we consider the cellular materials required to build a phagocytosis cup. To engulf a large particle, such as a cancer cell, a macrophage must assemble enough actin cytoskeleton and incorporate sufficient excess membrane to fully wrap and exert force upon an object that is roughly the same size as the macrophage. By contrast, trogocytosis also requires notably less cellular machinery than phagocytosis. For example, we measured the distribution of trogocytic 'bite sizes' for HL60 and Jurkat cells, and, on average, each 'bite' is ~1 µm in diameter, which is considerably easier to engulf than cell-sized objects.

However, if, as we hypothesize, the receptors and downstream signalling molecules involved in the phagocytosis and trogocytosis pathways are the same, how might the response differ between the two processes? Both processes require a minimum surface antibody density (~$10^2$ molecules per µm²) to initiate. If the target has low cortical tension, membrane fluctuations increase the probability of binding events. Subsequently, the machinery required for macrophage protrusion and force generation (for example, myosin motors and nucleation-promoting factors that activate Arp 2/3 to produce branched actin networks at the cell–cell interface) is recruited to a smaller area. When the cortical tension is low, the surface is more prone to deformation and scission than a high-tension surface.

With enough FcγRs clustered to locally initiate assembly of a protrusive cup, a soft and deformable target membrane could easily be pinched off at the macrophage–target interface. Indeed, in high-resolution fluorescence microscopy videos of macrophages with HL60 cells, we see multiple 'bites' being extracted from the HL60 cell by the macrophage along the surface of the cell–cell interface (Supplementary Video 4). This observation hints at the intriguing possibility that macrophages are constantly surveying and extracting small bites from their surroundings. Only if they encounter a foreign pathogen, such as a stiff bacterial cell (100–200 MPa for *Pseudomonas aeruginosa*[48]), do they opt for phagocytosis.

Our identification of cortical stiffness as a key factor in the decision between trogocytosis and phagocytosis raises the possibility that it could be beneficial for tumour cells to increase their deformability to reduce the possibility of phagocytic interactions with macrophages. Some tumour cells are known to have reduced stiffness compared with their healthy counterparts[49,50], and immunotherapeutic antibodies targeting tumour cells have been observed to be trogocytosed[12], removing the antibody and leaving the tumour cells alive and less visible to the immune system. Therapies that stiffen the cortex of target cells could have the benefit of promoting phagocytosis and limiting trogocytosis.

## Online content

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

## Methods

### Lipids

Phosphocholine lipids (Avanti Polar Lipids) were used as purchased without further purification. Lipid stock solutions in chloroform contained a quaternary mixture of 97.5 mol% POPC, 1 mol% biotin-PE, 1 mol% PEG2K DSPE and 0.5 mol% lissamine rhodamine PE. GUVs are diluted in an ionic solution of phosphate-buffered saline (PBS), and all lipids in our mixtures are zwitterionic. We added PEG2K DSPE to block GUVs from aggregating in the charge-screened PBS solution.

### Antibodies

Antibodies used to opsonize GUVs and cultured cells were purchased from Santa Cruz Biotechnology and used without further labelling or purification. Biotin and CD47 were bound by, respectively, AlexaFluor647-labelled anti-biotin mouse IgG (clone BK-1/39, Santa Cruz Biotechnologies) and AlexaFluor647-labelled anti-CD47 mouse IgG (B6H12, Santa Cruz Biotechnologies). The FcγRs on RAW cells were bound by anti-CD16/32 mouse IgG (Biolegend).

### RAW 264.7, J774A.1, Jurkat T, Raji B, M2 and HL60 cell culture

RAW 264.7 murine male macrophage-like cell line, J774A.1 murine female macrophage-like cell line, Jurkat T human cell line, Raji B human B cell line, M2 human melanoma cell line and HL60 human promyeloblast cell lines were obtained from the UC Berkeley Cell Culture Facility. RAW 264.7, J774A.1, Jurkat T, M2 and HL60 cells were cultured in Roswell Park Memorial Institute (RPMI) 1640 media (Corning) supplemented with 10% heat-inactivated foetal bovine serum (HI-FBS, Thermo Fisher Scientific) and 1% penicillin–streptomycin (Thermo Fisher Scientific). Raji B cells were cultured in RPMI 1640 media supplemented with 10% HI-FBS, 1% sodium pyruvate (Thermo Fisher Scientific) and 1% penicillin–streptomycin. RAW 264.7 and J774A.1 cells were cultured in non-tissue culture-treated 10-cm dishes (VWR); all other cell lines were cultured in tissue culture-treated 10-cm dishes (Corning) at 37 °C with 5% $CO_2$. Pre-stimulated RAW 264.7 cells were incubated with 100 ng ml$^{-1}$ LPS (Sigma) for 24 h before coculture with target cells.

### Bone marrow-derived macrophages

BMDMs from male C57BL/6 (B6) mice were a kind gift from the Portnoy Lab (UC Berkeley). BMDMs were grown in RPMI 1640 media supplemented with 10% HI-FBS and 1% penicillin–streptomycin at 37 °C with 5% $CO_2$. BMDMs were used in experiments within 24 h of thawing.

### GUV electroformation

Solutions containing 0.25 mg total lipids were spread evenly on slides coated with indium tin oxide (70–100 Ω sq$^{-1}$; Sigma Aldrich). The slides were placed under vacuum for >30 min to permit complete evaporation of chloroform. A capacitor was created by sandwiching a 0.3-mm rubber septum between two lipid-coated slides. The gap was filled with ~200 µl of 285 mM sucrose (osmotically matched to PBS using an osmometer (Precision Systems)). GUVs 10–100 µm in diameter were electroformed[51] by application of an a.c. voltage of 1.5 V at 10 Hz across the capacitor for 1 h at 55 °C.

### Size separation of GUVs

Sucrose solutions containing electroformed GUVs were diluted in 1 ml of 285 mM glucose and injected into a home-built differential density column, lovingly named the SeparatorMax 3000 (Extended Data Fig. 5). Specifically, ten ports with an inner diameter of 2 cm, spaced 2.5 cm apart, were drilled into a cast acrylic tube of 33.6 cm. This tube was connected to two different baths containing a low-density glucose solution (285 mM) and a high-density glucose solution (300 mM) in series. Solutions were loaded into the column via a peristaltic pump with vigorous hand mixing. The resulting solution column is a linear gradient of glucose density from 300 mM glucose (bottom) to 285 mM glucose (top). Vesicles were allowed to

sediment overnight (~12 h) and collected from each of the ten ports of the column. We kept the GUVs from ports 5–8 (labelling from the top to bottom), and they had diameters from 5 to 20 µm. We further purified GUVs by spinning the solution at 300g for 10 min and taking the bottom 100 µl of solution. This solution was then diluted in PBS containing 4 µM AlexaFluor647-labelled anti-biotin mouse IgG (clone BK-1/39, Santa Cruz Biotechnologies). GUVs used in flow cytometer-based phagocytosis and trogocytosis assays were loaded with 1 µM soluble rhodamine dye dissolved in 285 mM sucrose. GUVs were purified from excess dye in solution by running them through the SeparatorMax 3000.

### Imaging techniques

All live cells were maintained at 37 °C and 5% $CO_2$ with a stage top incubator (Okolab) during imaging. For confocal microscopy, cells were imaged with a confocal microscope (Eclipse Ti, Nikon) with a spinning disk (Yokogawa CSU-X, Andor), sCMOS camera (Prime 95B, Photometrics) and 60× objective (Apo TIRF, 1.49NA, oil, Nikon). The spinning disk confocal microscope was controlled with Nikon Elements (Nikon). Images were analysed and prepared using Fiji (http://www.imagej.net/software/fiji).

### Phagocytosis and trogocytosis assays

**Phagocytosis and trogocytosis of labelled cultured cells.** A total of 100,000 macrophages were seeded in wells of a tissue culture flat-bottom 96-well plate (Falcon) in 100 µl RPMI 1640 medium. After seeding, cells were incubated at 37 °C with 5% $CO_2$ for 3–4 h before target addition. To stain the cytoplasm of macrophages, 1 µM CellTracker Green CMFDA was added. Cells were washed 2× with media to remove excess dye. In total, 150,000 target cells (for example, HL60, Jurkat T, M2 or Raji B cells) were diluted in 150 µl 1640 RPMI media containing the appropriate amount of IgG (either anti-CD47 or anti-biotin), 2 µM pHrodo succinimidyl ester (Thermo Fisher Scientific), 10 µM raptinal for apoptosis experiments and the appropriate amount of glutaraldehyde for drug experiments. Cells were labelled for 30 min at 37 °C (for apoptosis experiments, cells were incubated with raptinal for 2 h) and subsequently washed 3× in media. Cells labelled with anti-biotin were first treated with surface biotinylation reagent EZ-link NHS biotin (Thermo Fisher Scientific) for 30 min, followed by washing 3× in 100 mM glycine in PBS pH 8.0 to quench the biotinylation reaction. After washing, 100,000 target cells were added to macrophage-seeded wells and co-incubated for 1 h.

After incubation, wells were scraped with mini cell scrapers (Biotium) and gently pipette-mixed 2–3×. Cells were immediately analysed using an Attune NxT CytPix flow cytometer (Thermo Fisher Scientific). Cells were injected into the flow cytometer at a rate of 200 µl min$^{-1}$, and cells were gated according to the following protocol.

A control sample of CellTracker Green CMFDA-labelled macrophages were run each experiment to determine the 488-nm laser intensity threshold for events positive for CellTracker Green CMFDA. This sample was also used to determine background intensity levels of AlexaFluor647 (647-nm laser) and rhodamine (568-nm laser). Thresholds to establish samples positive for AlexaFluor647 and/or rhodamine were determined by running target cells labelled with AlexaFluor647 anti-CD47 IgG and pHrodo (Extended Data Fig. 8).

Events were determined positive for phagocytosis if they were positive for CellTracker Green CMFDA, pHrodo and AlexaFluor647. Events were determined positive for trogocytosis if they were positive only for CellTracker Green CMFDA and AlexaFluor647 but not for pHrodo (Extended Data Fig. 9). The trogocytic and phagocytic efficiencies were calculated by taking phagocytic and trogocytic events, respectively, and dividing that number by the total number of CellTracker Green CMFDA-positive events.

The surface density of antibody on target cells was measured by comparing cells with calibrated beads with known numbers of

AlexaFluor647 fluorophores (Quantum MESF Kits, Bangs Laboratories). Labelled antibodies from Santa Cruz Biotechnologies have five to seven fluorophores per IgG, as per the manufacturer; therefore, all surface densities calculated in this work considers the average diameter of the cell type (calculated from confocal micrographs) and the average fluorophore per IgG.

**Phagocytosis and trogocytosis of GUVs.** A total of 100 µl GUVs (~1 million GUVs counted with an impedance-based cell counter (Scepter, Sigma Aldrich)) were prepared with 4 µM AlexaFluor647 anti-biotin IgG in PBS and allowed to incubate with gentle rotation for >10 min. After washing, GUVs were added to macrophage-seeded wells as described above. Macrophages were incubated with GUVs for 30 min. After incubation, wells were scraped and immediately analysed on the flow cytometer as described above. Thresholds to establish samples positive for AlexaFluor647 and/or rhodamine were determined by running GUVs labelled with AlexaFluor647 anti-biotin IgG and containing soluble rhodamine.

**Micropipette aspiration to measure target cell cortical tension and GUV membrane tension**

Micropipettes were made from capillaries (1.0 mm outer diameter, 0.58 mm inner diameter, 100 mm length, borosilicate glass; G100-4, Harvard Apparatus) drawn out with a filament pipette puller (Sutter Instruments). Pipette tips were forged with an adapted microforge (MicroData Instruments) to obtain a smooth opening of ~5 µm in diameter. Pipettes were subsequently filled with 0.45 µm filtered PBS using a home-built syringe-pulling device, named The Siphonator 3000, for >30 min. Pipettes were treated for 15 min in a 0.45 µm filtered 10% bovine serum albumin solution before each experiment to passivate the pipette surface and prevent unwanted adhesion of GUVs and target cells to the pipette glass.

The treated pipettes were inserted into a TransferMan 4r micromanipulator (Eppendorf) mounted on a confocal microscope (Eclipse Ti2) to facilitate the manipulation of target cells and/or GUVs for surface tension measurements. To provide controlled suction, the pipettes were connected to a syringe pump (CellTram, Eppendorf) and a pressure sensor (DP103, Validyne Engineering), which measured the suction pressure applied to the pipette.

To measure surface tension, the pipette was aligned to the target interface and increasing suction pressure was applied until a membrane tube with a length equal to the pipette radius was pulled into the pipette. Using Laplace's law, the measured suction pressure ($\Delta P_{\text{suction}}$) was used to measure the interface tension ($\gamma$)

$$\gamma = \frac{\Delta P_{\text{suction}}}{2\left(\frac{1}{R_{\text{pipette}}} - \frac{1}{R_{\text{cell}}}\right)}, \tag{5}$$

where $\Delta P_{\text{suction}}$, $R_{\text{pipette}}$ and $R_{\text{cell}}$ are the suction pressure, the pipette radius and the cell radius, respectively. The interface tensions of GUVs and target cells were modulated by adjusting the suction pressure applied through the same pipettes, thereby varying the interfacial tension.

**Statistics and reproducibility**

All experiments were independently repeated at least three times with similar results. Sample sizes and statistical tests are included in the figure legends wherever appropriate. No statistical methods were used to pre-determine sample sizes. No randomization was used to collect the data. Data distribution was assumed to be normal, but this was not formally tested. Data collection and analysis were not performed in a blinded manner, and no data were excluded from analyses.

**Reporting summary**

Further information on research design is available in the Nature Portfolio Reporting Summary linked to this article.

**Data availability**

Datasets used to generate figures in the paper are available via the Open Science Framework at https://doi.org/10.17605/OSF.IO/6FP4V. All other data supporting the findings of this study are available from the corresponding author on reasonable request. Source data are provided with this paper.

**Code availability**

Original code to generate each of the subfigures (and uploaded as individual scripts) in the paper is available via GitHub at https://github.com/fletchlab-git/Trogocytosis.

## References

51. Dimitrov, D. S. & Angelova, M. I. Lipid swelling and liposome formation mediated by electric fields. *J. Electroanal. Chem.* **253**, 323–336 (1988).

## Acknowledgements

This work is supported by a National Science Foundation (NSF) Center for Cellular Construction grant DBI-1548297 (to D.A.F.), National Institutes of Health (NIH) grant R01GM134137 (to D.A.F.) and Chan Zuckerberg Biohub Investigator (to D.A.F.). It was also supported by the James S. McDonnell Foundation Postdoctoral Fellowship (to C.E.C.), the European Molecular Biology Organization (EMBO) Postdoctoral Fellowship (to A.C.), the Schmidt Science Fellowship in partnership with the Rhodes Trust (to D.K.), the Cancer Research Institute Postdoctoral Fellowship (to N.R.M.) and the Stichting Fonds Doctor Catharine van Tussenbroek Travel Grant (to L.B.). We thank the Physiology course at the Marine Biological Laboratory in Woods Hole, MA, for providing the resources and environment to generate the initial ideas and experiments for this work.

## Author contributions

C.E.C. and D.A.F. designed the project. C.E.C., A.C. and D.K. analysed the data. C.E.C., A.C., N.R.M. and L.B. performed micropipette aspiration on cells and GUVs. C.E.C., N.R.M. and L.B. performed flow cytometry measurements and analysis. C.E.C. performed image analysis. D.K. and C.E.C. performed theoretical analysis and wrote the scaling relationship. C.E.C., A.C., D.K. and D.A.F. wrote the manuscript. All authors contributed to editing the manuscript and figures.

## Competing interests

The authors declare no competing interests.

## Additional information

**Extended data** is available for this paper at https://doi.org/10.1038/s41556-025-01807-6.

**Correspondence and requests for materials** should be addressed to Daniel A. Fletcher.

**Extended Data Fig. 1 | Trogocytosis is observed much more frequently than phagocytosis in macrophages.** A series of confocal fluorescence images from the same experiment of RAW 264.7 macrophages (green; CellTracker Green CMFDA and LifeAct GFP) incubated with anti-CD47 opsonized Jurkat T cells (magenta and yellow; AlexaFluor 647 anti-CD47 (membrane) and pHrodo (endosomes)). Stars indicate phagocytosis events in which both pHrodo and AF647 signals from Jurkats are present inside a phagosome of a RAW cell. Arrows indicate macrophages that have internalized small trogocytic bites positive only for AF647 from Jurkat T cells. The experiment was repeated 4 independent times with similar results.

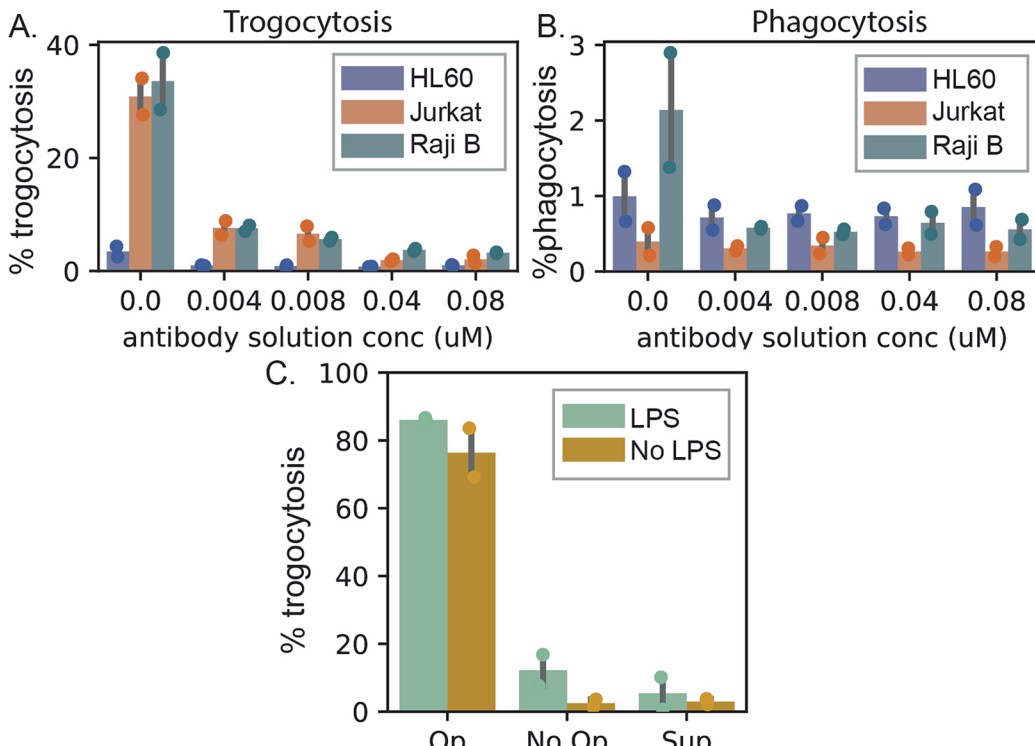

**Extended Data Fig. 2 | Titration of anti-CD16/32 (FcBlock) on macrophages before incubation with anti-CD47 opsonized target cells reduces both trogocytosis and phagocytosis.** (**A**) Trogocytosis in macrophages is reduced with increasing FcγR blocking antibody. $N = 2$ independent experiments pooling together $10^5$ RAWs that had co-incubated with $10^5$ target cells (either HL60s, Jurkats, or Raji Bs) for 0, 0.004, 0.008, 0.04, and 0.08 uM FcγR blocking antibody concentrations, respectively. (**B**) Phagocytosis in macrophages is reduced with increasing FcγR blocking antibody. $N = 2$ independent experiments pooling together $10^5$ RAWs that had co-incubated with $10^5$ target cells (either HL60s, Jurkats, or Raji Bs) for 0, 0.004, 0.008, 0.04, and 0.08 uM FcγR blocking antibody concentrations, respectively. (**C**) Trogocytosis of AlexaFluor647

anti-biotin IgG opsonized Jurkat T cells by LPS-stimulated macrophages or unstimulated macrophages. When Jurkat T cells are uncoated but labeled with AlexaFluor647 streptavidin, trogocytosis is decreased. Similarly, when the supernatant from opsonized Jurkats is added to macrophages, there is minimal uptake of AlexaFluor647 streptavidin-labeled membrane. $N = 2$ independent experiments pooling together $10^5$ LPS-stimulated RAWs or unstimulated RAWs that had co-incubated with either $10^5$ anti-CD47 opsonized Jurkat T cells, $10^5$ non-opsonized Jurkat T cells, or the supernatant from $10^5$ anti-CD47 opsonized Jurkat T cells, respectively. Error bars are drawn around the mean and represent the standard deviation between independent experiments ($N = 2$).

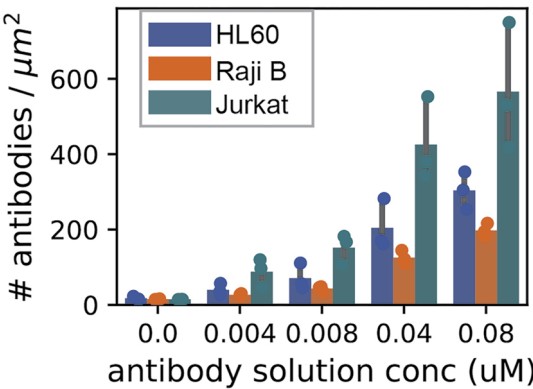

**Extended Data Fig. 3 | CD47 expression on the surface of Jurkat T cells is ~2.2x higher than HL60s and Raji B cells at a concentration of 0.04 µM.** Measure of the surface density of CD47 on HL60 cells, Jurkat T cells, and Raji B cells. $N = 3$ independent experiments pooling together $10^5$ HL60s, Jurkats, or Raji B and opsonized with 0, 0.004, 0.008, 0.04, and 0.08 uM AlexaFluor647 anti-CD47 antibody, respectively. Error bars are drawn around the mean and represent the standard deviation between independent experiments ($N = 3$).

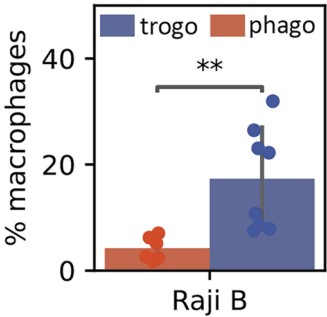

**Extended Data Fig. 4 | Macrophages trogocytose Raji B cells opsonized with anti-CD19.** Raji B cells were opsonized with AlexaFluor647 anti-CD19 at a surface density of 212 molecules/μm². Significance was determined via student's T test (p = 2.285×10⁻³) and error bars represent standard deviation between independent experiments (N = 8), pooling together 10⁵ RAWs that had co-incubated with 10⁵ Raji B cells opsonized with anti-CD19.

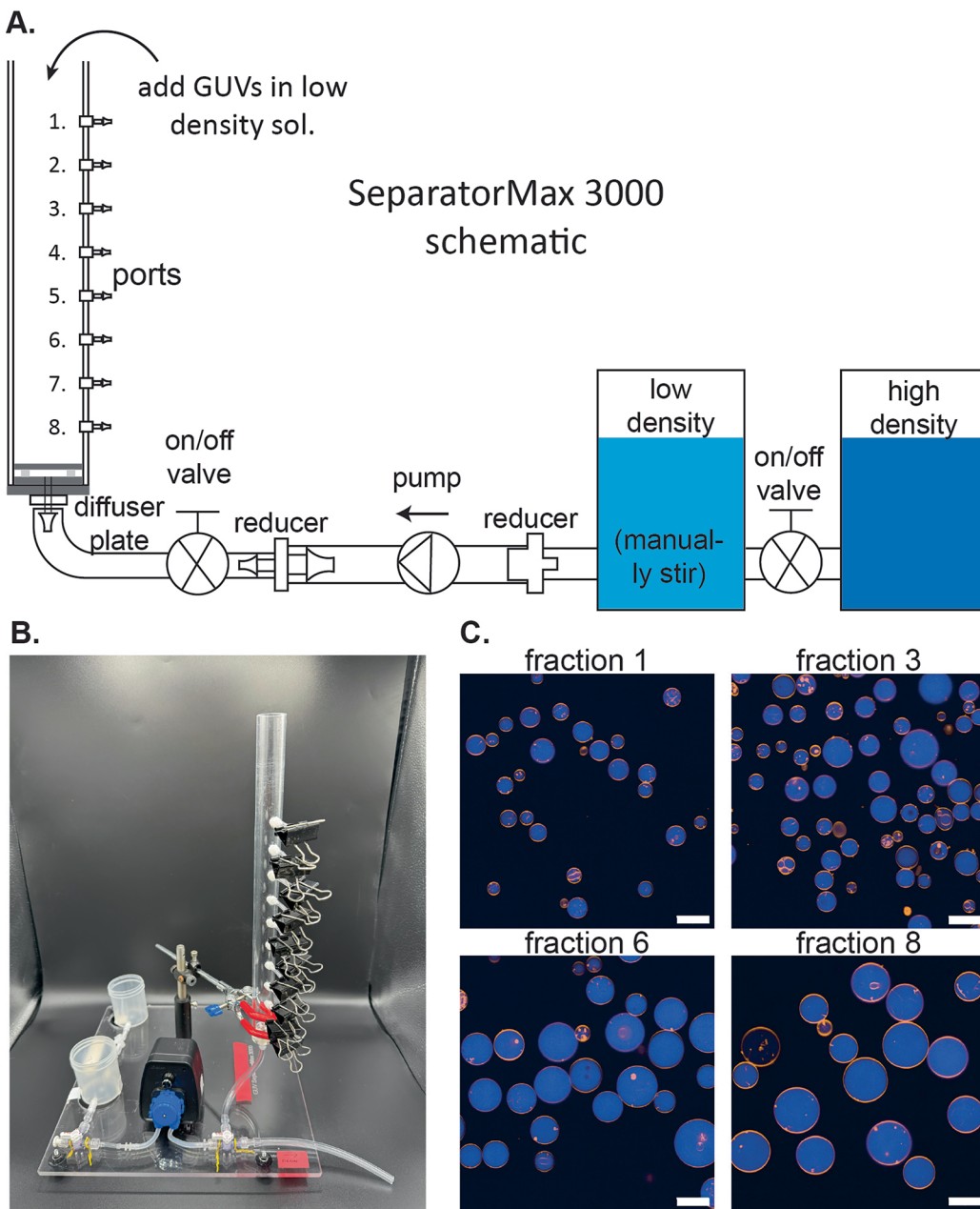

**Extended Data Fig. 5 | of the SeparatorMax 3000, a tool for separating GUVs based on size.** Design (**A**) Schematic of the SeparatorMax 3000. A primary feature of the design is two different reservoirs of different density solutions. Solutions are pumped into the column while mixing to create a uniform density gradient that opposes convection. GUVs are loaded from the top of the column and allowed to settle for 16-24hrs. GUVs are harvested from different sections of the columns through the outlet ports. (**B**) Photo of the actual SeparatorMax 3000. The ports for retrieving specific fractions are sealed with binder clips until needed. (**C**) Confocal fluorescence images of GUVs harvested from different fractions (fraction diameters: one = 14.6 ± 1.6 μm; three = 14.2 ± 5.3 μm; six = 25.9 ± 6.9 μm; eight = 31.1 ± 5.3 μm). Average diameters were calculated from 20 GUVs per fraction. GUVs are composed of 99.5 mol% POPC and 0.5 mol% liss rho PE and they are filled with soluble FITC. Scale bars are 25 μm.

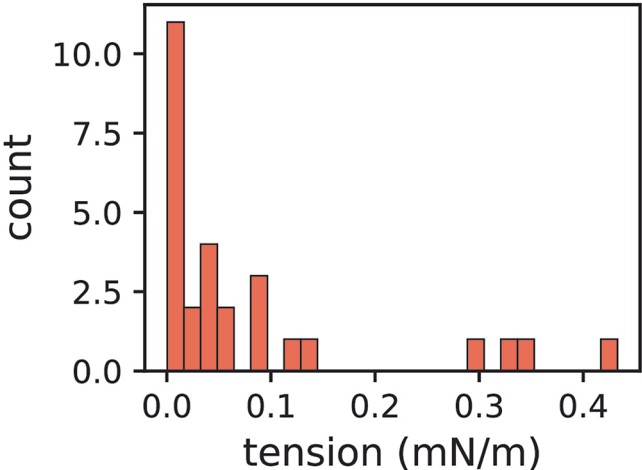

**Extended Data Fig. 6 | Membrane tensions of GUVs in an 'isotonic' solution still vary over an order of magnitude (*n* = 27).** Histogram of individual membrane tensions measured for *N* = 27 GUVs in an isotonic solution (that is the inner sucrose solution is osmotically matched to the PBS buffer).

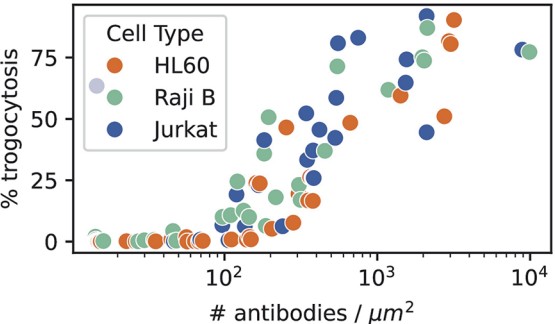 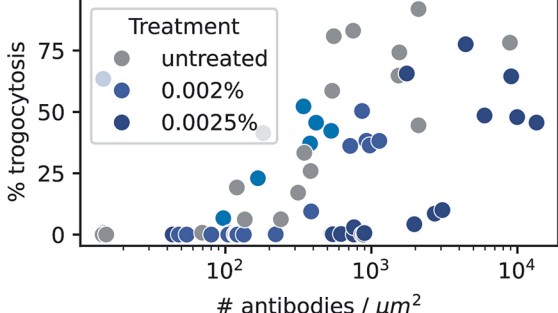

**Extended Data Fig. 7 | Trogocytosis is partially recovered for glutaraldehyde-treated Jurkat cells if the antibody surface density is increased.** The amount of trogocytosis in macrophages increases with surface antibody density for all target cells (left panel). Each point represents a single pooled well of $10^5$ RAWs after co-incubation with Jurkat T cells, opsonized with either anti-CD47 or anti-Biotin, from $N$ = 3 independent experiments.

While trogocytosis increases with surface antibody density on glutaraldehyde-treated Jurkats, the amount of trogocytosis is always lower than in wild-type cells (right panel). Each point represents a single pooled well of $10^5$ RAWs after co-incubation with Jurkat T cells, opsonized with either anti-CD47 or anti-Biotin, from N = 3 independent experiments, and treated with either 0% glutaraldehyde, 0.002% glutaraldehye, or 0.0025% glutaraldehyde, respectively.

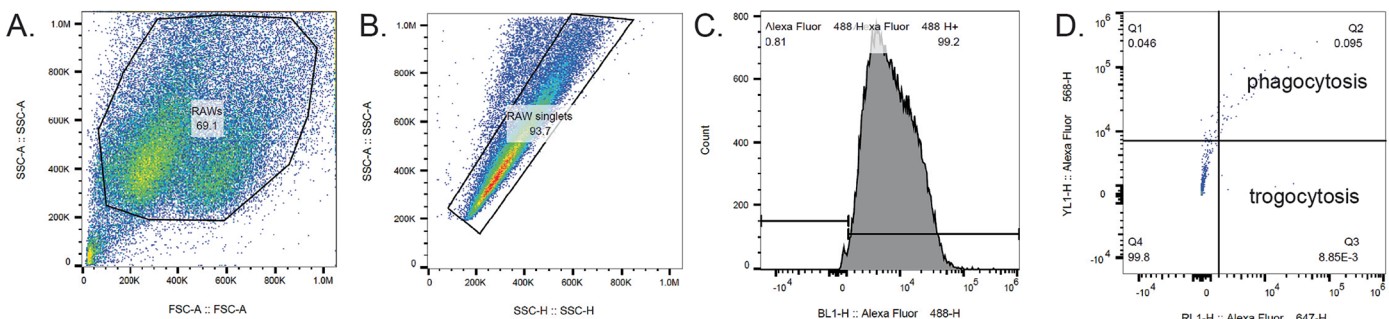

**Extended Data Fig. 8 | Schematic of flow cytometry gating pipeline. A.** A gate is drawn in the SSC-A vs FSC-A dot plot to identify cells. **B.** Singlets are gated out by drawing a diagonal gate in the SSC-A vs SSC-H dot plot. **C.** Macrophages (labeled with CellTracker Green CMFDA) are separated from target cells by drawing a gate around AlexaFluor 488 positive events. **D.** Phagocytosis and trogocytosis gates are drawn as quadrants 2 (phagocytosis: pHrodo (+) and AlexaFluor 647 (+)) and 3 (trogocytosis: pHrodo (-) and AlexaFluor 647 (+)). The example gates here are drawn for a control sample that contains only CellTracker Green CMFDA labeled RAW 264.7 macrophages.

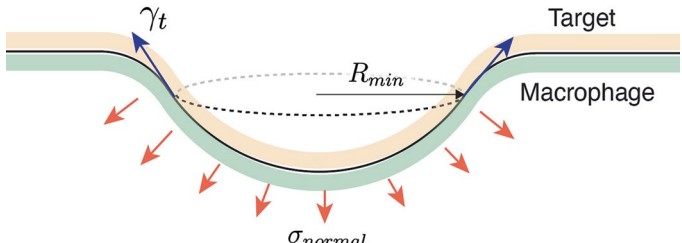

**Extended Data Fig. 9 | Balance of stresses at the phagocytic synapse between a macrophage and target cell.** Schematic of the balance of stresses at the interface between a macrophage and a target cell.

# Reporting Summary

## Statistics

For all statistical analyses, confirm that the following items are present in the figure legend, table legend, main text, or Methods section.

| n/a | Confirmed | |
|---|---|---|
| ☐ | ☒ | The exact sample size (*n*) for each experimental group/condition, given as a discrete number and unit of measurement |
| ☐ | ☒ | A statement on whether measurements were taken from distinct samples or whether the same sample was measured repeatedly |
| ☐ | ☒ | The statistical test(s) used AND whether they are one- or two-sided *Only common tests should be described solely by name; describe more complex techniques in the Methods section.* |
| ☒ | ☐ | A description of all covariates tested |
| ☐ | ☒ | A description of any assumptions or corrections, such as tests of normality and adjustment for multiple comparisons |
| ☐ | ☒ | A full description of the statistical parameters including central tendency (e.g. means) or other basic estimates (e.g. regression coefficient) AND variation (e.g. standard deviation) or associated estimates of uncertainty (e.g. confidence intervals) |
| ☒ | ☐ | For null hypothesis testing, the test statistic (e.g. *F*, *t*, *r*) with confidence intervals, effect sizes, degrees of freedom and *P* value noted *Give P values as exact values whenever suitable.* |
| ☒ | ☐ | For Bayesian analysis, information on the choice of priors and Markov chain Monte Carlo settings |
| ☒ | ☐ | For hierarchical and complex designs, identification of the appropriate level for tests and full reporting of outcomes |
| ☒ | ☐ | Estimates of effect sizes (e.g. Cohen's *d*, Pearson's *r*), indicating how they were calculated |

*Our web collection on statistics for biologists contains articles on many of the points above.*

## Software and code

Policy information about availability of computer code

| Data collection | To collect flow cytometry data for this manuscript, Attune Cytometric Software was used . To collect image data for this manuscript, Nikon Elements was used. |
|---|---|
| Data analysis | All data analysis was done using FlowJo (for flow cytometry data), ImageJ or Nikon Elements (image data), and python (to plot collected data). All code to plot figures in the manuscript is available at Github (https://github.com/fletchlab-git/Trogocytosis) and Code Ocean. |

For manuscripts utilizing custom algorithms or software that are central to the research but not yet described in published literature, software must be made available to editors and reviewers. We strongly encourage code deposition in a community repository (e.g. GitHub). See the Nature Portfolio guidelines for submitting code & software for further information.

## Data

Policy information about availability of data

All manuscripts must include a data availability statement. This statement should provide the following information, where applicable:
- Accession codes, unique identifiers, or web links for publicly available datasets
- A description of any restrictions on data availability
- For clinical datasets or third party data, please ensure that the statement adheres to our policy

All raw data and processed data needed to plot figures is available in an Open Science Framework repository DOI 10.17605/OSF.IO/6FP4V. All processed data

needed to plot figures from the paper is available in a Code Ocean repository. It is also available as a Statistical Source Data file associated with the manuscript. The data required to plot each subfigure is provided in separate sheets of the .xlsx file.

## Research involving human participants, their data, or biological material

Policy information about studies with human participants or human data. See also policy information about sex, gender (identity/presentation), and sexual orientation and race, ethnicity and racism.

| | |
|---|---|
| Reporting on sex and gender | *Use the terms sex (biological attribute) and gender (shaped by social and cultural circumstances) carefully in order to avoid confusing both terms. Indicate if findings apply to only one sex or gender; describe whether sex and gender were considered in study design; whether sex and/or gender was determined based on self-reporting or assigned and methods used.*<br>*Provide in the source data disaggregated sex and gender data, where this information has been collected, and if consent has been obtained for sharing of individual-level data; provide overall numbers in this Reporting Summary. Please state if this information has not been collected.*<br>*Report sex- and gender-based analyses where performed, justify reasons for lack of sex- and gender-based analysis.* |
| Reporting on race, ethnicity, or other socially relevant groupings | *Please specify the socially constructed or socially relevant categorization variable(s) used in your manuscript and explain why they were used. Please note that such variables should not be used as proxies for other socially constructed/relevant variables (for example, race or ethnicity should not be used as a proxy for socioeconomic status).*<br>*Provide clear definitions of the relevant terms used, how they were provided (by the participants/respondents, the researchers, or third parties), and the method(s) used to classify people into the different categories (e.g. self-report, census or administrative data, social media data, etc.)*<br>*Please provide details about how you controlled for confounding variables in your analyses.* |
| Population characteristics | *Describe the covariate-relevant population characteristics of the human research participants (e.g. age, genotypic information, past and current diagnosis and treatment categories). If you filled out the behavioural & social sciences study design questions and have nothing to add here, write "See above."* |
| Recruitment | *Describe how participants were recruited. Outline any potential self-selection bias or other biases that may be present and how these are likely to impact results.* |
| Ethics oversight | *Identify the organization(s) that approved the study protocol.* |

Note that full information on the approval of the study protocol must also be provided in the manuscript.

# Field-specific reporting

Please select the one below that is the best fit for your research. If you are not sure, read the appropriate sections before making your selection.

☒ Life sciences　　　　☐ Behavioural & social sciences　　　　☐ Ecological, evolutionary & environmental sciences

For a reference copy of the document with all sections, see nature.com/documents/nr-reporting-summary-flat.pdf

# Life sciences study design

All studies must disclose on these points even when the disclosure is negative.

| | |
|---|---|
| Sample size | No sample-size calculation was performed. For measurements of populations of macrophages, approx. 100,000 cells per experimental condition per biological replicate were analyzed. For each experiment, three biological replicates were performed on three different days. |
| Data exclusions | No data was excluded from the manuscript. |
| Replication | All experiments were repeated at least three times on three different days to ensure reproducibility. Variation is reflected in the error bars present on figures in the manuscript and described in figure captions. All attempts at replication were successful. |
| Randomization | This is not relevant to our study as we were investigating cell behavior in cultured immortalized cell lines. Experiments done with bone marrow derived macrophages were performed on cells derived from one mouse. |
| Blinding | Investigators were not blinded during data acquisition and analysis. As the analytical methods employed require little-to-no subjective input from the investigators, this did not seem necessary. |

# Reporting for specific materials, systems and methods

We require information from authors about some types of materials, experimental systems and methods used in many studies. Here, indicate whether each material, system or method listed is relevant to your study. If you are not sure if a list item applies to your research, read the appropriate section before selecting a response.

## Materials & experimental systems

| n/a | Involved in the study |
|---|---|
| ☐ | ☒ Antibodies |
| ☐ | ☒ Eukaryotic cell lines |
| ☒ | ☐ Palaeontology and archaeology |
| ☒ | ☐ Animals and other organisms |
| ☒ | ☐ Clinical data |
| ☒ | ☐ Dual use research of concern |
| ☒ | ☐ Plants |

## Methods

| n/a | Involved in the study |
|---|---|
| ☒ | ☐ ChIP-seq |
| ☐ | ☒ Flow cytometry |
| ☒ | ☐ MRI-based neuroimaging |

## Antibodies

| | |
|---|---|
| Antibodies used | AlexaFluor647-labeled anti-biotin mouse IgG (clone BK-1/39, Santa Cruz Biotechnologies); AlexaFluor647-labeled anti-CD47 mouse IgG (B6H12, Santa Cruz Biotechnologies); anti-CD16/32 mouse IgG (Biolegend) |
| Validation | Anti-biotin and anti-CD47 were both validated for use in Western Blots or Immunofluorescence. The manufacturer's website provides a list of citations for each antibody as well as details about the validation. Anti-CD16/32 was validated for use as an Fc-receptor blocking antibody through immunofluorescent staining and flow cytometry. A list of citations and information on validation is available on the manufacturer's website. We used all three antibodies in this study without further purification or labeling. |

## Eukaryotic cell lines

Policy information about cell lines and Sex and Gender in Research

| | |
|---|---|
| Cell line source(s) | RAW 264.7 cells: Mouse origin, male; Jurkat T cells: Human origin, male; HL60 cells: Human origin, female; Raji B cells: Human origin, male; J774A.1 cells: Mouse origin, female; BMDM cells: Mouse origin, JAX B6 female. M2 cells: Human origin, sex unspecified. All cells except bone marrow derived macrophages were acquired from the Barker Cell Culture Facility at UC Berkeley. |
| Authentication | All cell lines from the Barker Cell Culture Facility at UC Berkeley are authenticated by the facility. |
| Mycoplasma contamination | All cell lines tested negative for mycoplasma contamination. |
| Commonly misidentified lines (See ICLAC register) | No cell lines commonly misidentified per ICLAC register were used |

## Plants

| | |
|---|---|
| Seed stocks | *Report on the source of all seed stocks or other plant material used. If applicable, state the seed stock centre and catalogue number. If plant specimens were collected from the field, describe the collection location, date and sampling procedures.* |
| Novel plant genotypes | *Describe the methods by which all novel plant genotypes were produced. This includes those generated by transgenic approaches, gene editing, chemical/radiation-based mutagenesis and hybridization. For transgenic lines, describe the transformation method, the number of independent lines analyzed and the generation upon which experiments were performed. For gene-edited lines, describe the editor used, the endogenous sequence targeted for editing, the targeting guide RNA sequence (if applicable) and how the editor was applied.* |
| Authentication | *Describe any authentication procedures for each seed stock used or novel genotype generated. Describe any experiments used to assess the effect of a mutation and, where applicable, how potential secondary effects (e.g. second site T-DNA insertions, mosiacism, off-target gene editing) were examined.* |

## Flow Cytometry

### Plots

Confirm that:

☒ The axis labels state the marker and fluorochrome used (e.g. CD4-FITC).

☒ The axis scales are clearly visible. Include numbers along axes only for bottom left plot of group (a 'group' is an analysis of identical markers).

☒ All plots are contour plots with outliers or pseudocolor plots.

☒ A numerical value for number of cells or percentage (with statistics) is provided.

## Methodology

**Sample preparation**

RAW 264.7, J774A.1, or bone marrow-derived macrophages were co-cultured in 96-well TC-treated plates with either HL60s, Jurkat T, Raji B cells, M2 cells, or GUVs. After co-incubation, wells were scraped with miniature cell scrapers, pipette mixed, and immediately analyzed on the flow cytometer.

**Instrument**

The instrument used was an Attune CytPix Flow Cytometer.

**Software**

Flow cytometry data was collected using the Attune Cytometric Software. Exported FCS files were then analyzed in FlowJo_v10.8.1.

**Cell population abundance**

Cells were only analyzed on the flow cytometer, they were not sorted and further experimented upon.

**Gating strategy**

To determine cells that had trogocytosed or phagocytosed, cells were first gated by FSC and SSC to eliminate small cell debris. Single cells were then selected by drawing a diagonal gate around a SSA vs SSH plot. To isolate macrophages from their target cells, this population was then gated for cells positive for CellTracker Green CMFDA. The populations of cells containing AlexaFluor 647 and/or pHrodo red were then analyzed. Extended Data Figure 8 describes the gating strategy.

☒ Tick this box to confirm that a figure exemplifying the gating strategy is provided in the Supplementary Information.

