## [Peer Review File · Nature Cell Biology]

Target cell cortical tension regulates macrophage trogocytosis

Corresponding Author: Dr Daniel Fletcher

Version 0:

Decision Letter:

Dear Dr Fletcher,

I apologize for the delay. Your manuscript "Target cell tension regulates macrophage trogocytosis", has now been seen by 3 referees, who are experts in mechanics and T cells (referee 1); T cells and endocytosis and phagocytosis (referee 2); and in vitro membrane reconstitutions and internalization (referee 3), and whose comments are pasted below. In light of their advice, we regret that we cannot offer to publish the study in Nature Cell Biology.

As you will see, although the reviewers find this work interesting, they raise serious concerns that question the conceptual advance that these findings represent over previous work, and the strength of the data and of the novel conclusions that can be drawn at this stage. These concerns include unclear mechanisms underlying use of trogocytosis vs. phagocytosis, unclear physiological relevance, as well as concerns about methods to perturb and measure membrane tension (from all Reviewers).

Although we cannot publish your paper, it may be appropriate for another journal in the Nature Portfolio. If you wish to explore the journals and transfer your manuscript please use our manuscript transfer portal. You will not have to re-supply manuscript metadata and files, unless you wish to make modifications. For more information, please see our [manuscript transfer FAQ](http://www.nature.com/authors/author_resources/transfer_manuscripts.html?WT.mc_id=EMI_NPG_1511_AUTHORTRANSF&WT.ec_id=AUTHOR) page.

If you would be interested in the option to transfer the manuscript to the EMBO Journals, Nature Communications, or Communications Biology, please let me know if you would like me to initiate a consultation with my colleagues there to explore whether they would commit to take the manuscript forward with the existing peer review history.

We are very sorry that we could not be more positive on this occasion, but we thank you for the opportunity to consider this work.

With kind regards,
Daryl

Daryl Jason Verzosa David, PhD

Senior Editor, Nature Cell Biology
Advisory Editor, npj Biological Physics and Mechanics
Nature Portfolio

Heidelberger Platz 3, 14197 Berlin, Germany
Email: daryl.david@nature.com
ORCID: <https://orcid.org/0000-0002-9253-4805>

Reviewers' comments:

Reviewer #1 (Remarks to the Author):

This manuscript explores how target cell cortical tension regulates macrophage trogocytosis versus phagocytosis. The

authors propose that the target cell, rather than the macrophage, determines the choice between trogocytosis and phagocytosis and that macrophages do not require a distinct molecular pathway for trogocytosis. This work is potentially important because it shifts the focus from traditional molecular signaling mechanisms to the biophysical properties of target cells, offering new insights into how mechanical cues influence immune cell behavior. Understanding how target cell mechanics dictate macrophage responses could have broad implications for fields such as cancer immunotherapy, infection control, and the clearance of apoptotic cells, where changes in cell stiffness and membrane tension often occur. However, I believe that the current experimental data are still too preliminary to fully support the authors' conclusions. While the study likely establishes a correlation between target cell tension and the macrophage's response, the evidence provided is insufficient to establish a direct causal relationship. If the authors can provide more convincing data to address the concerns regarding causality, I would be happy to support the publication of this work. Below are my detailed comments and experimental suggestions that I believe will significantly strengthen the manuscript's conclusions :

Comment 1: Many of my concerns arise from the dynamic micropipette aspiration experiment, primarily due to the low throughput of this method. This issue is particularly significant in Figure 3, as the data presented there form the core of the manuscript's conclusions. The authors' approach of using GUVs (Giant Unilamellar Vesicles) as substitutes for target cells is conceptually clever because it effectively eliminates potential confounding factors arising from target cell membrane composition and cytoskeletal interactions, allowing for a more direct examination of the role of membrane tension. However, the current data are not sufficiently convincing. The authors only show a few representative cases to support the claim that increasing target membrane tension can induce a shift in macrophage behavior from trogocytosis to phagocytosis. The sample size in these experiments appears limited, and the observed timing of phagocytosis onset could be coincidental rather than indicative of a true causal relationship.

To address this concern, I suggest that the authors prepare a series of GUVs with different membrane tensions by varying lipid compositions (e.g., adjusting the POPC/DOPE ratio). By using the membrane tension probe (Flipper-TR® fluorescent cell membrane tension probe), they can validate and quantify the differences in membrane tension among these GUV preparations. Subsequently, real-time fluorescence imaging can be used to monitor macrophage interactions with these tension-defined GUVs. This approach would not only enable the collection of a larger dataset in a single experimental session, thereby improving reliability and reproducibility, but also provide direct visual evidence supporting the causal relationship between target membrane tension and the macrophage's choice between trogocytosis and phagocytosis.

2. Similarly, the use of the Flipper-TR® membrane tension probe in the experiments presented in Figure 2 and Figure 4 would greatly enhance the robustness and persuasiveness of the data. Incorporating this probe into the experimental design would provide real-time, quantitative, and visual evidence of membrane tension differences, allowing for a more direct validation of the authors' central hypothesis that target cell tension regulates the choice between trogocytosis and phagocytosis. This addition is particularly important because, in both experiments, the current assessment of target cell tension relies on indirect methods or static measurements, which may not accurately reflect the actual membrane tension experienced by macrophages during the interaction. Without direct and dynamic tension measurements, it remains unclear whether the observed changes in macrophage behavior are truly driven by differences in membrane tension or potentially confounded by other factors, such as alterations in membrane composition, cell surface protein distribution, or fixation-induced artifacts. By applying membrane tension probe to the target cells used in Figure 2, the authors could verify whether differences in trogocytosis and phagocytosis frequencies correlate directly with measurable variations in membrane tension across different cell types. Similarly, incorporating the probe into the Figure 4 experiments—where target cell tension is increased through glutaraldehyde fixation—would confirm whether the observed suppression of trogocytosis and potential enhancement of phagocytosis correspond to genuine tension increases, rather than nonspecific effects of the fixation process.

3. While the manuscript focuses on how target cell membrane tension and antibody density influence the choice between trogocytosis and phagocytosis, it lacks a detailed discussion of the intracellular signaling mechanisms by which macrophages make this decision. Both processes require Fcγ receptor activation, but it remains unclear how differences in signal strength, duration, or downstream pathway activation lead to the selection of one response over the other.

4. The study focuses exclusively on antibody-mediated trogocytosis and phagocytosis. Have the authors considered whether the tension-regulation mechanism applies more broadly beyond antibody-dependent processes? For example, would similar trends be observed with other opsonins (e.g., complement components) or natural ligands such as phosphatidylserine on apoptotic cells? Investigating this could help clarify whether the proposed mechanism is specific to Fcγ receptor engagement or part of a more general mechanobiological response across different forms of target recognition.

Reviewer #2 (Remarks to the Author):

In this paper, the authors convincingly demonstrate that the process of trogocytosis, the “nibbling” of the plasma membrane by phagocytes (such as macrophages) is strongly influenced by the cell surface tension of the target cell membrane. Sophisticated methods are employed to reach this conclusion, with convincing results. The authors conclude that this the major factor in the “decision” to either trogocytosis or phagocytose a target cell. I have a number of concerns regarding this conclusion and the physiological relevance of their findings.

1. One major concern with their conclusion is that although the role of target cell surface tension in the efficiency of trogocytosis is clearly demonstrated, no evidence is provided that cells with high surface tension are preferentially phagocytosed. In their system, phagocytosis is a rare event, and increasing surface tension with low concentrations of glutaraldehyde did not appear to increase this (certainly not anywhere near the levels of trogocytosis they observe in

untreated cells). While their conclusions regarding trogocytosis seem solid to this reviewer, conclusions regarding phagocytosis are not supported by data.

2. While trogocytosis has been described as having a role in immune evasion in pathogenic amoeba and in nematodes, it is unclear if trogocytosis plays essential roles in mammalian physiology. No attempt to identify a physiological role for the mechanism they describe is offered. While treatment with glutaraldehyde is useful for their analysis, what physiological conditions influence surface tension sufficiently to affect trogocytosis? Further, when trogocytosis is inhibited (but phagocytosis is presumably intact) what are the consequences? For example, we can speculate that trogocytosis of anti-CD47-treated cells might limit phagocytic killing of these cells; does surface tension account for differences in the ability of anti-CD47 (or SIRPa) treatment to result in killing of some cell lines but not other? (Note, I am not insisting on this experiment but offer it as an example). There are other claims as to the function of trogocytosis in mammalian cells. In the absence of a physiological consequence of trogocytosis, we are left with only a mechanism for a mildly interesting phenomenon without context.

3. Does trogocytosis of living cells depend on inhibition of CD47-SIRPa interaction? The biotin experiment doesn't answer this question, since CD47 is biotinylated in this situation. Are other antibodies to cell surface proteins engulfed in the absence of blocking CD47? If so, does surface tension play a role in such situations? This is important, since it is assumed that surface tension controls the ability of phagocytes to "nibble" small pieces of membrane, but if the effects depend on blocking CD47-SIRPa, then it remains possible that surface tension influences this blockade.

4. The results in Figure S2 are clear but unconvincing. Since trogocytosis is measured here with uptake of labeled antibody, the ability of anti-FcR to "block" trogocytosis is not surprising. Does inhibition of FcR prevent uptake of biotinylated proteins in this setting?

5. Figure 2. The conclusion that HL60 cells "are phagocytosed more than the other lines" is not clear—are these small differences significant (and sufficiently powered)?

6. Figure 4. As mentioned in #1. Increasing cell surface tension with glutaraldehyde clearly inhibits trogocytosis, however the prediction would be that it increases phagocytosis. Is this true? Only trogocytosis was measured. If phagocytosis does not appreciably increase as trogocytosis decreases (and I have to assume that if this were the case, it would have been shown), then the study does not describe the difference between these two processes, but rather a requirement for efficient trogocytosis itself.

7. (minor) Cells without glutaraldehyde treatment are repeatedly referred to as "wild-type." "Untreated" would be more accurate.

Reviewer #2 (Remarks on code availability):

N/A

Reviewer #3 (Remarks to the Author):

In their manuscript, Cornell et al., try to understand why a different macrophage lines including primary macrophages would be directed to trigger trogocytosis versus phagocytosis of opsinized target cells. The propensity of different target cells to undergo predominantly trogocytosis or phagocytosis presented some clues for the authors, since they had observed that the predominantly trogocytosed, Jurkat and Raji cells had a lower cortical tension than the more phagocytic-prone HL60 cells. Furthermore at similar surface antibody levels, Jurkat and Raji cells appear to show similar tendencies to trigger trogocytosis in RAW-macrophages. They test this hypothesis by using biotinylated cell surfaces with anti-biotin antibodies, where they find that at matched antibody concentrations they see similar fractions of trogocytosis versus phagocytosis (T vs P), and by modulating cortical tension using low concentrations of glutaraldehyde they are able to alter the T vs P fractions. Finally they provide a simple mechanical model based on force balance that they claim provides an explanation for the different regimes of their observations.

While the findings and explanation of the study are compelling, there are several issues that need addressing before their conclusions may be fully justified.

Specific comments:

1) In the context of the tripartite relationship between cortical tension and cell surface adhesion strength due to the interaction between antibodies and macrophage Fc-receptors, and T vs P ratios of target cells there is a lack of clarity (or a lack of the exploration) of the quantitative possibilities offered by the experiments in cells and the artificial GUVs. The authors follow their observation that higher tension reduces the fraction of macrophages that undergo trogocytosis and increases the fraction that exhibit phagocytosis by showing that an increase in the adhesion strength (sites) can overcome this discrepancy. They follow this up with an exploration of T vs P in NHS-biotin conjugated cells, labelled with the same concentration of anti-biotin antibody. They find that anti-biotin-labelled cells undergo more trogocytosis compared to matched anti-CD67 labelled cells, but lower phagocytosis in each case (compare Fig 2B with C). It is likely that the number of anti-biotin antibodies per μm^2 in Fig 2C is higher than the anti-CD67 antibodies in 2B. It would be important to provide a quantitative analysis of the amount of surface anti-biotin antibody in each case to provide an explanation for the difference between these two results. A detailed measurement of the number of anti-biotin antibodies per μm^2 , and a titration of this is required.

2) They subsequently use biotin-lipid labelled GUVs opsinized by anti-biotin antibodies to show that they can skew the fraction of macrophages that exhibit of T vs P macrophages by altering the tension in the vesicle. This is followed up by altering the tension in HL60 and Jurkat cells using glutaraldehyde; this is at best an unusual way of altering cortical tension in these cells. It has not been well characterized except in Red blood cells (ref 19), which are enucleated cells lacking a similar cortical actin network as the nucleated macrophages. An alternative method to address changes in cortical tension would be required to provide a rigorous test of this idea. They authors could use non-muscle myosin2 inhibitors or activators

that may also provide similar changes in cortical tension and explore the consequences thereof.

3) In the abstract and the discussion, it may not be correct to say that there is no molecular switch required for the two processes- it is likely that the mechanical context induces a switch in cell behavior (see the movies supplied) that cause a change in cell engulfment behavior.

4) Finally, they introduce a theoretical framework that links the tension-mediated length scale, R_{min} (estimated at 0.1–1 μm), to the decision-making process of macrophages, whether to engage in trogocytosis versus phagocytosis. They also predict that the critical antibody density, below which active stresses are too small to trigger significant trogocytosis, must be proportional to the target cell tension, and they support this prediction with experiments (but see above). By reducing the complex interplay of cellular forces to a force balance between active stress and cortical tension, the authors provide a clear physical picture of how mechanical properties dictate immune cell behaviour in an intuitively simple framework. The predictions, such as the critical antibody density threshold and the expected deformation (or bite) sizes, are consistent with experimental measurements using both cultured cells and GUVs.

Areas for Improvement

- While the predicted length scales (0.1–1 μm) align well with the observed trogocytic bite sizes, a more precise quantitative comparison could enhance the model's assumptions and conclusions. The current analysis does not account for higher-order shape deformations and appears to be based on smooth shape changes ($R_{min} \gg h$, where h is the dimple height). It could be expanded to account for higher-order deformations.
- The definitions of phagocytosis (>50% surface coverage by macrophage extensions") and trogocytosis (presence of trogocytic "bites") could be clarified. For example, it is unclear why a threshold of 50% is chosen. Does it imply that even partial engulfment is classified as phagocytosis? (as seen in Fig. 3C). Or that < 50% engulfment always proceeds towards trogocytosis?
- The current framework addresses a steady-state balance of forces. Incorporating time-dependent aspects might further refine its predictive power since it appears that the macrophages dynamically respond to changes in cortical tension of the target cell.
- If this is a general mechanical scaling law- in Fig 5C, the p_{crit} should scale with antibody density and tension- why do HL60 and Jurkat cells have such a different scaling relationship?
- The reduction of antibody density to the active stresses that act at the cell interface by assuming a linear scaling of active stresses with antibody density is at best an approximation. Discussing potential non-linearities, such as dynamic remodeling of the cytoskeleton, spatial heterogeneities, receptor clustering, or cooperative effects, would help contextualize the model's limitations.

General concerns:

In most cases where the authors wish to draw our attention to differences in capacities for T vs P, they do not provide any statistical justification for their statements results. (eg. Figs 1B, C; Fig 2A-C; Fig 4C).

Reviewer #3 (Remarks on code availability):

I am afraid that I did not have the time to review this code. It will need more time for me to review the code- so I decided to send back my comments before this is also further delayed.

**For Nature Portfolio general information and news for authors, see <http://npg.nature.com/authors>.

Version 1:

Decision Letter:

Dear Dr Fletcher,

Thank you for your email asking us to reconsider our decision on your manuscript, "Target cell tension regulates macrophage trogocytosis". We are always willing to hear the authors' perspective, but we must first prioritize decisions on new submissions. We appreciate your patience while we considered this appeal.

I have now discussed your manuscript, the referees' comments and your rebuttal, in detail with my colleagues, and we would be willing to reconsider a revised manuscript provided the following issues can be addressed, and that nothing similar is accepted for publication at Nature Cell Biology or published elsewhere in the meantime. Please resubmit your revised manuscript along with source data and the included reporting summary, editorial policy checklist, and light microscopy reporting table, as outlined below:

In addition, please pay close attention to our guidelines on statistical and methodological reporting (listed below) as failure to do so may delay the reconsideration of the revised manuscript. In particular please provide:

- a Supplementary Table including all numerical source data in Excel format, with data for different figures provided as

different sheets within a single Excel file. The file should include source data giving rise to graphical representations and statistical descriptions in the paper and for all instances where the figures present representative experiments of multiple independent repeats, the source data of all repeats should be provided.

- for any revision that includes light microscopy data, we ask our authors to please include a completed light microscopy reporting table https://www.nature.com/documents/Light_microscopy_reporting_table.xlsx to ensure the methods are described thoroughly. The table will be available to reviewers and ultimately published should the manuscript be accepted at the journal.

On resubmission please provide the completed Reporting Summary (found here <https://www.nature.com/documents/nr-reporting-summary.pdf>). This is essential for reconsideration of the manuscript and this document will be available to editors and referees in the event of peer review. For more information see below. Please also ensure that the presentation of statistical information in the revised submission complies with Nature Cell Biology's statistical guidelines (see below).

Please use the link below to submit the complete manuscript files, and include a point-by-point response to the complete reviewer comments, verbatim as provided in their reports.

Link Redacted

Please let us know how you wish to proceed and when we can expect your revised manuscript.

With kind regards,

Daryl

Daryl Jason Verzosa David, PhD

Senior Editor, Nature Cell Biology
Advisory Editor, npj Biological Physics and Mechanics
Nature Portfolio

Heidelberger Platz 3, 14197 Berlin, Germany
Email: daryl.david@nature.com
ORCID: <https://orcid.org/0000-0002-9253-4805>

GUIDELINES FOR EXPERIMENTAL AND STATISTICAL REPORTING

REPORTING REQUIREMENTS – To improve the quality of methods and statistics reporting in our papers we have recently revised the reporting checklist we introduced in 2013. We are now asking all life sciences authors to complete a reporting summary (found here <https://www.nature.com/documents/nr-reporting-summary.pdf>) that collects information on experimental design and reagents. This document is available to referees to aid the evaluation of the manuscript. Please note that this form is a dynamic 'smart pdf' and must therefore be downloaded and completed in Adobe Reader. We will then flatten it for ease of use by the reviewers. If you would like to reference the guidance text as you complete the template, please access these flattened versions at <http://www.nature.com/authors/policies/availability.html>.

STATISTICS – Wherever statistics have been derived the legend needs to provide the n number (i.e. the sample size used to derive statistics) as a precise value (not a range), and define what this value represents. Error bars need to be defined in the legends (e.g. SD, SEM) together with a measure of centre (e.g. mean, median). Box plots need to be defined in terms of minima, maxima, centre, and percentiles. Ranges are more appropriate than standard errors for small data sets. Wherever statistical significance has been derived, precise p values need to be provided and the statistical test used needs to be stated in the legend. Statistics such as error bars must not be derived from n<3. For sample sizes of n<5 please plot the individual data points rather than providing bar graphs. Deriving statistics from technical replicate samples, rather than biological replicates is strongly discouraged. Wherever statistical significance has been derived, precise p values need to be provided and the statistical test stated in the legend.

Version 2:

Decision Letter:

Our ref: NCB-A56787B

3rd September 2025

Dear Dr. Fletcher,

Thank you for submitting your revised manuscript "Target cell tension regulates macrophage trogocytosis" (NCB-A56787B). It has now been seen by the original referees and their comments are below. The reviewers find that the paper has improved in revision, and therefore we'll be happy in principle to publish it in Nature Cell Biology, pending minor revisions to satisfy the referees' final requests and to comply with our editorial and formatting guidelines.

The current version of your manuscript is in a PDF format. Therefore, please email us a copy of the file in an editable format (Microsoft Word or LaTeX)-- we can not proceed with PDFs at this stage.

Thank you again for your interest in Nature Cell Biology Please do not hesitate to contact me if you have any questions.

Sincerely,
Daryl

Daryl Jason Verzosa David, PhD

Senior Editor, Nature Cell Biology
Advisory Editor, npj Biological Physics and Mechanics
Nature Portfolio

Heidelberger Platz 3, 14197 Berlin, Germany
Email: daryl.david@nature.com
ORCID: <https://orcid.org/0000-0002-9253-4805>

Reviewer #1 (Remarks to the Author):

The authors' diligent efforts have transformed what I initially considered a preliminary study into a robust and impactful piece of work. The conclusions are now well-supported by a combination of clever model system experiments, new data from physiologically relevant contexts, and sound theoretical reasoning. The manuscript is significantly strengthened, and I am happy to recommend it for publication.

I have one final, minor suggestion:

If the data are readily available, the authors could consider adding a panel to the supplementary materials (Figure S7) that overlays the density-response curves for different opsonins (e.g., anti-CD47 vs. anti-biotin) on the same axes. This would provide a more intuitive visual for their conclusion that the response is insensitive to antigen identity and is instead dominated by density. However, I would not recommend that new experiments be performed for this purpose if the data is not already on hand.

Reviewer #2 (Remarks to the Author):

The authors have done a nice job of addressing my concerns, and I have no further comments.

Reviewer #3 (Remarks to the Author):

In their revised manuscript, the authors have addressed most of the issues raised. Overall the manuscript is strengthened by their new data. In particular, the addition of new experimental data on the phagocytosis versus trogocytosis balance of apoptotic cells and filamin-A deficient cells with consistent consistent effects on the ratio of the two processes. Together with the inclusion of phagocytosis data with glutaraldehyde treatment, and the opsonization tests all strengthen the experimental case. The point that seems less fully resolved is the request for more direct, dynamic measurements of cortical tension. The

authors provide a reasonable explanation of the limitations of probes such as Flipper-TR and argue that cortical tension, not membrane tension alone, is the relevant parameter.

Some caveats still remain and these may be addressed in the discussion.

i) In the quantitative comparison of predicted vs. observed bite sizes: we had referred to 'higher-order deformations,' ie. non-smooth, nonlinear shape changes rather than thermally driven higher modes. Therefore the question still remains whether the scaling analysis, which assumes smooth deformations ($R_{\min} \gg h$), is still valid in regimes where shapes are more complex (e.g., sharp indentations or non-smooth geometries). This seems relevant, since macrophage bites may not always conform to a smooth-dimple approximation.

ii) Definitions of phagocytosis vs. trogocytosis in the GUVs: The authors have clarified that they define phagocytosis as >50% engulfment of the target and trogocytosis as <50%, motivated by the constraints of their assay on GUVs. This operational threshold is somewhat arbitrary. Because this definition underpins the classification of all events, it has important consequences for the interpretation of the findings. For instance, if partial engulfment (>50%) do not always proceed to full phagocytosis, or if some <50% engulfment represent stalled phagocytic attempts, then the categorization may not strictly reflect two distinct biological outcomes. It would be important to how sensitive their conclusions are to this threshold, and whether their key takeaways, particularly the phase diagram, remain robust if the cutoff were shifted from the 50% arbitrary choice.

iii) Scaling law: While the authors acknowledge that HL60 and Jurkat cells follow different apparent slopes and attribute this to nonlinearities, this response feels incomplete given that the scaling law forms a central pillar of their interpretation. It would be helpful for the authors to discuss more concretely whether the scaling is expected to be universal or whether it should be viewed as an approximate trend with cell-type-specific deviations. If the latter is the case, it would be best not to refer to this as a scaling law, since this will change with cell-type and cortical tension of target.

iv) The authors are encouraged to make the limitations of their interpretation more explicit so that readers can better appreciate the scope of their conclusions.

Version 3:

Decision Letter:

Dear Dr Fletcher,

I am pleased to inform you that your manuscript, "Target cell cortical tension regulates macrophage trogocytosis", has now been accepted for publication in Nature Cell Biology.

Please note that *Nature Cell Biology* is a Transformative Journal (TJ). Authors may publish their research with us through the traditional subscription access route or make their paper immediately open access through payment of an article-processing charge (APC). Authors will not be required to make a final decision about access to their article until it has been accepted. Find out more about Transformative Journals

Authors may need to take specific actions to achieve compliance with funder and institutional open access mandates. If your research is supported by a funder that requires immediate open access (e.g. according to Plan S principles or the NIH public access policy) then you should select the gold OA route, and we will direct you to the compliant route where possible. Because authors warrant under our subscription licensing terms that they haven't committed to licensing any version of their article under a licence inconsistent with the terms of our agreement – including the applicable embargo period – publication under the subscription model isn't suitable for authors whose funders require no embargo.

If you have not already done so, we strongly recommend that you upload the step-by-step protocols used in this manuscript to protocols.io (<https://protocols.io>), an open online resource that allows researchers to share their detailed experimental know-how. All uploaded protocols are made freely available and are assigned DOIs for ease of citation. Protocols and Nature Portfolio journal papers in which they are used can be linked to one another, and this link is clearly and prominently visible in the online versions of both. Authors who performed the specific experiments can act as primary authors for the Protocol as they will be best placed to share the methodology details, but the Corresponding Author of the present research paper should be included as one of the authors. By uploading your Protocols onto protocols.io, you are enabling researchers to more readily reproduce or adapt the methodology you use, as well as increasing the visibility of your protocols and papers. You can also establish a dedicated workspace to collect your lab Protocols. Further information can be found at <https://www.protocols.io/help/publish-articles>.

Nature Cell Biology encourages authors presenting evidence for cell, biological, molecular, and genetic interactions to consider communicating these findings using Biofactoid (<https://biofactoid.org/>). This tool helps users share a searchable representation of interactions (e.g. binding, gene expression, post-translational modification) between genes, gene products, or chemicals. Information added to Biofactoid, with author attribution, is shared on social media and public databases, such as Pathway Commons, where it can be discovered and analyzed in the context of a large and growing corpus of knowledge.

With kind regards,

Daryl

Daryl Jason Verzosa David, PhD

Senior Editor, Nature Cell Biology
Advisory Editor, npj Biological Physics and Mechanics
Nature Portfolio

Heidelberger Platz 3, 14197 Berlin, Germany
Email: daryl.david@nature.com
ORCID: <https://orcid.org/0000-0002-9253-4805>

** Visit the Springer Nature Editorial and Publishing website at www.springernature.com/editorial-and-publishing-jobs for more information about our career opportunities. If you have any questions please click here.**

Response to Reviewers

“Target cell tension regulates macrophage trogocytosis”

Caitlin E. Cornell et al

Reviewer #1 (Remarks to the Author):

This manuscript explores how target cell cortical tension regulates macrophage trogocytosis versus phagocytosis. The authors propose that the target cell, rather than the macrophage, determines the choice between trogocytosis and phagocytosis and that macrophages do not require a distinct molecular pathway for trogocytosis. This work is potentially important because it shifts the focus from traditional molecular signaling mechanisms to the biophysical properties of target cells, offering new insights into how mechanical cues influence immune cell behavior. Understanding how target cell mechanics dictate macrophage responses could have broad implications for fields such as cancer immunotherapy, infection control, and the clearance of apoptotic cells, where changes in cell stiffness and membrane tension often occur. However, I believe that the current experimental data are still too preliminary to fully support the authors' conclusions. While the study likely establishes a correlation between target cell tension and the macrophage's response, the evidence provided is insufficient to establish a direct causal relationship. If the authors can provide more convincing data to address the concerns regarding causality, I would be happy to support the publication of this work. Below are my detailed comments and experimental suggestions that I believe will significantly strengthen the manuscript's conclusions

We thank the reviewer for their positive assessment of our work. We are glad that we have presented convincing data establishing a correlation between target cell tension and macrophage response. To address the reviewer's concerns, we have included more experimental data to further support a causal relationship. The new experiments and findings are described below.

Comment 1: Many of my concerns arise from the dynamic micropipette aspiration experiment, primarily due to the low throughput of this method. This issue is particularly significant in Figure 3, as the data presented there form the core of the manuscript's conclusions. The authors' approach of using GUVs (Giant Unilamellar Vesicles) as substitutes for target cells is conceptually clever because it effectively eliminates potential confounding factors arising from target cell membrane composition and cytoskeletal interactions, allowing for a more direct examination of the role of membrane tension. However, the current data are not sufficiently convincing. The authors only show a few representative cases to support the claim that increasing target membrane tension can induce a shift in macrophage behavior from trogocytosis to phagocytosis. The sample size in these experiments appears limited, and the observed timing of phagocytosis onset could be coincidental rather than indicative of a true causal relationship.

To address this concern, I suggest that the authors prepare a series of GUVs with different membrane tensions by varying lipid compositions (e.g., adjusting the POPC/DOPE ratio). By using the membrane tension probe (Flipper-TR® fluorescent cell membrane tension probe), they

can validate and quantify the differences in membrane tension among these GUV preparations. Subsequently, real-time fluorescence imaging can be used to monitor macrophage interactions with these tension-defined GUVs. This approach would not only enable the collection of a larger dataset in a single experimental session, thereby improving reliability and reproducibility, but also provide direct visual evidence supporting the causal relationship between target membrane tension and the macrophage's choice between trogocytosis and phagocytosis.

We agree that pipette aspiration is a low throughput technique (but a powerful one!). The dynamic experiments were intended to illustrate how changes in GUV membrane tension cause macrophages to switch from trogocytosis to phagocytosis. To provide additional data beyond the dynamic experiments, we conducted a more extensive set of pipette aspiration measurements (N=27) shown in Figure 3C, where we measured the membrane tension of GUVs that were either being trogocytosed or phagocytosed. In those experiments, we found clear membrane tension regimes for phagocytosis (>1 mN/m) and trogocytosis (<1 mN/m).

We appreciate the reviewer's suggestion to vary lipid compositions and use the fluorescent probe Flipper-TR. To test this approach, we first made GUVs with two different lipid compositions (POPC and POPC + 20% chol). While the elastic modulus of a membrane depends on lipid composition¹, membrane tension will be dependent on the stress applied (e.g., osmotic or pipette aspiration). Even for a single composition of GUV, it has been shown that the osmotic pressure oscillates for a single GUV compared to the bulk of GUVs due to random pore formation, even in an isotonic solution². At any given time, the membrane tension for a series of GUVs of the same composition, as measured by micropipette aspiration, can vary greatly, and this variation appears to blur any tension differences there may be between compositions (Response Figure 1A).

We next followed the suggestion to measure GUV membrane tension in our system using Flipper-TR, which reports changes in lipid order and relative values of membrane tension³. We imaged GUVs (79 mol% POPC, 20 mol% cholesterol, 1 mol% Biotin-PE, 0.5 mol% PEG2K-PE, and 1 mol% Flipper-TR) on a Stellaris laser scanning confocal microscope equipped with a FLIM system to measure lifetimes of Flipper-TR in GUVs. The GUVs were electroformed in sucrose osmotically matched to L-15 cell culture media. We observed that a visibly low-tension GUV can have a higher lifetime than a visibly high-tension GUV, while in another frame, a low-tension GUV has a lower lifetime than a high-tension GUV (Response Figure 1B). This discrepancy suggests that Flipper-TR may not be the ideal read-out of membrane tension in our system. It also highlights the difficulty of using changes in osmotic pressure to reliably and systematically vary the membrane tension of GUVs. Stochastic pore formation in response to changes in osmotic pressure makes it challenging to control the membrane tension of GUVs in bulk. The assay also requires co-incubation of GUVs with living cells, which limits the range of osmotic variations that can be investigated.

Response Figure 1. Investigation of the connection between membrane composition and membrane tension. **A.** GUVs made primarily of POPC lipids, or GUVs containing POPC + 20 mol% cholesterol, have a non-significant (via student's T test) difference in membrane tension as measured by micropipette aspiration. **B.** In one field of GUVs in L-15 media, a visibly taut vesicle has a shorter lifetime than a visibly floppy vesicle. In another field of the same sample, a visibly taut vesicle has a longer lifetime than a visibly floppy vesicle.

Comment 2: Similarly, the use of the Flipper-TR® membrane tension probe in the experiments presented in Figure 2 and Figure 4 would greatly enhance the robustness and persuasiveness of the data. Incorporating this probe into the experimental design would provide real-time, quantitative, and visual evidence of membrane tension differences, allowing for a more direct validation of the authors' central hypothesis that target cell tension regulates the choice between trogocytosis and phagocytosis. This addition is particularly important because, in both experiments, the current assessment of target cell tension relies on indirect methods or static measurements, which may not accurately reflect the actual membrane tension experienced by macrophages during the interaction. Without direct and dynamic tension measurements, it remains unclear whether the observed changes in macrophage behavior are truly driven by differences in membrane tension or potentially confounded by other factors, such as alterations in membrane composition, cell surface protein distribution, or fixation-induced artifacts. By applying membrane tension probe to the target cells used in Figure 2, the authors could verify whether differences in trogocytosis and phagocytosis frequencies correlate directly with measurable variations in membrane tension across different cell types. Similarly, incorporating the probe into the Figure 4 experiments—where target cell tension is increased through glutaraldehyde fixation—would confirm whether the observed suppression of trogocytosis and potential enhancement of phagocytosis correspond to genuine tension increases, rather than nonspecific effects of the fixation process.

We appreciate that our measurements of membrane tension in Figures 2 and 4 would benefit from a more direct and dynamic method. However, in addition to the membrane composition issues with Flipper-TR noted above, our manuscript argues that it is not membrane tension alone, but rather cortical tension (the combination of membrane tension and membrane-cortex attachment^{4,5}) that determines the macrophage response to a target cell. In GUVs, our micropipette measurements only probe membrane tension since there is no cortical cytoskeleton. However, in our cell experiments, we are probing cortical tension that includes membrane interactions with the cortical cytoskeleton. Since Flipper-TR does not capture membrane-cortex attachment in the same way as micropipette aspiration, it could miss an important contributor to the cell surface properties that appear to influence whether macrophages trogocytose or phagocytose.

Comment 3: While the manuscript focuses on how target cell membrane tension and antibody density influence the choice between trogocytosis and phagocytosis, it lacks a detailed discussion of the intracellular signaling mechanisms by which macrophages make this decision. Both processes require Fcγ receptor activation, but it remains unclear how differences in signal strength, duration, or downstream pathway activation lead to the selection of one response over the other.

We thank the reviewer for pointing out that we overlooked this important discussion point. To address this omission, we have included the following text in the discussion section:

Discussion Section, Paragraph 6: “But if, as we hypothesize, the receptors and downstream signaling molecules involved in the phagocytosis and trogocytosis pathways are the same, how might the response differ between the two processes? Both processes require a minimum surface antibody density ($\sim 10^2$ molecules/ μm^2) to initiate. If the target has low cortical tension, membrane fluctuations allow for a higher probability of binding events. Subsequently, the machinery required for macrophage protrusion and force generation (e.g. myosin motors and nucleation promoting factors that activate Arp 2/3 to produce branched actin networks at the cell-cell interface) is recruited to a smaller area. When the cortical tension is low, the surface is more permissible to deformation and scission than a high-tension surface.”

Comment 4: The study focuses exclusively on antibody-mediated trogocytosis and phagocytosis. Have the authors considered whether the tension-regulation mechanism applies more broadly beyond antibody-dependent processes? For example, would similar trends be observed with other opsonins (e.g., complement components) or natural ligands such as phosphatidylserine on apoptotic cells? Investigating this could help clarify whether the proposed mechanism is specific to Fcγ receptor engagement or part of a more general mechanobiological response across different forms of target recognition.

We agree that this is a very interesting avenue of study. We have included a new experiment in the manuscript examining phagocytosis/trogocytosis of healthy cells with and without antibody, as well as apoptotic cells with and without antibody. Strikingly, cells that have undergone apoptosis are significantly stiffer (up to 3x) than their healthy counterparts⁶⁻⁸. We hypothesized

that this increase in stiffness of apoptotic cells would bias macrophages towards phagocytosis over trogocytosis.

To induce apoptosis in Jurkat T cells, we used raptinal, a potent activator of caspase-3 which induces apoptosis within 30 minutes of incubation and results in >80% cell death (as measured by propidium iodide) within 2 hours^{9,10}. In the presence of 10 μ M raptinal for 2 hours, we measured 85% of Jurkat T cells dead as marked by propidium iodide (in contrast to 1% of untreated Jurkat T cells). When we co-incubated LPS-stimulated macrophages with IgG-opsonized and raptinal-treated Jurkats, we saw a 3x increase in phagocytosis (and 15x decrease in trogocytosis) over healthy Jurkat T cells (Response Figure 2A left). Because apoptotic cells have additional phagocytic ‘eat me’ signals (e.g., phosphatidyl serine lipids flipped to the outer leaflet of the plasma membrane), we repeated the experiment without IgG opsonization (Response Figure 2A right). To measure trogocytosis without a fluorescent IgG, we biotinylated the target cells and then labeled the membrane with AlexaFluor-647 streptavidin (and dark anti-biotin in the opsonized case). As shown below, phagocytosis and trogocytosis in both raptinal-treated and untreated Jurkats are attenuated. However, phagocytosis is still higher for raptinal-treated Jurkats, and trogocytosis is higher for untreated Jurkats. Response Figure 2 has been added to the revised manuscript as Figure 4A.

Results “Increasing tension in cultured cells suppresses trogocytosis and increases phagocytosis.” Paragraph 1-2: Interestingly, cells that have undergone apoptosis are significantly stiffer (up to 3x) than their healthy counterparts^{26–28}. Macrophages are responsible for efferocytosis, or the phagocytic clearance of apoptotic cells. We hypothesized that the increase in stiffness upon apoptosis would bias macrophages towards phagocytosis over trogocytosis.

To induce apoptosis in Jurkat T cells, we used raptinal, a potent activator of caspase-3, which induces apoptosis within 30 minutes of incubation, resulting in >80% cell death (as measured by propidium iodide) within 2 hours^{29,30}. In the presence of 10 μ M raptinal for 2 hours, we measured 85% of Jurkat T cells dead as marked by propidium iodide (in contrast to 1% of untreated Jurkat T cells). After co-incubation with LPS-stimulated macrophages, we saw a 3x increase in phagocytosis (and a 15x decrease in trogocytosis) over healthy cells when Jurkats were opsonized with IgG (Fig. 4A, left). Because apoptotic cells have additional ‘eat me’ signals (e.g., phosphatidylserine lipids flipped to the outer leaflet of the plasma membrane), we repeated the experiment without IgG opsonization (Fig. 4A, right). To measure trogocytosis without a fluorescent IgG, we biotinylated the target cells and then labeled the membrane with AlexaFluor 647 streptavidin (and dark anti-biotin for opsonized cells). In the right panel of Fig. 4A, phagocytosis and trogocytosis in both raptinal-treated and untreated Jurkats are attenuated; however, phagocytosis is still higher for raptinal-treated Jurkats.”

Response Figure 2. Increasing cell tension through apoptosis suppresses trogocytosis and increases phagocytosis. Biotinylated Jurkat T cells were labeled with AlexaFluor647 streptavidin and opsonized with anti-biotin IgG at 1385 ± 189 molecules/ μm^2 (left) or unopsonized (right) and treated with 10 μM raptinal, or untreated, and co-incubated with LPS-stimulated macrophages for 2 hrs. Statistical significance was computed via student's T test.

Reviewer #2 (Remarks to the Author):

In this paper, the authors convincingly demonstrate that the process of trogocytosis, the “nibbling” of the plasma membrane by phagocytes (such as macrophages) is strongly influenced by the cell surface tension of the target cell membrane. Sophisticated methods are employed to reach this conclusion, with convincing results. The authors conclude that this the major factor in the “decision” to either trogocytosis or phagocytose a target cell. I have a number of concerns regarding this conclusion and the physiological relevance of their findings.

Comment 1: One major concern with their conclusion is that although the role of target cell surface tension in the efficiency of trogocytosis is clearly demonstrated, no evidence is provided that cells with high surface tension are preferentially phagocytosed. In their system, phagocytosis is a rare event, and increasing surface tension with low concentrations of glutaraldehyde did not appear to increase this (certainly not anywhere near the levels of trogocytosis they observe in untreated cells). While their conclusions regarding trogocytosis seem solid to this reviewer, conclusions regarding phagocytosis are not supported by data.

We thank the reviewer for noting this omission. We were focused on the new finding about the tension dependence of trogocytosis and did not highlight the changes in phagocytosis that occur upon an increase in cell tension. We have revised Figure 4E in the main text to present the increase in phagocytosis that occurs with gentle glutaraldehyde fixation, in addition to the decrease in trogocytosis that occurs.

To further address the reviewer's question about phagocytosis, we have added a subfigure with new experiments measuring phagocytosis and trogocytosis of Jurkat T cells, HL60s, and M2 cells with LPS-stimulated RAW 264.7 macrophages (Response Figure 3A-B; Figure 4C). M2 cells are deficient in filamin A, a crosslinker of F-actin at the cortex, and they are known to be softer than similar cells expressing Filamin A^{11,12}. In comparison to Jurkat T cells and HL60s, M2 cells opsonized at the same surface density of anti-CD47 are trogocytosed significantly more (Response Figure 3B), consistent with our findings that lower cortical tension results in greater trogocytosis.

To investigate the amount of phagocytosis that can occur, we co-incubated the macrophages and target cells for 2 hours. In this case, we see that trogocytosis is clearly higher with Jurkats than with HL60s, and that LPS-stimulated macrophages phagocytose HL60s at a greater percentage than they trogocytose them. While macrophages do not phagocytose HL60s more than Jurkats, the *ratio* of phagocytosis to trogocytosis is 2x greater for Jurkats. We have added Response Figure 3 as Fig. 4B-C in the revised manuscript and updated the text to reflect our findings:

Results section, “Increasing tension in cultured cells suppresses trogocytosis and increases phagocytosis.” Paragraph 3: “To further investigate the role of cortical tension in trogocytosis of living cells, we co-incubated LPS-stimulated macrophages with a melanoma-derived cell type, M2. M2 cells are deficient in filamin A, a crosslinker of F-actin at the cortex. The cortex of M2 cells is weakened and, as a result, they are measurably softer than similar cells expressing Filamin-A^{31,32}. In comparison to Jurkat T cells and HL60s, M2 cells opsonized at the same surface density of anti-CD47 are trogocytosed significantly more (Fig. 4C), consistent with our findings that lower cortical tension results in greater trogocytosis.”

Results section, “Trogocytosis differs for target cells depending on their cortical tension”, Paragraph 1: “Notably, this is also the case for LPS-stimulated macrophages, especially for the filamin-deficient soft melanoma cell line, M2 (Fig. 4C). Strikingly, while LPS-stimulated macrophages phagocytose HL60s and Jurkats at a statistically similar amount, the ratio of trogocytosis to phagocytosis is significantly higher for Jurkats than for HL60s (~2x).”

Response Figure 3. Filamin-A-deficient cancer cells with lower cortical tension are trogocytosed. **A.** Confocal micrographs of LPS-stimulated RAW 264.7 cells labeled with CellTracker Green CMFDA trogocytosing M2 cells opsonized with AlexaFluor647 anti-CD47 IgG. White arrows indicate trogocytic bites. **B.** Percent of LPS-stimulated macrophages that have phagocytosed (red bars) or trogocytosed (blue bars) anti-CD47-opsionized HL60s, Jurkat T cells,

and M2 cells. The antibody surface density was matched between the three cell types at 163 ± 8 molecules/ μm^2 . Statistical significance was determined via a student's T-test.

Comment 2: While trogocytosis has been described as having a role in immune evasion in pathogenic amoeba and in nematodes, it is unclear if trogocytosis plays essential roles in mammalian physiology. No attempt to identify a physiological role for the mechanism they describe is offered. While treatment with glutaraldehyde is useful for their analysis, what physiological conditions influence surface tension sufficiently to affect trogocytosis? Further, when trogocytosis is inhibited (but phagocytosis is presumably intact) what are the consequences? For example, we can speculate that trogocytosis of anti-CD47-treated cells might limit phagocytic killing of these cells; does surface tension account for differences in the ability of anti-CD47 (or SIRPa) treatment to result in killing of some cell lines but not other? (Note, I am not insisting on this experiment but offer it as an example). There are other claims as to the function of trogocytosis in mammalian cells. In the absence of a physiological consequence of trogocytosis, we are left with only a mechanism for a mildly interesting phenomenon without context.

In the manuscript, we provide several recent examples of studies in which the physiological relevance of trogocytosis is established. For example, we reference Gao, X. et al., *Science* 2024, which demonstrates a physiologically relevant role for trogocytosis in bone marrow-derived macrophages (BMDMs) and hematopoietic stem cells (HSCs) in mice and in humans. In that paper, trogocytosis between BMDMs and HSCs results in a subpopulation of HSCs that display BMDM proteins, such as F4/80. This subpopulation of HSCs remains in the bone marrow and never egresses to the bloodstream, unlike the other HSCs in the marrow¹³.

Since submitting our paper, several other papers have been published or submitted to BiorXiv, pointing to the clear relevance of trogocytosis in contexts as diverse as tissue regulation and stem cell development^{14,15}, to clearance of parasitic worms¹⁶ and solid tumors^{17,18}. We have included references to each of these publications in the introduction section of the revised manuscript. These references point to an emerging physiological role for trogocytosis and emphasize the need for a more mechanistic and cell biological understanding of the phenomenon, which our manuscript aims to provide.

Introduction Paragraph 2 “For example, human and mouse macrophages have been observed to nibble hematopoietic stem cells, marking them for retention in the bone marrow⁷. In zebrafish, hematopoietic stem cells are also selectively nibbled by macrophages, marking them for clonal hematopoiesis⁸. Similarly, tissue resident macrophages of the brain, microglia, selectively nibble synapses of neurons to prune connections in the developing mouse brain⁹. Strikingly, macrophages have also been shown to trogocytose targets that are too large to phagocytose, such as the parasitic worm, *Schistosoma*¹⁰, or large tumors¹¹.”

Lastly, the first author attended the recent Gordon Research Conference on Phagocytes, where she gave a talk on this work. There was interest from clinical researchers as well as basic researchers, and we are following up now on potential collaborations to use our methods to investigate new systems.

Comment 3: Does trogocytosis of living cells depend on inhibition of CD47-SIRP α interaction? The biotin experiment doesn't answer this question, since CD47 is biotinylated in this situation. Are other antibodies to cell surface proteins engulfed in the absence of blocking CD47? If so, does surface tension play a role in such situations? This is important, since it is assumed that surface tension controls the ability of phagocytes to "nibble" small pieces of membrane, but if the effects depend on blocking CD47-SIRP α , then it remains possible that surface tension influences this blockade.

We thank the reviewer for the question. To directly address it, we co-incubated RAWs with Raji B cells opsonized with anti-CD19, a surface protein abundant on B cells. The anti-CD19 antibodies do not block CD47, a potent 'don't eat me' signal, and will therefore not hinder binding of macrophage SIRP α to target CD47. According to our measurements, the average surface density of anti-CD19 on Raji B cells was 212 molecules/ μm^2 compared to an average surface density of 200 molecules/ μm^2 anti-CD47 (Fig. 2B of the revised manuscript). Raji B cells opsonized with anti-CD19 show ~4x more trogocytosis than phagocytosis (Response Figure 4), which is comparable to the results anti-CD47, albeit with an overall 2x lesser magnitude. We expect that the lesser magnitude of both trogocytosis and phagocytosis can be attributed to the lack of a CD47-SIRP α blocking effect from the anti-CD47 antibodies. Response Figure 4 below is included in the supplement as Fig. S4 of the revised manuscript. We have also added the following text into the main manuscript:

Results Section, "Trogocytosis differs for target cells depending on their cortical tension." Paragraph 2: "When antibody surface density is matched, we observe that trogocytic efficiency is highest in Jurkats, while phagocytic efficiency is highest in HL60s. To ensure that our results are not simply from blocking the 'don't eat me' interaction of target CD47 with macrophage SIRP α (due to anti-CD47 or biotinylation of CD47), we co-incubated macrophages with Raji B cells opsonized with anti-CD19. We observed that Raji B cells opsonized with anti-CD19 show ~4x more trogocytosis than phagocytosis, which is comparable to the results in Fig. 2B, albeit with an overall 2x lesser magnitude (Fig. S4)."

Response Figure 4. Phagocytosis and trogocytosis of Raji B cells opsonized with AlexaFluor647 anti-CD19 at a surface density of 212 molecules/ μm^2 .

Comment 4: The results in Figure S2 are clear but unconvincing. Since trogocytosis is measured here with uptake of labeled antibody, the ability of anti-FcR to “block” trogocytosis is not surprising. Does inhibition of FcR prevent uptake of biotinylated proteins in this setting?

We thank the reviewer for the question. To address this point, we carried out a new experiment that demonstrates a lack of membrane transfer from target Jurkat T cells to RAW 264.7 macrophages when Jurkat cells are not opsonized with antibody. We accomplish this by biotinylating the surface proteins of Jurkat T cells and adding AlexaFluor647 streptavidin to label the membrane. By tracking the transfer of fluorescent streptavidin, we show that trogocytosis occurs only minimally in the absence of opsonization (Response Figure 5). We’ve added this data to the supplement as part of Fig. S2 of the revised manuscript.

Results “Macrophages phagocytose and trogocytose cultured cells.” Paragraph 6: “Similarly, we observed that biotinylated, but non-opsonized, target cells with AlexaFluor647 streptavidin-labeled membranes are not trogocytosed or phagocytosed, confirming that trogocytosis is dependent on FcR engagement (Fig. S2C).”

We also show that trogocytosis is not the result of the uptake of extracellular vesicles by incubating RAWs with the supernatant from labeled Jurkats. Notably, when macrophages are stimulated with LPS, there is higher background trogocytosis, which is presumably due to the increased activity of LPS-stimulated macrophages.

Response Figure 5. Macrophage trogocytosis depends on FcR engagement. Trogocytosis of AlexaFluor647 anti-biotin IgG opsonized Jurkat T cells by LPS-stimulated macrophages are unstimulated macrophages. When Jurkat T cells are uncoated but labeled with AlexaFluor647 streptavidin, trogocytosis is decreased. Similarly, when the supernatant from opsonized Jurkats is added to macrophages, there is minimal uptake of AlexaFluor647 streptavidin-labeled membrane.

5. Figure 2. The conclusion that HL60 cells “are phagocytosed more than the other lines” is not clear—are these small differences significant (and sufficiently powered)?

We agree that this statement was not clear and have added supplemental experiments with LPS-stimulated RAWs to address this point. In this case, HL60s are not necessarily phagocytosed

more than the other cell lines, but Jurkats are clearly trogocytosed more than HL60s. We have updated the text accordingly (see also response to Reviewer 2 Comment 1).

6. *Figure 4. As mentioned in #1. Increasing cell surface tension with glutaraldehyde clearly inhibits trogocytosis, however the prediction would be that it increases phagocytosis. Is this true? Only trogocytosis was measured. If phagocytosis does not appreciably increase as trogocytosis decreases (and I have to assume that if this were the case, it would have been shown), then the study does not describe the difference between these two processes, but rather a requirement for efficient trogocytosis itself.*

We have updated the manuscript to include data on phagocytosis from the experiments with glutaraldehyde. Consistent with our model, phagocytosis increases significantly at higher concentrations of glutaraldehyde, while trogocytosis significantly decreases (Fig. 4E).

Reviewer #3 (Remarks to the Author):

In their manuscript, Cornell et al., try to understand why a different macrophage lines including primary macrophages would be directed to trigger trogocytosis versus phagocytosis of opsinized target cells. The propensity of different target cells to undergo predominantly trogocytosis or phagocytosis presented some clues for the authors, since they had observed that the predominantly trogocytosed, Jurkat and Raji cells had a lower cortical tension than the more phagocytic-prone HL60 cells. Furthermore at similar surface antibody levels, Jurkat and Raji cells appear to be show similar tendencies to trigger trogocytosis in RAW-macrophages. They test this hypothesis by using biotinylated cell surfaces with anti-biotin antibodies, where they find that at matched antibody concentrations they see similar fractions of trogocytosis versus phagocytosis (T vs P), and by modulating cortical tension using low concentrations of glutaraldehyde they are able to alter the Tvs P fractions. Finally they provide a simple mechanical model based on force balance that they claim provides an explanation for the different regimes of their observations. While the findings and explanation of the study are compelling, there are several issues that need addressing before their conclusions may be fully justified.

Comment 1: In the context of the tripartite relationship between cortical tension and cell surface adhesion strength due to the interaction between antibodies and macrophage Fc-receptors, and T vs P ratios of target cells there is a lack of clarity (or a lack of the exploration) of the quantitative possibilities offered by the experiments in cells and the artificial GUVs.

The authors follow their observation that higher tension reduces the fraction of macrophages that undergo trogocytosis and increases the fraction that exhibit phagocytosis by showing that an increase in the adhesion strength (sites) can overcome this discrepancy. They follow this up with an exploration of T vs P in NHS-biotin conjugated cells, labelled with the same concentration of anti-biotin antibody. They find that anti-biotin-labelled cells undergo more trogocytosis compared to matched anti-CD67 labelled cells, but lower phagocytosis in each case (compare Fig 2B with C). It is likely that the number of anti-biotin antibodies per μm^2 in Fig 2C is higher than the anti-CD67 antibodies in 2B. It would be important to provide a quantitative analysis of the amount of

surface anti-biotin antibody in each case to provide an explanation for the difference between these two results. A detailed measurement of the number of anti-biotin antibodies per μm^2 , and a titration of this is required.

We appreciate the reviewer's suggestion that we provide a quantitative analysis of the amount of surface antibody in our experiments using anti-biotin antibodies. In every experiment, we used fluorescent calibration beads to compute the number of antibodies present on the surface of cells based on the fluorescent signal from the flow cytometer and the average surface area of each cell type. We have updated the manuscript to include the average surface density of antibody in the figure captions for figures that do not explicitly show it. We highlight the important role of antibody surface density in revised manuscript Fig. S7, where we titrate antibody density and show the corresponding change in trogocytosis.

Comment 2: They subsequently use biotin-lipid labelled GUVs opsinized by anti-biotin antibodies to show that they can skew the fraction of macrophages that exhibit of T vs P macrophages by altering the tension in the vesicle. This is followed up by altering the tension in HL60 and Jurkat cells using glutaraldehyde; this is at best an unusual way of altering cortical tension in these cells. It has not been well characterized except in Red blood cells (ref 19), which are enucleated cells lacking a similar cortical actin network as the nucleated macrophages. An alternative method to address changes in cortical tension would be required to provide a rigorous test of this idea. They authors could use non-muscle myosin2 inhibitors or activators that may also provide similar changes in cortical tension and explore the consequences thereof.

We agree with the reviewer that glutaraldehyde is a non-physiological (but convenient) way to perturb cell cortical tension. We chose this method for two main reasons. First, our phagocytosis/trogocytosis assay is a co-incubation experiment. To have the desired effect on the cytoskeleton, drugs like non-muscle myosin2 inhibitors and other actin/motor/microtubule drugs need to be in the solution with the cells. The half-life of many of these drugs is shorter than the duration of a co-incubation experiment (i.e. blebbistatin's half-life ~ 20 min). If we were to leave the target cells bathing in the cytoskeletal drug, we would also drug the macrophages in the co-incubation experiment. The gentle glutaraldehyde treatment allowed us to pre-treat the target cells, wash them completely, and then co-incubate them with the macrophages. Second, we wanted a way to reliably titrate the cortical stiffness of cells. By titrating the concentration of glutaraldehyde, which is known to rigidify cells, we were able to effectively titrate the cortical stiffness of target cells.

However, we agree that a more physiological experiment would help to support our findings. To that end, we carried out two new experiments shown in Response Figures 2 and 3 (and included in Fig. 4A-C of the revised manuscript). Both new experiments confirmed that lower cortical tension correlates with increased trogocytosis:

1. In the first experiment, we co-incubated macrophages with Jurkat T cells treated with raptinal, a potent initiator of apoptosis. Cells that have undergone apoptosis are significantly stiffer (up to 3x) than their healthy counterparts⁶⁻⁸. In Response Figure 2, we show that apoptotic cells are phagocytosed significantly more than healthy cells.

The reverse is also true; healthy cells are trogocytosed significantly more than apoptotic cells.

2. In the second experiment, we co-incubated macrophages with a cell line derived from human melanoma with a filamin-A deficiency (M2) in which cells constitutively bleb and the membrane tension of blebs has been measured to be $\sim 2x$ less than membrane attached to the cortex¹¹. Filamin-A deficiency leads to an overall weakening of the cortex, which decreases cortical tension¹². With both decreased cortical tension and frequent membrane detachment from the cortex, we predicted that M2 cells would be trogocytosed preferentially over phagocytosis, which we show in Response Figure 3.

Comment 3: In the abstract and the discussion, it may not be correct to say that there is no molecular switch required for the two processes- it is likely that the mechanical context induces a switch in cell behavior (see the movies supplied) that cause a change in cell engulfment behavior.

We thank the reviewer for this correction, and we have changed the text in the abstract and discussion accordingly.

Comment 4: Finally, they introduce a theoretical framework that links the tension-mediated length scale, R_{min} (estimated at $0.1-1 \mu m$), to the decision-making process of macrophages, whether to engage in trogocytosis versus phagocytosis. They also predict that the critical antibody density, below which active stresses are too small to trigger significant trogocytosis, must be proportional to the target cell tension, and they support this prediction with experiments (but see above). By reducing the complex interplay of cellular forces to a force balance between active stress and cortical tension, the authors provide a clear physical picture of how mechanical properties dictate immune cell behaviour in an intuitively simple framework. The predictions, such as the critical antibody density threshold and the expected deformation (or bite) sizes, are consistent with experimental measurements using both cultured cells and GUVs.

We thank the reviewer for their positive assessment of the simple model we present to explain the tension dependence of trogocytosis. We also appreciate the suggestions for improvement. We will address each sub-comment individually.

Comment 4.1: While the predicted length scales ($0.1-1 \mu m$) align well with the observed trogocytic bite sizes, a more precise quantitative comparison could enhance the model's assumptions and conclusions. The current analysis does not account for higher-order shape deformations and appears to be based on smooth shape changes ($R_{min} \gg h$, where h is the dimple height). It could be expanded to account for higher-order deformations.

We agree that it is possible to include higher-order deformations in our model. We note that higher order, thermally driven deformations that lead to membrane fluctuations below the length scale $\sqrt{\kappa/\gamma_t}$ are highly unfavorable. In our system for the lower limit of tensions considered, $\gamma_t \sim 10^{-1}$ mN/m, and considering a membrane bending stiffness of $\kappa \approx 400 K_B T$, this length scale is $\sim 0.1 \mu m$. Since this length scale is below the scale of deformations predicted due to a

balance of active stresses and cortical tension, we have not included these higher order shape modes in our current scaling analysis. However, in a more detailed model where one is interested in the time dynamics of a trogocytic deformation/bite, these higher order shape deformations may play an important role in setting the kinetics of interfacial changes. We are working on such a spatio-temporally resolved statistical model, but we believe that it is outside the scope of this current work.

Comment 4.2: The definitions of phagocytosis (">50% surface coverage by macrophage extensions") and trogocytosis (presence of trogocytic "bites") could be clarified. For example, it is unclear why a threshold of 50% is chosen. Does it imply that even partial engulfment is classified as phagocytosis? (as seen in Fig. 3C). Or that < 50% engulfment always proceeds towards trogocytosis?

We chose the definition for phagocytosis as >50% engulfment and trogocytosis as <50% as a way of binning the data in the GUV micropipette experiments. Since we are unable to measure the tension of a fully engulfed vesicle within a macrophage, we chose an arbitrary percentage of engulfment with the assumption that engulfment greater than 50% would lead to completion of phagocytosis.

Comment 4.3: The current framework addresses a steady-state balance of forces. Incorporating time-dependent aspects might further refine its predictive power since it appears that the macrophages dynamically respond to changes in cortical tension of the target cell.

We fully agree with the reviewer that a time-resolved model would be interesting and possibly lead to new qualitative and quantitative insights about the phenomenon of trogocytosis. However, such a model involves a detailed accounting of the reaction-diffusion of species at the interface, including receptors, antibody and bystander proteins, and a coupling between these and the mechanical deformation of the interface. These take the form of coupled PDEs for the concentrations of these species as well as the height and gap at the interface. We are currently working on such a model, but we believe it is beyond the scope of this current work.

Comment 4.4: If this is a general mechanical scaling law- in Fig 5C, the p_{crit} should scale with antibody density and tension- why do HL60 and Jurkat cells have such a different scaling relationship?

We thank the reviewer for this insightful comment. While the scaling law is general, there are important non-linearities in the scaling between the active stresses developed at the interface and the antibody density (as the reviewer rightly points out in another comment) that likely lead to these observed differences in scaling between HL60s and Jurkats. Differences in this scaling across different cell lines might result from differences in the surface proteome in terms of the presence of tall bystander proteins, differences in mobility of the antibody-tagged proteins, and differences in membrane properties not captured by tension. Since it is the antibody density and tension that are experimentally measured, we do expect a positive correlation between the critical

antibody density and cortical tension, as observed and shown in Fig. 5C of the revised manuscript. We have added the following text to address this:

Results section, “A mechanical model captures the relationship between cell tensions and antibody surface density in trogocytosis” Paragraph 4: “We assume that while this scaling law is general, the difference in slope between HL60s and Jurkats is likely due to important non-linearities in the scaling between the active stresses developed at the interface and the antibody density. By taking this approximation, we do not consider the possible contributions from dynamic remodeling of the cytoskeleton, spatial heterogeneities due to lipid rafts, or receptor clustering and cooperative effects. Since we experimentally measure antibody density and cortical tension, we do, however, expect a positive correlation between the critical antibody density and cortical tension, as shown in Fig. 5C.

Comment 4.5: The reduction of antibody density to the active stresses that act at the cell interface by assuming a linear scaling of active stresses with antibody density is at best an approximation. Discussing potential non-linearities, such as dynamic remodeling of the cytoskeleton, spatial heterogeneities, receptor clustering, or cooperative effects, would help contextualize the model’s limitations.

We agree that the scaling between antibody density and magnitude of active stresses is important, and our assumed scaling, while nonlinear, is still an approximation. New experimental measurements coupled with theoretical models are likely required to elucidate this scaling in greater detail. As the reviewer points out, this scaling contains interesting nonlinearities due to effects they enumerate, of which receptor clustering and cooperativity are likely important, given their importance has already been demonstrated in the context of phagocytosis^{19,20}. We have included text that discusses these other possible non-linearities and their contribution to the effect.

Results section, “A mechanical model captures the relationship between cell tensions and antibody surface density in trogocytosis” Paragraph 4: “We assume that while this scaling law is general, the difference in slope between HL60s and Jurkats is likely due to important non-linearities in the scaling between the active stresses developed at the interface and the antibody density. By taking this approximation, we do not consider the possible contributions from dynamic remodeling of the cytoskeleton, spatial heterogeneities due to lipid rafts, or receptor clustering and cooperative effects. Since we experimentally measure antibody density and cortical tension, we do expect a positive correlation between the critical antibody density and cortical tension, as shown in Fig. 5C.

References

1. Saeedimazine, M., Montanino, A., Kleiven, S. & Villa, A. Role of lipid composition on the structural and mechanical features of axonal membranes: a molecular simulation study. *Sci Rep* **9**, (2019).
2. Karatekin, E. *et al.* Cascades of Transient Pores in Giant Vesicles: Line Tension and Transport. *Biophysical Journal* vol. 84 (2003).
3. Colom, A. *et al.* A fluorescent membrane tension probe. *Nat Chem* **10**, 1118–1125 (2018).
4. Sitarska, E. & Diz-Muñoz, A. Pay attention to membrane tension: Mechanobiology of the cell surface. *Current Opinion in Cell Biology* vol. 66 11–18 Preprint at <https://doi.org/10.1016/j.ceb.2020.04.001> (2020).
5. Chugh, P. *et al.* Actin cortex architecture regulates cell surface tension. *Nat Cell Biol* **19**, 689–697 (2017).
6. Lulevich, V., Zink, T., Chen, H. Y., Liu, F. T. & Liu, G. Y. Cell mechanics using atomic force microscopy-based single-cell compression. *Langmuir* **22**, 8151–8155 (2006).
7. Lam, W. A., Rosenbluth, M. J. & Fletcher, D. A. Chemotherapy exposure increases leukemia cell stiffness. (2007) doi:10.1182/blood-2006-08.
8. Islam, M. *et al.* Microfluidic Sorting of Cells by Viability Based on Differences in Cell Stiffness. *Sci Rep* **7**, (2017).
9. Palchaudhuri, R. *et al.* A Small Molecule that Induces Intrinsic Pathway Apoptosis with Unparalleled Speed. *Cell Rep* **13**, 2027–2036 (2015).
10. Smith, A. J. & Hergenrother, P. J. Raptinal: a powerful tool for rapid induction of apoptotic cell death. *Cell Death Discovery* vol. 10 Preprint at <https://doi.org/10.1038/s41420-024-02120-1> (2024).
11. Dai, J. & Sheetz, M. P. Membrane tether formation from blebbing cells. *Biophys J* **77**, 3363–3370 (1999).
12. Kasza, K. E. *et al.* Filamin A is essential for active cell stiffening but not passive stiffening under external force. *Biophys J* **96**, 4326–4335 (2009).
13. Gao, X. *et al.* Regulation of the hematopoietic stem cell pool by C-Kit-associated trogocytosis. *Science* **385**, eadp2065 (2024).
14. Rodrigues, C. P. *et al.* Transcripts of repetitive DNA elements signal to block phagocytosis of hematopoietic stem cells. doi:10.1126/science.eadn1629.
15. Volk, R. F. *et al.* Macrophages redeploy functional cancer cell surface proteins following phagocytosis. Preprint at <https://doi.org/10.1101/2024.09.23.613776> (2024).
16. Shen, J. *et al.* Macrophage-mediated trogocytosis contributes to destroying human schistosomes in a non-susceptible rodent host, *Microtus fortis*. *Cell Discov* **9**, (2023).
17. Barbera, S. *et al.* Trogocytosis of Chimeric Antigen Receptors between T Cells Is Regulated by Their Transmembrane Domains. *Sci. Immunol* vol. 10 <https://www.science.org> (2025).
18. Rollins, K., Fiaz, S. & Morrissey, M. Target cell adhesion limits macrophage phagocytosis and promotes trogocytosis. Preprint at <https://doi.org/10.1101/2025.02.06.636906> (2025).

19. Kern, N., Dong, R., Douglas, S. M., Vale, R. D. & Morrissey, M. A. Tight nanoscale clustering of $\text{Fc}\gamma$ receptors using dna origami promotes phagocytosis. *Elife* **10**, 1–29 (2021).
20. Jo, S. *et al.* IgG Surface Mobility Promotes Antibody Dependent Cellular Phagocytosis by Syk and Arp2/3 Mediated Reorganization of Macrophage $\text{Fc}\gamma$ Rs. *SSRN Electronic Journal* 1–24 (2021) doi:10.2139/ssrn.3908787.

Response to Reviewers

“Target cell tension regulates macrophage trogocytosis”

Caitlin E. Cornell et al

We thank all the reviewers for their thoughtful analysis and helpful comments. Please see below for point-by-point responses to the reviewer’s comments. All portions of text that are new additions to the manuscript are highlighted in blue. In the revised manuscript, all new text is similarly highlighted in blue.

Reviewer #1 (Remarks to the Author):

This manuscript explores how target cell cortical tension regulates macrophage trogocytosis versus phagocytosis. The authors propose that the target cell, rather than the macrophage, determines the choice between trogocytosis and phagocytosis and that macrophages do not require a distinct molecular pathway for trogocytosis. This work is potentially important because it shifts the focus from traditional molecular signaling mechanisms to the biophysical properties of target cells, offering new insights into how mechanical cues influence immune cell behavior. Understanding how target cell mechanics dictate macrophage responses could have broad implications for fields such as cancer immunotherapy, infection control, and the clearance of apoptotic cells, where changes in cell stiffness and membrane tension often occur.

However, I believe that the current experimental data are still too preliminary to fully support the authors' conclusions. While the study likely establishes a correlation between target cell tension and the macrophage's response, the evidence provided is insufficient to establish a direct causal relationship. If the authors can provide more convincing data to address the concerns regarding causality, I would be happy to support the publication of this work. Below are my detailed comments and experimental suggestions that I believe will significantly strengthen the manuscript's conclusions.

We thank the reviewer for their positive assessment of our work. We are glad that we have presented convincing data establishing a correlation between target cell tension and macrophage response. To address the reviewer’s concerns, we have included more experimental data to further support a causal relationship. The new experiments and findings are described below.

Comment 1: Many of my concerns arise from the dynamic micropipette aspiration experiment, primarily due to the low throughput of this method. This issue is particularly significant in Figure 3, as the data presented there form the core of the manuscript's conclusions. The authors' approach of using GUVs (Giant Unilamellar Vesicles) as substitutes for target cells is conceptually clever because it effectively eliminates potential confounding factors arising from target cell membrane composition and cytoskeletal interactions, allowing for a more direct examination of the role of membrane tension. However, the current data are not sufficiently convincing. The authors only show a few representative cases to support the claim that increasing target membrane tension can induce a shift in macrophage behavior from trogocytosis to phagocytosis. The sample size in these experiments appears limited, and the observed timing of phagocytosis onset could be coincidental rather than indicative of a true causal relationship.

To address this concern, I suggest that the authors prepare a series of GUVs with different membrane tensions by varying lipid compositions (e.g., adjusting the POPC/DOPE ratio). By using the membrane tension probe (Flipper-TR® fluorescent cell membrane tension probe), they can validate and quantify the differences in membrane tension among these GUV preparations. Subsequently, real-time fluorescence imaging can be used to monitor macrophage interactions with these tension-defined GUVs. This approach would not only enable the collection of a larger dataset in a single experimental session, thereby improving reliability and reproducibility, but also provide direct visual evidence supporting the causal relationship between target membrane tension and the macrophage's choice between trogocytosis and phagocytosis.

We agree that pipette aspiration is a low-throughput technique (but a powerful one!). The dynamic experiments were intended to illustrate how changes in GUV membrane tension cause macrophages to switch from trogocytosis to phagocytosis. To provide additional data beyond the dynamic experiments, we conducted a more extensive set of pipette aspiration measurements (N=27) shown in Figure 3C, where we measured the membrane tension of GUVs that were either being trogocytosed or phagocytosed. In those experiments, we found clear membrane tension regimes for phagocytosis (>1 mN/m) and trogocytosis (<1 mN/m).

We appreciate the reviewer's suggestion to vary lipid compositions and use the fluorescent probe Flipper-TR. To test this approach, we first made GUVs with two different lipid compositions (POPC and POPC + 20% chol). While the elastic modulus of a membrane depends on lipid composition¹, membrane tension will be dependent on the stress applied (e.g., osmotic or pipette aspiration). Even for a single composition of GUV, it has been shown that the osmotic pressure oscillates for a single GUV compared to the bulk of GUVs due to random pore formation, even in an isotonic solution². At any given time, the membrane tension for a series of GUVs of the same composition, as measured by micropipette aspiration, can vary greatly, and this variation appears to blur any tension differences there may be between compositions (Response Figure 1A).

We next followed the suggestion to measure GUV membrane tension in our system using Flipper-TR, which reports changes in lipid order and relative values of membrane tension³. We imaged GUVs (79 mol% POPC, 20 mol% cholesterol, 1 mol% Biotin-PE, 0.5 mol% PEG2K-PE, and 1 mol% Flipper-TR) on a Stellaris laser scanning confocal microscope equipped with a FLIM system to measure lifetimes of Flipper-TR in GUVs. The GUVs were electroformed in sucrose osmotically matched to L-15 cell culture media. We observed that a visibly low-tension GUV can have a higher lifetime than a visibly high-tension GUV, while in another frame, a low-tension GUV has a lower lifetime than a high-tension GUV (Response Figure 1B). This discrepancy suggests that Flipper-TR may not be the ideal read-out of membrane tension in our system. It also highlights the difficulty of using changes in osmotic pressure to reliably and systematically vary the membrane tension of GUVs. Stochastic pore formation in response to changes in osmotic pressure makes it challenging to control the membrane tension of GUVs in bulk. The assay also requires co-incubation of GUVs with living cells, which limits the range of osmotic variations that can be investigated.

Response Figure 1. Investigation of the connection between membrane composition and membrane tension. **A.** GUVs made primarily of POPC lipids, or GUVs containing POPC + 20 mol% cholesterol, have a non-significant (via student's T test) difference in membrane tension as measured by micropipette aspiration. **B.** In one field of GUVs in L-15 media, a visibly taut vesicle has a shorter lifetime than a visibly floppy vesicle. In another field of the same sample, a visibly taut vesicle has a longer lifetime than a visibly floppy vesicle.

Comment 2: Similarly, the use of the Flipper-TR® membrane tension probe in the experiments presented in Figure 2 and Figure 4 would greatly enhance the robustness and persuasiveness of the data. Incorporating this probe into the experimental design would provide real-time, quantitative, and visual evidence of membrane tension differences, allowing for a more direct validation of the authors' central hypothesis that target cell tension regulates the choice between trogocytosis and phagocytosis. This addition is particularly important because, in both experiments, the current assessment of target cell tension relies on indirect methods or static measurements, which may not accurately reflect the actual membrane tension experienced by macrophages during the interaction. Without direct and dynamic tension measurements, it remains unclear whether the observed changes in macrophage behavior are truly driven by differences in membrane tension or potentially confounded by other factors, such as alterations in membrane composition, cell surface protein distribution, or fixation-induced artifacts. By applying membrane tension probe to the target cells used in Figure 2, the authors could verify whether differences in trogocytosis and phagocytosis frequencies correlate directly with measurable variations in membrane tension across different cell types. Similarly, incorporating the probe into the Figure 4 experiments—where target cell tension is increased through glutaraldehyde fixation—would confirm whether the observed suppression of trogocytosis and potential enhancement of phagocytosis correspond to genuine tension increases, rather than nonspecific effects of the fixation process.

We appreciate that our measurements of membrane tension in Figures 2 and 4 would benefit from a more direct and dynamic method. However, in addition to the membrane composition issues with Flipper-TR noted above, our manuscript argues that it is not membrane tension alone, but rather cortical tension (the combination of membrane tension and membrane-cortex attachment^{4,5}) that determines the macrophage response to a target cell. In GUVs, our micropipette measurements only probe membrane tension since there is no cortical cytoskeleton. However, in our cell experiments, we are probing cortical tension that includes membrane interactions with the cortical cytoskeleton. Since Flipper-TR does not capture membrane-cortex attachment in the same way as micropipette aspiration, it could miss an important contributor to the cell surface properties that appear to influence whether macrophages trogocytose or phagocytose.

Comment 3: While the manuscript focuses on how target cell membrane tension and antibody density influence the choice between trogocytosis and phagocytosis, it lacks a detailed discussion of the intracellular signaling mechanisms by which macrophages make this decision. Both processes require Fcγ receptor activation, but it remains unclear how differences in signal strength, duration, or downstream pathway activation lead to the selection of one response over the other.

We thank the reviewer for pointing out that we overlooked this important discussion point. To address this omission, we have included the following text in the discussion section:

Discussion Section, Paragraph 6: “But if, as we hypothesize, the receptors and downstream signaling molecules involved in the phagocytosis and trogocytosis pathways are the same, how might the response differ between the two processes? Both processes require a minimum surface antibody density ($\sim 10^2$ molecules/ μm^2) to initiate. If the target has low cortical tension, membrane fluctuations allow for a higher probability of binding events. Subsequently, the machinery required for macrophage protrusion and force generation (e.g., myosin motors and nucleation-promoting factors that activate Arp 2/3 to produce branched actin networks at the cell-cell interface) is recruited to a smaller area. When the cortical tension is low, the surface is more permissible to deformation and scission than a high-tension surface.”

Comment 4: The study focuses exclusively on antibody-mediated trogocytosis and phagocytosis. Have the authors considered whether the tension-regulation mechanism applies more broadly beyond antibody-dependent processes? For example, would similar trends be observed with other opsonins (e.g., complement components) or natural ligands such as phosphatidylserine on apoptotic cells? Investigating this could help clarify whether the proposed mechanism is specific to Fcγ receptor engagement or part of a more general mechanobiological response across different forms of target recognition.

We agree that this is a very interesting avenue of study. We have included a new experiment in the manuscript examining phagocytosis/trogocytosis of healthy cells with and without antibody, as well as apoptotic cells with and without antibody. Strikingly, cells that have undergone apoptosis are significantly stiffer (up to 3x) than their healthy counterparts⁶⁻⁸. We hypothesized

that this increase in stiffness of apoptotic cells would bias macrophages towards phagocytosis over trogocytosis.

To induce apoptosis in Jurkat T cells, we used raptinal, a potent activator of caspase-3 which induces apoptosis within 30 minutes of incubation and results in >80% cell death (as measured by propidium iodide) within 2 hours^{9,10}. In the presence of 10 μ M raptinal for 2 hours, we measured 85% of Jurkat T cells dead as marked by propidium iodide (in contrast to 1% of untreated Jurkat T cells). When we co-incubated LPS-stimulated macrophages with IgG-opsonized and raptinal-treated Jurkats, we saw a 3x increase in phagocytosis (and 15x decrease in trogocytosis) over healthy Jurkat T cells (Response Figure 2A left). Because apoptotic cells have additional phagocytic ‘eat me’ signals (e.g., phosphatidyl serine lipids flipped to the outer leaflet of the plasma membrane), we repeated the experiment without IgG opsonization (Response Figure 2A right). To measure trogocytosis without a fluorescent IgG, we biotinylated the target cells and then labeled the membrane with AlexaFluor-647 streptavidin (and dark anti-biotin in the opsonized case). As shown below, phagocytosis and trogocytosis in both raptinal-treated and untreated Jurkats are attenuated. However, phagocytosis is still higher for raptinal-treated Jurkats, and trogocytosis is higher for untreated Jurkats. Response Figure 2 has been added to the revised manuscript as Figure 4A.

Results “Increasing tension in cultured cells suppresses trogocytosis and increases phagocytosis.” Paragraph 1-2: Interestingly, cells that have undergone apoptosis are significantly stiffer (up to 3x) than their healthy counterparts^{26–28}. Macrophages are responsible for efferocytosis, or the phagocytic clearance of apoptotic cells. We hypothesized that the increase in stiffness upon apoptosis would bias macrophages towards phagocytosis over trogocytosis.

To induce apoptosis in Jurkat T cells, we used raptinal, a potent activator of caspase-3, which induces apoptosis within 30 minutes of incubation, resulting in >80% cell death (as measured by propidium iodide) within 2 hours^{29,30}. In the presence of 10 μ M raptinal for 2 hours, we measured 85% of Jurkat T cells dead as marked by propidium iodide (in contrast to 1% of untreated Jurkat T cells). After co-incubation with LPS-stimulated macrophages, we saw a 3x increase in phagocytosis (and a 15x decrease in trogocytosis) over healthy cells when Jurkats were opsonized with IgG (Fig. 4A, left). Because apoptotic cells have additional ‘eat me’ signals (e.g., phosphatidylserine lipids flipped to the outer leaflet of the plasma membrane), we repeated the experiment without IgG opsonization (Fig. 4A, right). To measure trogocytosis without a fluorescent IgG, we biotinylated the target cells and then labeled the membrane with AlexaFluor 647 streptavidin (and dark anti-biotin for opsonized cells). In the right panel of Fig. 4A, phagocytosis and trogocytosis in both raptinal-treated and untreated Jurkats are attenuated; however, phagocytosis is still higher for raptinal-treated Jurkats.”

Response Figure 2. Increasing cell tension through apoptosis suppresses trogocytosis and increases phagocytosis. Biotinylated Jurkat T cells were labeled with AlexaFluor647 streptavidin and opsonized with anti-biotin IgG at 1385 ± 189 molecules/ μm^2 (left) or unopsonized (right) and treated with 10 μM raptinal, or untreated, and co-incubated with LPS-stimulated macrophages for 2 hrs. Statistical significance was computed via student's T test.

Reviewer #2 (Remarks to the Author):

In this paper, the authors convincingly demonstrate that the process of trogocytosis, the “nibbling” of the plasma membrane by phagocytes (such as macrophages) is strongly influenced by the cell surface tension of the target cell membrane. Sophisticated methods are employed to reach this conclusion, with convincing results. The authors conclude that this the major factor in the “decision” to either trogocytosis or phagocytose a target cell. I have a number of concerns regarding this conclusion and the physiological relevance of their findings.

Comment 1: One major concern with their conclusion is that although the role of target cell surface tension in the efficiency of trogocytosis is clearly demonstrated, no evidence is provided that cells with high surface tension are preferentially phagocytosed. In their system, phagocytosis is a rare event, and increasing surface tension with low concentrations of glutaraldehyde did not appear to increase this (certainly not anywhere near the levels of trogocytosis they observe in untreated cells). While their conclusions regarding trogocytosis seem solid to this reviewer, conclusions regarding phagocytosis are not supported by data.

We thank the reviewer for noting this omission. We were focused on the new finding about the tension dependence of trogocytosis and did not highlight the changes in phagocytosis that occur upon an increase in cell tension. We have revised Figure 4E in the main text to present the increase in phagocytosis that occurs with gentle glutaraldehyde fixation, in addition to the decrease in trogocytosis that occurs.

To further address the reviewer's question about phagocytosis, we have added a subfigure with new experiments measuring phagocytosis and trogocytosis of Jurkat T cells, HL60s, and M2 cells with LPS-stimulated RAW 264.7 macrophages (Response Figure 3A-B; Figure 4C). M2 cells are deficient in filamin A, a crosslinker of F-actin at the cortex, and they are known to be softer than similar cells expressing Filamin A^{11,12}. In comparison to Jurkat T cells and HL60s, M2 cells opsonized at the same surface density of anti-CD47 are trogocytosed significantly more (Response Figure 3B), consistent with our findings that lower cortical tension results in greater trogocytosis.

To investigate the amount of phagocytosis that can occur, we co-incubated the macrophages and target cells for 2 hours. In this case, we see that trogocytosis is clearly higher with Jurkats than with HL60s, and that LPS-stimulated macrophages phagocytose HL60s at a greater percentage than they trogocytose them. While macrophages do not phagocytose HL60s more than Jurkats, the *ratio* of phagocytosis to trogocytosis is 2x greater for Jurkats. We have added Response Figure 3 as Fig. 4B-C in the revised manuscript and updated the text to reflect our findings:

Results section, “Increasing tension in cultured cells suppresses trogocytosis and increases phagocytosis.” Paragraph 3: “To further investigate the role of cortical tension in trogocytosis of living cells, we co-incubated LPS-stimulated macrophages with a melanoma-derived cell type, M2. M2 cells are deficient in filamin A, a crosslinker of F-actin at the cortex. The cortex of M2 cells is weakened and, as a result, they are measurably softer than similar cells expressing Filamin-A^{31,32}. In comparison to Jurkat T cells and HL60s, M2 cells opsonized at the same surface density of anti-CD47 are trogocytosed significantly more (Fig. 4C), consistent with our findings that lower cortical tension results in greater trogocytosis.”

Results section, “Trogocytosis differs for target cells depending on their cortical tension”, Paragraph 1: “Notably, this is also the case for LPS-stimulated macrophages, especially for the filamin-deficient soft melanoma cell line, M2 (Fig. 4C). Strikingly, while LPS-stimulated macrophages phagocytose HL60s and Jurkats at a statistically similar amount, the ratio of trogocytosis to phagocytosis is significantly higher for Jurkats than for HL60s (~2x).”

Response Figure 3. Filamin-A-deficient cancer cells with lower cortical tension are trogocytosed. **A.** Confocal micrographs of LPS-stimulated RAW 264.7 cells labeled with CellTracker Green CMFDA trogocytosing M2 cells opsonized with AlexaFluor647 anti-CD47 IgG. White arrows indicate trogocytic bites. **B.** Percent of LPS-stimulated macrophages that have phagocytosed (red bars) or trogocytosed (blue bars) anti-CD47-opsionized HL60s, Jurkat T cells,

and M2 cells. The antibody surface density was matched between the three cell types at 163 ± 8 molecules/ μm^2 . Statistical significance was determined via a student's T-test.

Comment 2: While trogocytosis has been described as having a role in immune evasion in pathogenic amoeba and in nematodes, it is unclear if trogocytosis plays essential roles in mammalian physiology. No attempt to identify a physiological role for the mechanism they describe is offered. While treatment with glutaraldehyde is useful for their analysis, what physiological conditions influence surface tension sufficiently to affect trogocytosis? Further, when trogocytosis is inhibited (but phagocytosis is presumably intact) what are the consequences? For example, we can speculate that trogocytosis of anti-CD47-treated cells might limit phagocytic killing of these cells; does surface tension account for differences in the ability of anti-CD47 (or SIRPa) treatment to result in killing of some cell lines but not other? (Note, I am not insisting on this experiment but offer it as an example). There are other claims as to the function of trogocytosis in mammalian cells. In the absence of a physiological consequence of trogocytosis, we are left with only a mechanism for a mildly interesting phenomenon without context.

In the manuscript, we provide several recent examples of studies in which the physiological relevance of trogocytosis is established. For example, we reference Gao, X. et al., *Science* 2024, which demonstrates a physiologically relevant role for trogocytosis in bone marrow-derived macrophages (BMDMs) and hematopoietic stem cells (HSCs) in mice and in humans. In that paper, trogocytosis between BMDMs and HSCs results in a subpopulation of HSCs that display BMDM proteins, such as F4/80. This subpopulation of HSCs remains in the bone marrow and never egresses to the bloodstream, unlike the other HSCs in the marrow¹³.

Since submitting our paper, several other papers have been published or submitted to BiorXiv, pointing to the clear relevance of trogocytosis in contexts as diverse as tissue regulation and stem cell development^{14,15}, to clearance of parasitic worms¹⁶ and solid tumors^{17,18}. We have included references to each of these publications in the introduction section of the revised manuscript. These references point to an emerging physiological role for trogocytosis and emphasize the need for a more mechanistic and cell biological understanding of the phenomenon, which our manuscript aims to provide.

Introduction Paragraph 2 “For example, human and mouse macrophages have been observed to nibble hematopoietic stem cells, marking them for retention in the bone marrow⁷. In zebrafish, hematopoietic stem cells are also selectively nibbled by macrophages, marking them for clonal hematopoiesis⁸. Similarly, tissue resident macrophages of the brain, microglia, selectively nibble synapses of neurons to prune connections in the developing mouse brain⁹. Strikingly, macrophages have also been shown to trogocytose targets that are too large to phagocytose, such as the parasitic worm, *Schistosoma*¹⁰, or large tumors¹¹. ”

Lastly, the first author attended the recent Gordon Research Conference on Phagocytes, where she gave a talk on this work. There was interest from clinical researchers as well as basic researchers, and we are following up now on potential collaborations to use our methods to investigate new systems.

Comment 3: Does trogocytosis of living cells depend on inhibition of CD47-SIRP α interaction? The biotin experiment doesn't answer this question, since CD47 is biotinylated in this situation. Are other antibodies to cell surface proteins engulfed in the absence of blocking CD47? If so, does surface tension play a role in such situations? This is important, since it is assumed that surface tension controls the ability of phagocytes to "nibble" small pieces of membrane, but if the effects depend on blocking CD47-SIRP α , then it remains possible that surface tension influences this blockade.

We thank the reviewer for the question. To directly address it, we co-incubated RAWs with Raji B cells opsonized with anti-CD19, a surface protein abundant on B cells. The anti-CD19 antibodies do not block CD47, a potent 'don't eat me' signal, and will therefore not hinder binding of macrophage SIRP α to target CD47. According to our measurements, the average surface density of anti-CD19 on Raji B cells was 212 molecules/ μm^2 compared to an average surface density of 200 molecules/ μm^2 anti-CD47 (Fig. 2B of the revised manuscript). Raji B cells opsonized with anti-CD19 show ~4x more trogocytosis than phagocytosis (Response Figure 4), which is comparable to the results anti-CD47, albeit with an overall 2x lesser magnitude. We expect that the lesser magnitude of both trogocytosis and phagocytosis can be attributed to the lack of a CD47-SIRP α blocking effect from the anti-CD47 antibodies. Response Figure 4 below is included in the supplement as Fig. S4 of the revised manuscript. We have also added the following text into the main manuscript:

*Results Section, "Trogocytosis differs for target cells depending on their cortical tension."
Paragraph 2: "When antibody surface density is matched, we observe that trogocytic efficiency is highest in Jurkats, while phagocytic efficiency is highest in HL60s. To ensure that our results are not simply from blocking the 'don't eat me' interaction of target CD47 with macrophage SIRP α (due to anti-CD47 or biotinylation of CD47), we co-incubated macrophages with Raji B cells opsonized with anti-CD19. We observed that Raji B cells opsonized with anti-CD19 show ~4x more trogocytosis than phagocytosis, which is comparable to the results in Fig. 2B, albeit with an overall 2x lesser magnitude (Fig. S4)."*

Response Figure 4. Phagocytosis and trogocytosis of Raji B cells opsonized with AlexaFluor647 anti-CD19 at a surface density of 212 molecules/ μm^2 .

Comment 4: The results in Figure S2 are clear but unconvincing. Since trogocytosis is measured here with uptake of labeled antibody, the ability of anti-FcR to “block” trogocytosis is not surprising. Does inhibition of FcR prevent uptake of biotinylated proteins in this setting?

We thank the reviewer for the question. To address this point, we carried out a new experiment that demonstrates a lack of membrane transfer from target Jurkat T cells to RAW 264.7 macrophages when Jurkat cells are not opsonized with antibody. We accomplish this by biotinylating the surface proteins of Jurkat T cells and adding AlexaFluor647 streptavidin to label the membrane. By tracking the transfer of fluorescent streptavidin, we show that trogocytosis occurs only minimally in the absence of opsonization (Response Figure 5). We’ve added this data to the supplement as part of Fig. S2 of the revised manuscript.

Results “Macrophages phagocytose and trogocytose cultured cells.” Paragraph 6: “Similarly, we observed that biotinylated, but non-opsonized, target cells with AlexaFluor647 streptavidin-labeled membranes are not trogocytosed or phagocytosed, confirming that trogocytosis is dependent on FcR engagement (Fig. S2C).”

We also show that trogocytosis is not the result of the uptake of extracellular vesicles by incubating RAWs with the supernatant from labeled Jurkats. Notably, when macrophages are stimulated with LPS, there is higher background trogocytosis, which is presumably due to the increased activity of LPS-stimulated macrophages.

Response Figure 5. Macrophage trogocytosis depends on FcR engagement. Trogocytosis of AlexaFluor647 anti-biotin IgG opsonized Jurkat T cells by LPS-stimulated macrophages are unstimulated macrophages. When Jurkat T cells are uncoated but labeled with AlexaFluor647 streptavidin, trogocytosis is decreased. Similarly, when the supernatant from opsonized Jurkats is added to macrophages, there is minimal uptake of AlexaFluor647 streptavidin-labeled membrane.

5. Figure 2. The conclusion that HL60 cells “are phagocytosed more than the other lines” is not clear—are these small differences significant (and sufficiently powered)?

We agree that this statement was not clear and have added supplemental experiments with LPS-stimulated RAWs to address this point. In this case, HL60s are not necessarily phagocytosed

more than the other cell lines, but Jurkats are clearly trogocytosed more than HL60s. We have updated the text accordingly (see also response to Reviewer 2 Comment 1).

6. *Figure 4. As mentioned in #1. Increasing cell surface tension with glutaraldehyde clearly inhibits trogocytosis, however the prediction would be that it increases phagocytosis. Is this true? Only trogocytosis was measured. If phagocytosis does not appreciably increase as trogocytosis decreases (and I have to assume that if this were the case, it would have been shown), then the study does not describe the difference between these two processes, but rather a requirement for efficient trogocytosis itself.*

We have updated the manuscript to include data on phagocytosis from the experiments with glutaraldehyde. Consistent with our model, phagocytosis increases significantly at higher concentrations of glutaraldehyde, while trogocytosis significantly decreases (Fig. 4E).

Reviewer #3 (Remarks to the Author):

In their manuscript, Cornell et al., try to understand why a different macrophage lines including primary macrophages would be directed to trigger trogocytosis versus phagocytosis of opsinized target cells. The propensity of different target cells to undergo predominantly trogocytosis or phagocytosis presented some clues for the authors, since they had observed that the predominantly trogocytosed, Jurkat and Raji cells had a lower cortical tension than the more phagocytic-prone HL60 cells. Furthermore at similar surface antibody levels, Jurkat and Raji cells appear to be show similar tendencies to trigger trogocytosis in RAW-macrophages. They test this hypothesis by using biotinylated cell surfaces with anti-biotin antibodies, where they find that at matched antibody concentrations they see similar fractions of trogocytosis versus phagocytosis (T vs P), and by modulating cortical tension using low concentrations of glutaraldehyde they are able to alter the Tvs P fractions. Finally they provide a simple mechanical model based on force balance that they claim provides an explanation for the different regimes of their observations. While the findings and explanation of the study are compelling, there are several issues that need addressing before their conclusions may be fully justified.

Comment 1: In the context of the tripartite relationship between cortical tension and cell surface adhesion strength due to the interaction between antibodies and macrophage Fc-receptors, and T vs P ratios of target cells there is a lack of clarity (or a lack of the exploration) of the quantitative possibilities offered by the experiments in cells and the artificial GUVs.

The authors follow their observation that higher tension reduces the fraction of macrophages that undergo trogocytosis and increases the fraction that exhibit phagocytosis by showing that an increase in the adhesion strength (sites) can overcome this discrepancy. They follow this up with an exploration of T vs P in NHS-biotin conjugated cells, labelled with the same concentration of anti-biotin antibody. They find that anti-biotin-labelled cells undergo more trogocytosis compared to matched anti-CD67 labelled cells, but lower phagocytosis in each case (compare Fig 2B with C). It is likely that the number of anti-biotin antibodies per μm^2 in Fig 2C is higher than the anti-CD67 antibodies in 2B. It would be important to provide a quantitative analysis of the amount of

surface anti-biotin antibody in each case to provide an explanation for the difference between these two results. A detailed measurement of the number of anti-biotin antibodies per μm^2 , and a titration of this is required.

We appreciate the reviewer's suggestion that we provide a quantitative analysis of the amount of surface antibody in our experiments using anti-biotin antibodies. In every experiment, we used fluorescent calibration beads to compute the number of antibodies present on the surface of cells based on the fluorescent signal from the flow cytometer and the average surface area of each cell type. We have updated the manuscript to include the average surface density of antibody in the figure captions for figures that do not explicitly show it. We highlight the important role of antibody surface density in the revised manuscript Fig. S7, where we titrate antibody density and show the corresponding change in trogocytosis.

Comment 2: They subsequently use biotin-lipid labelled GUVs opsinized by anti-biotin antibodies to show that they can skew the fraction of macrophages that exhibit of T vs P macrophages by altering the tension in the vesicle. This is followed up by altering the tension in HL60 and Jurkat cells using glutaraldehyde; this is at best an unusual way of altering cortical tension in these cells. It has not been well characterized except in Red blood cells (ref 19), which are enucleated cells lacking a similar cortical actin network as the nucleated macrophages. An alternative method to address changes in cortical tension would be required to provide a rigorous test of this idea. They authors could use non-muscle myosin2 inhibitors or activators that may also provide similar changes in cortical tension and explore the consequences thereof.

We agree with the reviewer that glutaraldehyde is a non-physiological (but convenient) way to perturb cell cortical tension. We chose this method for two main reasons. First, our phagocytosis/trogocytosis assay is a co-incubation experiment. To have the desired effect on the cytoskeleton, drugs like non-muscle myosin2 inhibitors and other actin/motor/microtubule drugs need to be in the solution with the cells. The half-life of many of these drugs is shorter than the duration of a co-incubation experiment (i.e. blebbistatin's half-life ~ 20 min). If we were to leave the target cells bathing in the cytoskeletal drug, we would also drug the macrophages in the co-incubation experiment. The gentle glutaraldehyde treatment allowed us to pre-treat the target cells, wash them completely, and then co-incubate them with the macrophages. Second, we wanted a way to reliably titrate the cortical stiffness of cells. By titrating the concentration of glutaraldehyde, which is known to rigidify cells, we were able to effectively titrate the cortical stiffness of target cells.

However, we agree that a more physiological experiment would help to support our findings. To that end, we carried out two new experiments shown in Response Figures 2 and 3 (and included in Fig. 4A-C of the revised manuscript). Both new experiments confirmed that lower cortical tension correlates with increased trogocytosis:

1. In the first experiment, we co-incubated macrophages with Jurkat T cells treated with raptinal, a potent initiator of apoptosis. Cells that have undergone apoptosis are significantly stiffer (up to 3x) than their healthy counterparts⁶⁻⁸. In Response Figure 2, we show that apoptotic cells are phagocytosed significantly more than healthy cells.

The reverse is also true; healthy cells are trogocytosed significantly more than apoptotic cells.

2. In the second experiment, we co-incubated macrophages with a cell line derived from human melanoma with a filamin-A deficiency (M2) in which cells constitutively bleb and the membrane tension of blebs has been measured to be $\sim 2x$ less than membrane attached to the cortex¹¹. Filamin-A deficiency leads to an overall weakening of the cortex, which decreases cortical tension¹². With both decreased cortical tension and frequent membrane detachment from the cortex, we predicted that M2 cells would be trogocytosed preferentially over phagocytosis, which we show in Response Figure 3.

Comment 3: In the abstract and the discussion, it may not be correct to say that there is no molecular switch required for the two processes- it is likely that the mechanical context induces a switch in cell behavior (see the movies supplied) that cause a change in cell engulfment behavior.

We thank the reviewer for this correction, and we have changed the text in the abstract and discussion accordingly.

Comment 4: Finally, they introduce a theoretical framework that links the tension-mediated length scale, R_{min} (estimated at $0.1-1 \mu m$), to the decision-making process of macrophages, whether to engage in trogocytosis versus phagocytosis. They also predict that the critical antibody density, below which active stresses are too small to trigger significant trogocytosis, must be proportional to the target cell tension, and they support this prediction with experiments (but see above). By reducing the complex interplay of cellular forces to a force balance between active stress and cortical tension, the authors provide a clear physical picture of how mechanical properties dictate immune cell behaviour in an intuitively simple framework. The predictions, such as the critical antibody density threshold and the expected deformation (or bite) sizes, are consistent with experimental measurements using both cultured cells and GUVs.

We thank the reviewer for their positive assessment of the simple model we present to explain the tension dependence of trogocytosis. We also appreciate the suggestions for improvement. We will address each sub-comment individually.

Comment 4.1: While the predicted length scales ($0.1-1 \mu m$) align well with the observed trogocytic bite sizes, a more precise quantitative comparison could enhance the model's assumptions and conclusions. The current analysis does not account for higher-order shape deformations and appears to be based on smooth shape changes ($R_{min} \gg h$, where h is the dimple height). It could be expanded to account for higher-order deformations.

We agree that it is possible to include higher-order deformations in our model. We note that higher order, thermally driven deformations that lead to membrane fluctuations below the length scale $\sqrt{\kappa/\gamma_t}$ are highly unfavorable. In our system for the lower limit of tensions considered, $\gamma_t \sim 10^{-1}$ mN/m, and considering a membrane bending stiffness of $\kappa \approx 400 K_B T$, this length scale is $\sim 0.1 \mu m$. Since this length scale is below the scale of deformations predicted due to a

balance of active stresses and cortical tension, we have not included these higher order shape modes in our current scaling analysis. However, in a more detailed model where one is interested in the time dynamics of a trogocytic deformation/bite, these higher order shape deformations may play an important role in setting the kinetics of interfacial changes. We are working on such a spatio-temporally resolved statistical model, but we believe that it is outside the scope of this current work.

Comment 4.2: The definitions of phagocytosis (" $>50\%$ surface coverage by macrophage extensions") and trogocytosis (presence of trogocytic "bites") could be clarified. For example, it is unclear why a threshold of 50% is chosen. Does it imply that even partial engulfment is classified as phagocytosis? (as seen in Fig. 3C). Or that $< 50\%$ engulfment always proceeds towards trogocytosis?

We chose the definition for phagocytosis as $>50\%$ engulfment and trogocytosis as $<50\%$ as a way of binning the data in the GUV micropipette experiments. Since we are unable to measure the tension of a fully engulfed vesicle within a macrophage, we chose an arbitrary percentage of engulfment with the assumption that engulfment greater than 50% would lead to completion of phagocytosis.

Comment 4.3: The current framework addresses a steady-state balance of forces. Incorporating time-dependent aspects might further refine its predictive power since it appears that the macrophages dynamically respond to changes in cortical tension of the target cell.

We fully agree with the reviewer that a time-resolved model would be interesting and possibly lead to new qualitative and quantitative insights about the phenomenon of trogocytosis. However, such a model involves a detailed accounting of the reaction-diffusion of species at the interface, including receptors, antibody and bystander proteins, and a coupling between these and the mechanical deformation of the interface. These take the form of coupled PDEs for the concentrations of these species as well as the height and gap at the interface. We are currently working on such a model, but we believe it is beyond the scope of this current work.

Comment 4.4: If this is a general mechanical scaling law- in Fig 5C, the $p(\text{crit})$ should scale with antibody density and tension- why do HL60 and Jurkat cells have such a different scaling relationship?

We thank the reviewer for this insightful comment. While the scaling law is general, there are important non-linearities in the scaling between the active stresses developed at the interface and the antibody density (as the reviewer rightly points out in another comment) that likely lead to these observed differences in scaling between HL60s and Jurkats. Differences in this scaling across different cell lines might result from differences in the surface proteome in terms of the presence of tall bystander proteins, differences in mobility of the antibody-tagged proteins, and differences in membrane properties not captured by tension. Since it is the antibody density and tension that are experimentally measured, we do expect a positive correlation between the critical

antibody density and cortical tension, as observed and shown in Fig. 5C of the revised manuscript. We have added the following text to address this:

Results section, “A mechanical model captures the relationship between cell tensions and antibody surface density in trogocytosis” Paragraph 4: “We assume that while this scaling law is general, the difference in slope between HL60s and Jurkats is likely due to important non-linearities in the scaling between the active stresses developed at the interface and the antibody density. By taking this approximation, we do not consider the possible contributions from dynamic remodeling of the cytoskeleton, spatial heterogeneities due to lipid rafts, or receptor clustering and cooperative effects. Since we experimentally measure antibody density and cortical tension, we do, however, expect a positive correlation between the critical antibody density and cortical tension, as shown in Fig. 5C.

Comment 4.5: The reduction of antibody density to the active stresses that act at the cell interface by assuming a linear scaling of active stresses with antibody density is at best an approximation. Discussing potential non-linearities, such as dynamic remodeling of the cytoskeleton, spatial heterogeneities, receptor clustering, or cooperative effects, would help contextualize the model’s limitations.

We agree that the scaling between antibody density and magnitude of active stresses is important, and our assumed scaling, while nonlinear, is still an approximation. New experimental measurements coupled with theoretical models are likely required to elucidate this scaling in greater detail. As the reviewer points out, this scaling contains interesting nonlinearities due to effects they enumerate, of which receptor clustering and cooperativity are likely important, given their importance has already been demonstrated in the context of phagocytosis^{19,20}. We have included text that discusses these other possible non-linearities and their contribution to the effect.

Results section, “A mechanical model captures the relationship between cell tensions and antibody surface density in trogocytosis” Paragraph 4: “We assume that while this scaling law is general, the difference in slope between HL60s and Jurkats is likely due to important non-linearities in the scaling between the active stresses developed at the interface and the antibody density. By taking this approximation, we do not consider the possible contributions from dynamic remodeling of the cytoskeleton, spatial heterogeneities due to lipid rafts, or receptor clustering and cooperative effects. Since we experimentally measure antibody density and cortical tension, we do expect a positive correlation between the critical antibody density and cortical tension, as shown in Fig. 5C.

References

1. Saedimazine, M., Montanino, A., Kleiven, S. & Villa, A. Role of lipid composition on the structural and mechanical features of axonal membranes: a molecular simulation study. *Sci Rep* **9**, (2019).
2. Karatekin, E. *et al.* Cascades of Transient Pores in Giant Vesicles: Line Tension and Transport. *Biophysical Journal* vol. 84 (2003).
3. Colom, A. *et al.* A fluorescent membrane tension probe. *Nat Chem* **10**, 1118–1125 (2018).
4. Sitarska, E. & Diz-Muñoz, A. Pay attention to membrane tension: Mechanobiology of the cell surface. *Current Opinion in Cell Biology* vol. 66 11–18 Preprint at <https://doi.org/10.1016/j.ceb.2020.04.001> (2020).
5. Chugh, P. *et al.* Actin cortex architecture regulates cell surface tension. *Nat Cell Biol* **19**, 689–697 (2017).
6. Lulevich, V., Zink, T., Chen, H. Y., Liu, F. T. & Liu, G. Y. Cell mechanics using atomic force microscopy-based single-cell compression. *Langmuir* **22**, 8151–8155 (2006).
7. Lam, W. A., Rosenbluth, M. J. & Fletcher, D. A. Chemotherapy exposure increases leukemia cell stiffness. (2007) doi:10.1182/blood-2006-08.
8. Islam, M. *et al.* Microfluidic Sorting of Cells by Viability Based on Differences in Cell Stiffness. *Sci Rep* **7**, (2017).
9. Palchadhuri, R. *et al.* A Small Molecule that Induces Intrinsic Pathway Apoptosis with Unparalleled Speed. *Cell Rep* **13**, 2027–2036 (2015).
10. Smith, A. J. & Hergenrother, P. J. Raptinal: a powerful tool for rapid induction of apoptotic cell death. *Cell Death Discovery* vol. 10 Preprint at <https://doi.org/10.1038/s41420-024-02120-1> (2024).
11. Dai, J. & Sheetz, M. P. Membrane tether formation from blebbing cells. *Biophys J* **77**, 3363–3370 (1999).
12. Kasza, K. E. *et al.* Filamin A is essential for active cell stiffening but not passive stiffening under external force. *Biophys J* **96**, 4326–4335 (2009).
13. Gao, X. *et al.* Regulation of the hematopoietic stem cell pool by C-Kit-associated trogocytosis. *Science* **385**, eadp2065 (2024).
14. Rodrigues, C. P. *et al.* Transcripts of repetitive DNA elements signal to block phagocytosis of hematopoietic stem cells. doi:10.1126/science.eadn1629.
15. Volk, R. F. *et al.* Macrophages redeploy functional cancer cell surface proteins following phagocytosis. Preprint at <https://doi.org/10.1101/2024.09.23.613776> (2024).
16. Shen, J. *et al.* Macrophage-mediated trogocytosis contributes to destroying human schistosomes in a non-susceptible rodent host, *Microtus fortis*. *Cell Discov* **9**, (2023).
17. Barbera, S. *et al.* Trogocytosis of Chimeric Antigen Receptors between T Cells Is Regulated by Their Transmembrane Domains. *Sci. Immunol* vol. 10 <https://www.science.org> (2025).
18. Rollins, K., Fiaz, S. & Morrissey, M. Target cell adhesion limits macrophage phagocytosis and promotes trogocytosis. Preprint at <https://doi.org/10.1101/2025.02.06.636906> (2025).

19. Kern, N., Dong, R., Douglas, S. M., Vale, R. D. & Morrissey, M. A. Tight nanoscale clustering of $\text{Fc}\gamma$ receptors using dna origami promotes phagocytosis. *Elife* **10**, 1–29 (2021).
20. Jo, S. *et al.* IgG Surface Mobility Promotes Antibody Dependent Cellular Phagocytosis by Syk and Arp2/3 Mediated Reorganization of Macrophage $\text{Fc}\gamma$ Rs. *SSRN Electronic Journal* 1–24 (2021) doi:10.2139/ssrn.3908787.

Response to Reviewers

“Target cell cortical tension regulates macrophage trogocytosis”

Caitlin E. Cornell et al

We thank all the reviewers for their thoughtful analysis and helpful comments on our revised manuscript. Please see below for point-by-point responses to the reviewer’s comments. Additions to the main text have been highlighted in blue.

Reviewer #1 (Remarks to the Author):

The authors' diligent efforts have transformed what I initially considered a preliminary study into a robust and impactful piece of work. The conclusions are now well-supported by a combination of clever model system experiments, new data from physiologically relevant contexts, and sound theoretical reasoning. The manuscript is significantly strengthened, and I am happy to recommend it for publication.

I have one final, minor suggestion:

If the data are readily available, the authors could consider adding a panel to the supplementary materials (Figure S7) that overlays the density-response curves for different opsonins (e.g., anti-CD47 vs. anti-biotin) on the same axes. This would provide a more intuitive visual for their conclusion that the response is insensitive to antigen identity and is instead dominated by density. However, I would not recommend that new experiments be performed for this purpose if the data is not already on hand.

We thank the reviewer and agree with the suggestion. The density-response curves in Figure S7 already include an overlay of both anti-CD47 and anti-biotin opsonins, and we apologize for not making this clear in the figure legend. We overlaid both opsonins because we achieve higher antibody densities with anti-biotin. We have added text to the figure legend to clarify this.

Reviewer #2 (Remarks to the Author):

The authors have done a nice job of addressing my concerns, and I have no further comments.

Reviewer #3 (Remarks to the Author):

In their revised manuscript, the authors have addressed most of the issues raised. Overall the manuscript is strengthened by their new data. In particular, the addition of new experimental data on the phagocytosis versus trogocytosis balance of apoptotic cells and filamin-A deficient cells with consistent consistent effects on the ratio of the two processes. Together with the inclusion of phagocytosis data with glutaraldehyde treatment, and the opsonization tests all strengthen the experimental case. The point that seems less fully resolved is the request for more direct, dynamic measurements of cortical tension. The authors provide a reasonable explanation of the limitations of probes such as Flipper-TR and argue that cortical tension, not membrane tension alone, is the relevant parameter.

Some caveats still remain and these may be addressed in the discussion.

i) In the quantitative comparison of predicted vs. observed bite sizes: we had referred to ‘higher-order deformations,’ ie. non-smooth, nonlinear shape changes rather than thermally driven higher modes. Therefore the question still remains whether the scaling analysis, which assumes smooth deformations ($R_{min} \gg h$), is still valid in regimes where shapes are more complex (e.g., sharp indentations or non-smooth geometries). This seems relevant, since macrophage bites may not always conform to a smooth-dimple approximation.

We thank the reviewer for this thoughtful comment. Under the effect of active stresses, the smoothness of the membrane is controlled by the resistance of the membrane to stretching or bending. For very sharp deformations, we expect the contribution from bending to dominate.

Thus, the bending stiffness of the membrane sets a lower limit $R_{bend} \sim \left[\frac{\kappa}{\sigma_{normal}(\rho_{AB})} \right]^{1/3}$ (which is ~ 250 nm for $\kappa = 400 k_B T$ and $\sigma_{normal} \approx 100$ Pa) on the radius of curvature of any membrane deformation. Further, the relatively weak scaling with σ_{normal} implies that to see “sharper” indentations below the 100 nm length scale would take a local active stress beyond 1kPa, which is unphysiologically high (see Response Fig. 1). In practice, we believe the membrane will not show sharp indentations below this ~ 100 nm length scale.

Response Fig. 1: Membrane indentation radius scales inversely with active stress generated by the macrophage. For a physiologically relevant range of active stresses (50-150 Pa), R_{bend} is large enough to assume the membrane is smooth.

To address the reviewer's concern about our smooth-dimple approximation for macrophage bites, we have added the following text to the manuscript (Page 8, Section *A mechanical model captures the relationship between cell tensions and antibody surface density in trogocytosis* Paragraph 4):

“Our scaling analysis assumes smooth deformations in the membrane, which is controlled by the resistance of the membrane to stretching or bending. For sharp deformations, we expect the contribution from bending to dominate. However, to see sharp indentations below a radius of 0.1 μm would require local active stresses beyond what are expected physiologically (see Supplementary Note, equation (5)).”

ii) Definitions of phagocytosis vs. trogocytosis in the GUVs: The authors have clarified that they define phagocytosis as >50% engulfment of the target and trogocytosis as <50%, motivated by the constraints of their assay on GUVs. This operational threshold is somewhat arbitrary. Because this definition underpins the classification of all events, it has important consequences for the interpretation of the findings. For instance, if partial engulfment (>50%) do not always proceed to full phagocytosis, or if some <50% engulfment represent stalled phagocytic attempts, then the categorization may not strictly reflect two distinct biological outcomes. It would be important to how sensitive their conclusions are to this threshold, and whether their key takeaways, particularly the phase diagram, remain robust if the cutoff were shifted from the 50% arbitrary choice.

We appreciate the reviewer's concern about our 50% engulfment threshold for distinguishing phagocytosis from trogocytosis. We were constrained by the practical limitations of the GUV micropipette assay because the membrane of a fully engulfed GUV is not accessible by the micropipette. Our time-lapse imaging of macrophage-GUV interactions has consistently shown that when macrophages formed phagocytic cups covering more than 50% of the GUV's surface, these GUVs were predominantly engulfed afterward, supporting this metric as a useful indicator of phagocytosis. Our observation is consistent with observations from Settle et al. with macrophage engulfment of polymer spheres (1). We acknowledge that adjusting this threshold might change the tension limit between trogocytosis and phagocytosis. However, it's important to note that our phase diagram in Figure 5 is qualitative and is not based on data from Figure 3. Therefore, the 50% threshold used to distinguish between trogocytosis and phagocytosis does not affect the overall phase diagram.

To further explain our choice of threshold, we have added the following text to the manuscript (Page 6, Section *Membrane tension of a cell mimic drives trogocytosis over phagocytosis* Paragraph 4):

“We chose >50% wrapping as the threshold for phagocytosis since the micropipette occupies one side of the GUV and since we and others²⁶ have found that >50% coverage leads to full engulfment >75% of the time.”

iii) Scaling law: While the authors acknowledge that HL60 and Jurkat cells follow different apparent slopes and attribute this to nonlinearities, this response feels incomplete given that the

scaling law forms a central pillar of their interpretation. It would be helpful for the authors to discuss more concretely whether the scaling is expected to be universal or whether it should be viewed as an approximate trend with cell-type-specific deviations. If the latter is the case, it would be best not to refer to this as a scaling law, since this will change with cell-type and cortical tension of target.

We thank the reviewer for this comment. We agree, as the reviewer points out, there will likely be cell-type-specific differences that occur because the scaling requires one to know the active stress at the interface σ_{normal} , which is, in turn, a function of the local antibody density. This relationship is unknown but is expected to be positively correlated over a reasonable antibody density range (i.e., it's reasonable that a higher local antibody density leads to a higher active stress magnitude since more receptors are engaged). Importantly, this relationship will likely have cell-type-specific differences, as they maintain different surface proteins, leading to differences in the surface biophysical properties. However, antibody density is the experimental measure in our system; hence, the different scaling of critical tension with antibody density for disparate cells is to be expected. In the future, one could use techniques such as macrophage engagement with deformable beads to measure the relationship between active stresses and the local antibody density.

We have made modifications in the main text to make this clearer, including replacing all instances of “scaling law” with “scaling relationship” (Page 9, Section *A mechanical model captures the relationship between cell tensions and antibody surface density in trogocytosis*):

“We assume that while this scaling relationship is general, the difference in slope between HL60s and Jurkats is likely due to important differences in how the active stresses developed at the interface are related to the local antibody density. In this simple scaling, we do not consider the possible contributions from dynamic remodeling of the cytoskeleton, spatial heterogeneities due to lipid rafts, or receptor clustering and cooperative effects, which will likely have cell-type-specific differences due to distinct surface proteomes and properties. For the parameters we measured, antibody density and cortical tension, our modeling is consistent with the positive correlation between the critical antibody density and cortical tension that we observe (Fig. 5C).”

iv) The authors are encouraged to make the limitations of their interpretation more explicit so that readers can better appreciate the scope of their conclusions.

We agree with the reviewer and have made the modifications as outlined above in the main text to clarify this point.

References

1. Settle, A.H., Winer, B.Y., de Jesus, M.M., Seeman, L., Wang, Z., Chan, E., Romin, Y., Li, Z., Miele, M.M., Hendrickson, R.C., Vorselen, D., Perry, J.S.A., and Huse, M. $\beta 2$ integrins impose a mechanical checkpoint on macrophage phagocytosis. *Nat. Comm.* 15:8182 (2024).